# Large-scale drivers of Caucasus climate variability in meteorological records and Mt Elbrus ice cores

Anna Kozachek[1,2,3], Vladimir Mikhalenko[2], Valérie Masson-Delmotte[3], Alexey Ekaykin[1,4], Patrick Ginot[5,6], Stanislav Kutuzov[2], Michel Legrand[5], Vladimir Lipenkov[1], Susanne Preunkert[5]

1. Climate and Environmental Research Laboratory, Arctic and Antarctic Research Institute, St Petersburg, 199397, Russia
2. Institute of Geography, Russian Academy of Sciences, Moscow, 119017, Russia
3. Laboratoire des Sciences du Climat et de l'Environnement, CEA/CNRS/UVSQ/IPSL, Gif-sur-Yvette, 91191, France
4. Institute of Earth Sciences, St Petersburg State University, St Petersburg, 199178, Russia
5. Laboratoire de Glaciologie et Géophysique de l'Environnement, CNRS/UGA, Grenoble, 38400, France
6. Observatoire des Sciences de l'Univers de Grenoble, IRD/UGA/CNRS, Grenoble, 38400, France

*Correspondence to:* Anna Kozachek (kozachek@aari.ru)

## Abstract

A 181.8 m ice core was recovered from a borehole drilled into bedrock on the western plateau of Mt Elbrus (43°20'53.9'' N, 42°25'36.0'' E; 5115 m a.s.l.) in the Caucasus, Russia, in 2009 (Mikhalenko et al., 2015). Here, we report on the results of the water stable isotope composition from this ice core with additional data from the shallow cores. The distinct seasonal cycle of the isotopic composition allows dating by annual layer counting. Dating has been performed for the upper 126 m of the deep core combined with 20 m from the shallow cores. The whole record covers 100 years, from 2013 back to 1914. Due to the high accumulation rate (1380 mm w.e. per year) and limited melting we obtained isotopic composition and accumulation rate records with seasonal resolution. These values were compared with available meteorological data from 13 weather stations in the region, and also with atmosphere circulation indices, back-trajectory calculations and GNIP data in order to decipher the drivers of accumulation and ice core isotopic composition in the Caucasus region. In the warm season (May-October) the isotopic composition depends on local temperatures, but the correlation is not persistent over time, while in the cold season (November–April), atmospheric circulation is the predominant driver of the ice core's isotopic composition. The snow accumulation rate correlates well with the precipitation rate in the region all year round, which made it possible to reconstruct and expand the precipitation record at the Caucasus highlands from 1914 till 1966, when reliable meteorological observations of precipitation at high elevation began.

## 1 Introduction

Large-scale modes of variability such as the NAO (North Atlantic Oscillation) are known to influence European climate variability (see review in Panagiotopulos et al., 2002). However, most studies of large-scale drivers of European climate

change have been focused on low elevation instrumental records from weather stations, and there is very limited information about climate variability at high altitudes, and about differences in climate variability and trends at different elevations (EDW research group, 2015). Such differences were calculated in many mountain regions (EDW research group, 2015), except for the Caucasus, due to the lack of high elevation instrumental observations in this region.

The Caucasus is located southwards of the East European Plain. It is a high mountain region, with typical elevations of 3200-3500 m a.s.l., and with the highest point reaching 5642 m for Elbrus. The Main Caucasus Ridge acts as a barrier between subtropical and temperate mid-latitude climates, as observed for other high mountain regions such as the Himalaya. As in other mountain regions, there is a lack of high elevation meteorological records in the Caucasus. Moreover, existing records are relatively short: for example, reliable Caucasus precipitation measurements only started in 1966. Improved spatio-temporal coverage is required to investigate internal variability, to explore trends and spatial differences, and to evaluate the skills of atmospheric models providing atmospheric analysis products where no meteorological data are assimilated.

Measurements of the stable isotope composition of water, and annual accumulation rates in mid to high latitude ice cores are widely used proxies to estimate past temperature and precipitation rate changes. In many high mountain regions such as the Caucasus, and for elevations situated above the tree line, ice core data provides the only source of detailed information to document past climate changes, complementing punctual information retrieved from changes in glacier extent and recent glacier mass balance. For example, a study of the water stable isotope composition of several ice cores obtained in the Alps was recently conducted by Mariani et al. (2014) and the same research in Alaska was performed by Tsushima et al. (2015). The authors explored the links between the ice cores' isotopic composition, local climate, and large-scale circulation patterns. They found that in mountain regions, the isotopic composition of the ice cores was governed both by local meteorological conditions and by regional and global factors. These studies discussed the complexity of interpreting ice core records from high-altitude glaciers due to the potential bias from post-depositional processes and frequent changes in the origin of moisture sources. For instance, even in areas without any seasonal melt, accumulation is the net effect of precipitation, sublimation, and wind erosion processes, and may significantly differ from precipitation. Water stable isotope records are in mid to high latitudes physically related to condensation temperature through distillation processes (Dansgaard, 1964), but the climate signal is archived through the snowfall deposition and post-deposition processes. One important artefact lies in the intermittency of precipitation, and the covariance between condensation temperature and precipitation, which may bias the climate record towards one season, or towards one particular weather regime, challenging an interpretation in terms of annual mean temperature (Persson et al., 2011). Moreover, water stable isotopes are integrated tracers of all phase changes occurring from evaporation to mountain condensation, and are also affected by non-local processes related to evaporation characteristics, or shifts in initial moisture sources. Such processes have the potential to alter the validity of an interpretation of the proxy record in terms of local, annual mean, or precipitation-weighted temperature. In some regions, isotopic records are more related to hydrological cycles, recycling, or rainout (Aemisegger et al., 2014). Finally, the condensation temperature may also strongly differ from surface air temperature; depending on elevation shifts in e.g. planetary boundary layer or convective activity (see Ekaykin and Lipenkov, 2009 for a review). While these processes

make the interpretation of ice core records complex, they do open the possibility that the ice core proxy record may be in fact more sensitive to large-scale climate variability than punctual precipitation amounts. For instance, Casado et al (2014) have evidenced a strong fingerprint of the NAO in water stable isotope records from central Western Europe and Greenland, either in long instrumental records based on precipitation sampling, in seasonal ice core records, or in atmospheric models including water stable isotopes. The connection of Greenland ice cores' isotopic composition with atmospheric circulation patterns was studied by Vinther et al. (2003 and 2010). The strong influence of the NAO pattern on the Greenland ice cores' isotopic composition has been discovered and the possibility to use the ice core data for the reconstruction of the past NAO changes was suggested (Vinther et al., 2003). The authors also revealed the importance of the study of the seasonally resolved ice cores records rather than annual records, as there are different factors governing formation of the isotopic composition of precipitation in warm and in cold seasons (Vinther et al., 2010).

We will now briefly review earlier studies performed on climate variability in the Caucasus area, which have already explored the relationships between regional climate, glacier expansion, and large-scale modes of variability: the NAO (North Atlantic Oscillation), AO (Arctic Oscillation), and NCP (North Sea–Caspian Pattern). For example, Shahgedanova et al. (2005) monitored the mass balance of the Djankuat glacier, situated at an altitude between 2700 and 3900 m a.s.l. While no significant correlation was identified between the accumulation rate and the winter NAO index, the years of high accumulation systematically occurred during winters with a very negative NAO index. Brunetti et al. (2011) explored the influence of the NCP mode on climate in Europe and around the Mediterranean region. They evidenced a negative correlation coefficient of -0.50 between temperature in the Caucasus and the NCP index. Baldini et al. (2008) investigated records of precipitation isotopic composition in Europe from the IAEA/GNIP stations, extrapolating a significant negative correlation between winter precipitation $\delta^{18}O$ in the Caucasus region and the NAO index (R = −0.50). Casado et al (2013) studied the influence of precipitation intermittency on the relationships between precipitation $\delta^{18}O$, temperature, and the NAO. The influence of the NAO index on European climate and precipitation $\delta^{18}O$ appeared more prominent in winter than in summer (Comas-Bru et al., 2016).

Here, we take advantage of the new Elbrus deep ice cores (Mikhalenko et al., 2015), and produce the first analysis of water stable isotope and accumulation records. Section 2 introduces the data and methods, with a description of the ice core analyses and age scale, an overview of regional meteorological information, and the source of information for indices of modes of variability. Section 3 presents the results of the comparison and statistical analyses of the relationships between regional climate parameters (temperature and precipitation), Elbrus ice core records, and modes of variability. In section 4, we summarize our key findings and the next steps envisaged to strengthen the climatic interpretation of the Caucasus ice core records.

**2 Data and methods**

**2.1 Ice core data**

### 2.1.1 Drilling site and drilling campaigns

Here, we report on results from the new, deepest ice core from Mt Elbrus, in comparison with results from shallow ice cores. Deep drilling was performed on the Western Plateau (43°20'53.9" N, 42°25'36.0" E; 5115 m a.s.l.) of Mt Elbrus (fig. 1) in September 2009, allowing recovery of a 181.8 m long ice core, down to bedrock. The drilling site and the drilling operations are thoroughly described in Mikhalenko et al. (2015).

In order to update the ice core records towards the present day, and enable a comparison of the measurements with local meteorological monitoring data, surface drilling operations were repeated at the same place in 2012 (11.5 m long) and in 2013 (20.5 m long). Results are also compared here with previously published isotopic composition data measured along the 22 m shallow ice core drilled at the same place in 2004 which covered the period from 1998 till 2004 (Mikhalenko et al, 2005).

In 2014, drilling operations were also successful at the Maili Plateau (Mt Kazbek), at the altitude of 4500 m a.s.l. in 200 km eastwards from Elbrus (fig. 1), delivering a 20-m ice core. The Kazbek core is shown for purposes of comparison only. A detailed description of it will be published elsewhere.

### 2.1.2 Sampling process and sampling resolution

For the upper and the lower parts of the deep core (0-106 m and 158-181.8 m) and for the shallow firn cores drilled in 2012 and 2013, sampling was performed using classic cutting-melting procedures. For the other depth intervals, melted samples were extracted from the continuous flow analysis system of LGGE (Grenoble, France), automatically sub-sampled, frozen and stored in vials for subsequent isotopic analysis. The description of the CFA system will be published elsewhere.

The sampling resolution was 15 cm for the upper 16 m of the deep core (see the sketch of the sampling resolution in fig. 2c). It was then increased to 5 cm in order to achieve better resolution, from 16 to 70 m depth and in the bottom part of the core (158-182 m depth). To ensure 15-20 samples per year, the sampling resolution was increased to 4 cm in the depth range from 70 to 106 m, similar to the sampling resolution of the CFA system (3.7 cm).

Samples from the shallow cores drilled in 2012 and 2013 were cut with a resolution of 10 and 5 cm, respectively.

### 2.1.3 Isotopic measurements

The methods for the isotopic measurements have been partially discussed in (Mikhalenko et al., 2015). Water stable isotope ratios ($\delta^{18}O$ and $\delta D$) were measured at the Climate and Environmental Research Laboratory (CERL) at the Arctic and Antarctic Research Institute (St Petersburg, Russia), using a Picarro L2120-i analyzer. Each sample was measured once. Sequences of measurements included the injection of 5 samples, followed by the injection of an internal laboratory standard

with an isotopic value close to that of the samples. We also repeated the measurements of about 10% of all the samples in
order to calculate the analytical precision: 0.06‰ for $\delta^{18}O$ and 0.30‰ for $\delta D$. The depth profile of $\delta^{18}O$ (Mikhalenko et al.,
2015; Kozachek et al., 2015) and of the deuterium excess ($d= \delta D - 8*\delta^{18}O$) are shown in fig. 2.
Moreover, 600 samples from the depth interval from 23 to 35 m were measured in the Laboratory of Isotope Hydrology of
the IAEA (Vienna, Austria). The two records are highly correlated (r=0.99, p < 0.05) for both isotopes (Figure S2b) with a
systematic offset of 0.2 ‰ for $\delta^{18}O$ and 1 ‰ for $\delta D$. The records of the second order parameter deuterium excess are also
significantly correlated (r=0.65, p < 0.05) without any specific trend or systematic offset. This inter-laboratory comparison
demonstrates the high quality of the isotopic measurements performed in CERL.
We also stress the close overlap of the upper part of the profiles of the water stable isotope records versus depth from the
different cores drilled in 2009, 2012 and 2013 (Fig. S2a). Based on this close agreement within the different shallow firn
cores, we decided to calculate a stack record for the period from 1914 till 2013, which is used for dating hereafter.
In the depth interval from 100 to 106 m depth, we also have an overlap of samples obtained with classic cutting method and
CFA method described above, without any significant difference (Fig. S2c), again allowing us to combine the two records
into one stack record.
**2.1.4 Dating**
The chronology is based on the identification of annual layers. These are prominent in $\delta^{18}O$ with the average seasonal
amplitude of 20 ‰. For annual mean values we calculated averages of $\delta^{18}O$ from one minimum of this parameter to another
one as well as from one maximum to another. As we found no significant differences between the records obtained with two
ways of year allocation we used minimum to minimum dating as a more common method. We compared annual layer
counting performed independently using the seasonal cycles in the isotopic composition and the ammonium concentration.
The discrepancy between two independent chronologies is 2 years at a depth of 126 m. We used the dating based on the
isotopic composition data in this paper. This dating is also best fit for the correlation analysis with the meteorological data.
For the estimation of the dating uncertainties we used the absolute age markers. These markers are the tritium peak in 1963
and the sulfate peak in 1912 which corresponds to the Katmai eruption (Mikhalenko et al., 2015). The comparison of
different dating methods on age control points shows that the overall error of our timescale at these two depth levels does not
exceed ±2 years which means that independent dating uncertainties should compensate each other at this points
Hereafter, we focus our analysis on one hundred years, from 1914 till 2013, which corresponds to the total of 140 m of the
ice thickness studied here (the 15 m covered by the shallow cores plus the 126 m covered by the deep ice core). This period
has been chosen because at this depth, the age scale is well defined by the time horizon found slightly below (Katmai 1912)
resulting in a relatively small dating uncertainty of ±2 years, and because of the availability of other records such as local
meteorological observations. In the bottom part of the core the cycles in the isotopic composition are less prominent and
dating becomes less reliable, leading to a significant increase in uncertainty. The isotopic composition of that part of the core
will be discussed elsewhere. In meteorological data we used average values from January to December of each year for the
comparison with the annual means of ice cores parameter.
For warm and cold seasons allocation, we used a method adapted slightly from (Vinther et al., 2010). The original method
requires ascribing of an equal accumulation rate for the warm and cold season of each year. Basically we used the same
approach as there is an obvious seasonal cycle of $\delta^{18}O$ which is coherent with the seasonal cycle of temperature in the region.
We assume that the maximum value of $\delta^{18}O$ in the annual cycle corresponds to July and the minimum value corresponds to
January and put the border so that these extreme values are in the middle of a season. This method is based on two
assumptions. Firstly, the months of the most extreme temperature lie in the middle of the corresponding season. Secondly,
the validity of the first assumption does not change over time. Both assumptions are confirmed with the weather
observations in the region. The middle of the warm season is the end of July-beginning of August. During the whole period
of observation the maximum temperature was observed outside this period in 1969 only, when the maximum temperature
was in June. In the cold season the middle of the season is the end of January–beginning of February. The minimum values
of temperature were observed outside this period in 1971, 1985, 1995, and 1997. We therefore consider the first assumption
being valid for the whole period of time discussed in the paper. .. We also used ammonium concentration as an independent
marker, using criteria described on (Mikhalenko et al., 2015). For equivocal situations, we also used additional data: melt
layers and dust layers (used to identify the warm season) (Kutuzov et al., 2013) as well as succinic acid concentration data
that also have seasonal variations (Mikhalenko et al., 2015).
Figure 3 illustrates the identification of seasons using the isotopic composition seasonal cycle. In the meteorological data we
used the period from November to April for the cold season and May to October for the warm season.
There some gaps in the isotopic composition data that came from technical problems during the drilling operations and the
process of analysis. The drilling problems are described in (Mikhalenko et al., 2015). The biggest gap appears at the depth of
31.3 and 32.1 m. A piece of the core was lost during the drilling operations. This part is covered by the bottom part of the
2004 core where the sampling resolution was 50 cm. It is evident that two seasons (one warm and one cold) are partially
missing. We did not use these values for the correlation analysis because of the large uncertainty of the seasonal values
calculations in this case. In case of a missing sample we considered its isotopic value to be the average between the two
neighboring samples. For a detailed description of the raw isotopic data and annual layers allocation for the upper 106 m of
the core, please refer to Mikhalenko et al. (2015). Mean annual and seasonal values of $\delta^{18}O$ and $d$ obtained as a result of the
dating are shown in fig. 5 and 6 respectively.
The annual accumulation rate is calculated as the thickness of the seasonal layer, multiplied by the layer density using the
density profile from Mikhalenko et al. (2015), and corrected for layer thinning using the Nye model (Nye, 1963; Dansgaard
and Johnsen, 1969), with the following parameters: accumulation rate 1.583 m of ice equivalent, pore close-off depth = 55 m
(Mikhalenko et al., 2015).

**2.1.5 Diffusion of stable isotopes**

We calculated the potential influence of diffusion on the stable isotopes record according to the (Johnsen, 2000) model. We used the following parameters for the calculation: Our calculation showed that the seasonal amplitude of $\delta^{18}O$ variations could be 10-20% less because of the diffusion (Mikhalenko et al., 2015). If it was the case we would observe a decreasing of $\delta^{18}O$ maxima and increasing of minima with depth. Moreover we would find a positive correlation between layer thickness and a seasonal amplitude of $\delta^{18}O$. These features have not been found in the ice core data. The correlation coefficient between seasonal amplitude and accumulation rate is -0.10 and is statistically insignificant. There is also no statistically significant trend in the seasonal amplitude; the seasonal amplitude varies stochastically from 10 to 25 ‰. The maximum value observed in 1984 and the minimum in 1925. We therefore consider that the diffusion does not sufficiently influence the isotopic composition record in the upper 126 m of the ice core. At the bottom part of the core (e.g. at a depth of 180 m) the annual cycle of $\delta^{18}O$ should have an amplitude of 4 ‰ which is detectable but the length of the cycle should be less than 1 cm. As the $d$ annual cycle is not prominent we cannot used the method based on the discrepancy between the $\delta^{18}O$ and $d$ cycles. Thus, for obtaining climatic information from the bottom part of the core, a very high sampling resolution is required.

**2.2 Meteorological data**

We used the daily meteorological data (precipitation rate and mean daily temperature) from several weather stations around the drilling site (see map in Fig. 1 and Table 1) for comparison with the ice core data. We also investigated records of precipitation isotopic composition based on monthly sampling, performed at three stations to the south of the Caucasus within the WMO-IAEA Global Network of Isotopes in Precipitation (GNIP) program (Table 1).

For comparison we used the NCEP/NCAR reanalysis temperature data (Kalnay et al., 1996) for the 500 mbar level which corresponds to the drilling site altitude. Two different models were used to calculate back trajectories: FLEXPART (Forster et al., 2007, Stohl et al., 2009), HYSPLIT (Draxler, 1999, Stein et al., 2015, Rolph, 2016). The LMDZiso model was used to estimate the precipitation isotopic composition at the drilling site (Risi et al., 2010).

**2.3. Circulation indices**

Circulation of the atmosphere sufficiently influences isotopic composition of the ice cores (Casado et al., 2013 and references therein). Atmospheric circulation is quantitatively characterized by circulation indices. In this research we used three indices: NAO, AO, and NCP, that are widely used to characterize European climate (Jones et al., 2003, Thompson and Wallace, 2001, Brunetti et al., 2011 and references therein). Time span and references for the indices are presented in table 1. NAO (North-Atlantic Oscillation) characterizes the type of circulation in Europe, strength of Azores maximum and Icelandic minimum. The positive values of the NAO index correspond to the lower than usual value of the atmospheric pressure in Iceland and the higher than usual value of atmospheric pressure at Azores. The negative index corresponds to the less prominent centres of action in the Northern Hemisphere. Usually this index is calculated as a difference of atmospheric

pressure measured at Reykjavik and Lisbon, Ponta Delgada or Gibraltar. Here we used data from (Vinther et al., 2003 and
https:\\crudata.uea.ac.uk\~timo\datapages\naoi.htm) that were calculated using data from Gibraltar station. The negative
NAO leads to an increase in the precipitation rate in Southern Europe, while a positive NAO leads to an increase in the
precipitation rate in Northern Europe (Hurrel, 1995, Jones et al., 2003, Vinther et al., 2003).
The Arctic Oscillation index (AO) is also a characteristic of the Northern Hemisphere circulation. It is used to analyze
climatic variability with periods longer than 10 years. It is calculated as EOF of 500 hPa surface. Negative values correspond
to high pressure at the Pole and the cooling of Europe, while positive values correspond to low pressure at the Pole and the
drying of the Mediterranean (Thompson and Wallace, 2001). We used AO data from NOAA
(http:\\www.cpc.ncep.noaa.gov\products\precip\CWlink\).
The NCP (North-Sea Caspian Pattern) index is less widely used, though it was proved that it is convenient to use it in
Mediterranean climate studies (Kutiel et al., 1997; Brunetti et al., 2011). The index is calculated as a normalized difference
of geopotential heights between the Caspian and Northern seas. Positive values correspond to stronger meridional circulation
in Europe and lower summer temperatures, while negative values reflect the strengthening of zonal circulation and higher
summer temperatures in Europe (Brunetti et al., 2011). We used NCP data from NOAA
(http:\\www.cpc.ncep.noaa.gov\products\precip\CWlink\).

**3 Results**

**3.1 Regional climate**

The main peculiarity of the drilling site is its location on the border between subtropical and temperate climatic zones
(Volodicheva, 2004). Back-trajectory calculations show that the drilling site is characterized by remarkable seasonal
differences in the locations of moisture sources. In winter, the origin of air masses varies from the Mediterranean to the
North Atlantic. In summer, local moisture sources from the surrounding continents or from the Black Sea are predominant
(see fig. S1 for examples).
Meteorological data depict large regional variations in the seasonal cycle of precipitation. To the south of the Caucasus, there
is no distinct seasonal cycle (Fig. 4a), showing the climatology for the Klukhorsky Pereval station. In fact, the Klukhorsky
Pereval station is situated north of the Main ridge, but in terms of the seasonal cycle of precipitation it undoubtedly belongs
to the southern group. However, we are nevertheless using this station as an example because of the uninterrupted record of
temperature and precipitation for the 1966-1990 period. By contrast, the north of the Caucasus is marked by a distinct
seasonality in precipitation amounts, which are maximum in summer and minimum in winter (Fig. 4b), showing the
climatology for the Mineralnye Vody station. More examples of the Caucasus weather stations climatologies are given in
(Mikhalenko et al., 2015). Moreover, the annual precipitation rate to the south of the Caucasus is much higher than to the
north. For example, the typical annual precipitation rate to the north of the Caucasus at an altitude close to sea level is 500

mm per year, while to the south of the Caucasus at the same altitude it is about 1500 mm. The amount of precipitation in the region is affected by the altitude and the distance from the sea shore.

The seasonal changes of temperature appear uniform throughout the region surrounding the Caucasus, with the warmest conditions observed in summer and the coldest observed in winter. The seasonal amplitude depends on the distance from the sea and the mean annual temperature depends on the altitude. The average regional lapse rate was calculated using the available meteorological data. We used the data from all the stations for the calculation. The lapse rate is lowest in December-February (2.3°C per 1000 m) and highest (5.2 °C per 1000 m) in June-August (Fig. S3).

Based on the lapse rate, we calculated the temperature at the drilling site taking into account its seasonal variability shown on the fig. S3. This record was used for the estimation of the $\delta^{18}$O-temperature relationship. For the comparison with the ice core data we used the dataset of the normalized temperature data. Normalized temperature time series were calculated for each station for each season or for the whole year, and results were then averaged (fig. 8). For precipitation data, available in this region since 1966, we show all the data (fig. S4), while in the calculations we used data from Klukhorsky Pereval station as an example of a station without a seasonal cycle, and from Mineralnye Vody station as an example of one with a prominent cycle. More examples of annual variations of temperature and precipitation at the Caucasus meteorological stations can be found in (Shahgedanova et al., 2014) and (Tielidze, 2016). At our drilling site, an automatic weather station (AWS) provided in situ measurements for the period from August 2007 till January 2008. The day to day variations of temperature at low elevation weather stations and at the AWS are coherent for the whole period of the AWS work (Mikhalenko et al., 2015).

We also compared the data from meteorological stations with the NCEP reanalysis (Kalnay et al., 1996) outputs (not shown) for the 500 mbar level. Despite the difference in absolute values on a daily scale when compared with the AWS data (the difference is random and varies from -1 to 1 °C), the observed regional data and reanalysis data have the same month to month variability. The maximum daily mean temperature at the drilling site according to the reanalysis data was -1.3°C for the whole dataset. The temperature in the glacier at 10m depth, which corresponds to the annual mean temperature at the drilling altitude, is -17°C (Mikhalenko et al., 2015), the annual mean temperature at the drilling altitude from the NCEP reanalysis is -14 °C, and the same value calculated from meteorological observations and corrected for the lapse rate is -11 °C.

We then investigated long-term trends in the meteorological records. Mean annual temperatures show a significant increase during the last two decades. We also observe higher than average values of mean decadal temperature in 1930-1940. And the beginning of the observations in the region, i.e. the period from 1881 till 1900, was as cold as the 1990s. It is evident that the last 20 years in the warm season were the warmest for the whole observation period (fig. 8), while in the cold season the recent warming is not unprecedented. For example, cold seasons in the 1960s–1970s were even warmer (fig. 8). Multi-decadal patterns of temperature variations also differ in the late 19[th] century, where negative anomalies are identified in cold season temperature (Fig. 8) but not in warm season temperature (Fig 8). On the other hand in cold season temperatures we can observe lower temperatures at the end of the 19[th] century that might be due to the impact of the volcanic eruptions

(Stoffel et al., 2015). We also noted the high temperature values in the 1910s-1920s that are not completely understood. We
did not find any trends in the precipitation rate for any of the groups of stations (fig. S4).
A significant anti-correlation is observed between temperature and the NAO index, both in the cold and warm seasons
(Table 2, the information about the time series used for the correlation analysis can be found in Table 1). Stronger anti-
correlations are identified between temperature and the NCP index, especially in the cold season, as also reported by Brunetti
et al. (2011). Relationships with indices of large scale modes of variability are systematically weaker for precipitation, with
contradictory results for the south/north Caucasus stack; they appear significant for the NCP in both seasons (Table 2).
GNIP data are only available at low elevation stations. They show a rather uniform distribution of the isotopic composition
of precipitation in the region during summer, as well as a gradual depletion of $\delta^{18}O$ at higher altitudes in winter.
GNIP records are too short and intermittent (one-two years with gaps) to investigate the variability and relationships with the
local temperature on an interannual scale. We therefore restrict discussion of GNIP data to seasonal variations. The $\delta^{18}O$ and
$\delta D$ in precipitation have a distinct seasonal cycle with maximum values observed in the warm season (JJA) and minimum
values observed in the cold season (DJF). As an example we show the seasonal cycle of $\delta^{18}O$ and $d$ for Bakuriani station in
2009 (fig. 7). This station is the only one in the region for which the whole uninterrupted dataset for one annual cycle is
available. The seasonal amplitude of $\delta^{18}O$ is about 17 ‰. The slope between $\delta^{18}O$ and temperature is 0.32 ‰/°C. The $d$
variations show no seasonal cycle varying randomly between 10 ‰ and 25 ‰. We found no significant correlation between
$\delta^{18}O$ and $d$.
Climate variability as a driver for glacier variations in the Caucasus has recently been explored by several authors.
Elizbarashvili et al. (2013) found the increased frequency of extremely hot months during the 20th century, especially over
Eastern Georgia, whereas the number of extremely cold months decreased faster in the Eastern than in the Western region. In
addition, the highest rates for positive trends of annual mean air temperature can be observed in the Caucasus Mountains.
Shahgedanova et al. (2014) evidenced significant glacier recession at the northern slopes of the Caucasus, consistent with
increasing air temperature of the ablation season. They report that the most recent decade (2001-2010) was 0.7–0.8 °C
warmer than in 1960-1986 at Terskol and Klukhorsky Pereval stations (see Table 1 for information on stations). However,
the warmest decade for JJA was 1951-1960 (Shahgedanova et al., 2014). Tielidze (2016) reports a recent increase in the
annual mean temperatures at different elevations in the Georgian Caucasus. The region experienced glacier area loss over the
20[th] century at an average annual rate of 0.4% with a higher rate in eastern Caucasus than in the central and western sections.
The analysis of temperature and radiation regime of glaciers at the ablation period has been performed at Elbrus vicinities
recently (Toropov et al., 2016). The authors prove that the observed waning of glaciers cannot be explained by an increase in
temperature during the ablation period because of an increase in precipitation during the accumulation period. They
concluded that the main driver of glacier retreat is the increase of the solar radiation balance for 4% for the 2001-2010 period
which corresponds to the increase of ablation for 140 mm per ablation season (Toropov et al., 2016).

**3.2 Ice core records**
The comparison of the four cores obtained at the Western Plateau of Elbrus shows similar variations during overlap periods
(see Fig. 2S). We therefore calculate a stack record for each season, based on the average value of individual ice cores for the
overlapping seasons. The inter-core disagreement is almost negligible (fig. 2S) and can be explained by different sampling
resolution.
We note that the shallow ice core from the Maili plateau of Kazbek shows the same mean values of $\delta^{18}O$ as the Elbrus ice
cores during their overlap period. This is a result of a mutual compensation of $\delta^{18}O$ increase due to the lower elevation
position (Kazbek drilling site is 500 m lower) and of $\delta^{18}O$ decrease because of the continentality effect (Kazbek is 200 km
further from the sea). We calculated the continental gradient and lapse rate for $\delta^{18}O$ using the data from the GNIP stations in
the region that are situated at the lower elevations. The lapse rate is -0.25 ‰/100 m and continental gradient is -0.85 ‰ /100
km. The mean value of $\delta^{18}O$ for the Kazbek ice core should be 1.25‰ more positive because of elevation difference and
1.7‰ more negative due to the continentality factor.
The inter-annual variability in isotopic composition is about twice larger in the cold season than in the warm season for $\delta^{18}O$.
Different patterns of inter-annual to multi-decadal variations appear in the instrumental temperature data (see section 3.1)
and ice core $\delta^{18}O$ records (Fig 5) emerge for the cold versus the warm season.
The $\delta D$ and $\delta^{18}O$ values are highly correlated (r = 0.99) on a sample to sample scale so hereafter we use the $\delta^{18}O$ information
for the dating and comparison with the other parameters. The slope between $\delta^{18}O$ and $\delta D$ is 8.03 on sample to sample scale
and 7.9 on a seasonal scale without any significant difference between the two seasons.
No significant (R squared is insignificant at p<0.05) centennial trend is identified in the cold/warm season $\delta^{18}O$, nor in the
cold/warm season accumulation rate or deuterium excess. We observe large variations in $\delta^{18}O$ with high and variable values
in the early 20[th] century, lower and more stable values in the 1940s-1960s, and a step increase in the 1970s with another
level. These variations are coherent in both seasons as well as in annual means but are not reflected in the meteorological
observations. There is also an increase of $\delta^{18}O$ in the last two decades in both seasons in regard to the 1970s-1980s values
but the absolute values of $\delta^{18}O$ are close to the multiannual seasonal averages (Table 3). The highest decadal values of $\delta^{18}O$
in both seasons are observed in 1912-1920. While a recent warming trend is observed in the regional meteorological data (in
warm season), it is much less prominent in the ice core $\delta^{18}O$ record, suggesting a divergence between $\delta^{18}O$ and regional
temperature. One of the possible explanations for this feature is the post-depositional change of the isotopic composition.
But we do not expect a significant influence of the post-depositional processes because of the high snow accumulation rate.
The highest $\delta^{18}O$ values for a single year correspond to the warm periods of 1984 and 1928, two years for which no unusual
feature is identified from meteorological observations. The highest snow accumulation rate (fig. 9) is observed in both
seasons of 2010, in coherence with the meteorological precipitation data, and also corresponding with a record low winter
NAO index.
Our deuterium excess record (fig. 2b) does not depict any robust seasonal variation. Moreover, the distribution of deuterium
excess as a function of $\delta^{18}O$ does not display any clear structure. By contrast, deuterium excess is weakly positively
correlated with the accumulation rate during the warm season (r = 0.31, p<0.05). This finding is consistent with the GNIP
data in the region that show no link between $\delta^{18}$O and deuterium excess. The smoothed values of deuterium excess have
prominent cycles with a period of about 25 years that are synchronous in both seasons (fig. 6). Deuterium excess is highly
sensitive to surface humidity, which itself is very different and depends on the arrival of maritime air masses or dry
continental air masses. This may add to the complexity of the deuterium excess signal (Pfahl and Wernli, 2008).

**3.3 Comparison of ice core records with regional meteorological data**

We compared the ice core data with the regional meteorological data and the large-scale modes of variability. The result of
the correlation analysis is summarized in Table 4. Multiannual variations of the parameters are shown in fig. 9 for the cold
season and in fig. 10 for the warm season.
We found no significant correlation between the ice core $\delta^{18}$O record and regional temperature, neither with the reanalysis
data, nor with the observation data, when using the whole period. A significant correlation (r = 0.44, p<0.05) emerges for
warm season data, when calculated for the period since 1984. The slope for this period is 0.6 per mille per °C. We also
repeated our linear correlation analysis using precipitation weighted temperature, and obtained the same results. The
precipitation weighted temperature was calculated using daily meteorological data. We used data from two stations:
Klukhorsky Pereval (as a representative of the southern stations) and Mineralnye Vody (as a representative of the northern
stations).
Obviously, the above inferences strongly depend on the uncertainties of the timescale used. If one concedes that the error of
the timescale could be significantly greater than ±2 year, quite different conclusions may be reached by adjusting the scale
of the $\delta^{18}$O and T records against each other. For instance, by contracting the $\delta^{18}$O record by 8 years with respect to the
initial timescale in Figs 9 and 10, one would find much better correlation between $\delta^{18}$O and temperature, thus reaching the
conclusion that the local temperature is the main driver of the $\delta^{18}$O variability. However, based on various experimental
evidences, as discussed in the dating section, we argue that the timescale developed for the Elbrus ice core is accurate within
±2 years. Therefore, the most realistic conclusion of those that can be drawn from the data obtained is that the temperature is
weakly correlated with the $\delta^{18}$O, and that this correlation is unstable in time.
We also did not find any statistically significant correlations when we compared 3-, 5-, 7-years running means of these
parameters. This result implies that the isotopic composition at Elbrus is controlled by both local and regional factors such as
changes in moisture sources. The possibilities for accurate reconstructions of past temperatures are therefore limited. For
more accurate investigation of the $\delta^{18}$O – temperature relation on-site experiments and subsequent modeling is required.
Our results are comparable to those obtained in the Alps by Mariani et al. (2014) for the Fiescherhorn glacier where the
authors found significant, though weak, correlation between temperature and $\delta^{18}$O. However for the Elbrus ice core this
correlation was found in the warm season only.

Another research performed in the Alps by Bohleber et al. (2013) revealed significant correlation of modified local temperature and the ice core isotopic composition at a decadal scale. The authors also report that there are some periods of correlation absence. The main finding is that for the periods of less than 25 years the difference between the modified dataset according to the authors' method and original dataset temperature is crucial, but for longer periods the two temperature datasets are close to each other. That conclusion implies that the isotopic composition reflects the local temperature in the high mountain regions to a limited extent. It seems to be impossible to calculate the modified temperature for the Caucasus region according to the methods described by Bohleber et al. (2013) because of the relatively short and sparse original datasets.

The seasonal accumulation rate is seasonal layer thickness corrected for densification using the density profile from Mikhalenko et al. (2015) and for the layer thinning due to glacier flow using the Nye model (Nye, 1963; Dansgaard and Johnsen, 1969). It is linked to the precipitation rate on the stations situated south of the Caucasus in both seasons (r = 0.49), and even more closely related to precipitation from Klukhorsky Pereval station (r = 0.63 for both seasons). We therefore establish a linear regression model for the period 1966-2013, and use this methodology to reconstruct past precipitation rates for the Klukhorsky Pereval station (1914-1965), when meteorological records are not reliable or unavailable. The reconstructed records are shown on fig. 9 and 10 for the cold and warm seasons respectively. We found no significant trend in the reconstructed precipitation values. Even so, these results may be useful for validation of regional climate models and water resource assessment.

Calculation of the seasonal cycle of precipitation isotopic composition using the LMDZiso model (Risi et al., 2010) do not correspond to the results obtained from the ice core in absolute values or in amplitude (Fig. S5). This can be explained by a complicated relief of the region that strongly influences the isotopic composition, but it is not taken into account in the model. Also, in summer, Elbrus is in a local convective precipitation system that is not included in the model.

**3.4 Comparison of ice core records with large-scale modes of variability**

We did not find any statistically significant correlations between ice core data and large scale modes of variability when using the mean annual values. We present the results of calculations in the table 4. We report a weak though significant (p<0.05) negative correlation (r = − 0.18) between the ice core accumulation rate record and NAO in the cold season. Moreover, the year of extremely high accumulation in both seasons (2010) coincides with an extremely low NAO winter index. The role of NAO in regional climate had also been evidenced by Shahgedanova et al. (2005) for the mass-balance of the Djankuat glacier situated in 30 km south-east of Elbrus for the period of 1967-2001. Interestingly, the accumulation record is related to the variability of regional precipitation, but the latter is not significantly related to the NAO. This may suggest different influences of large-scale atmospheric circulation on precipitation at lower versus higher elevations.

For the cold season, the ice core $\delta^{18}O$ record shows a positive correlation with the NAO index (r = 0.41), while the NAO index is negatively correlated with regional temperature (r = − 0.42). It also contradicts the findings of Baldini et al (2008)

who, based on the GNIP low elevation dataset, extrapolated a negative correlation between the $\delta^{18}O$ of precipitation and the
NAO in this region. This finding also suggests different drivers of temperature and $\delta^{18}O$ at low and higher elevation. We
propose the following explanation for this correlation. During the positive NAO phase, the predominant moisture source for
the Caucasus precipitation is the Mediterranean. During the negative NAO phase the moisture source is the Atlantic. In the
first case the precipitation $\delta^{18}O$ preserved in the ice core is higher because of the higher initial sea water isotopic composition
(Gat et al., 1996) and the shorter distillation pathway. The continental recycling of moisture (Eltahir and Bras, 1996) also
influences the water isotopic composition. Due to this process the $\delta^{18}O$ values became lower while the $d$ values increase
(Aemisegger et al., 2014), which is observed in our ice core data. In the opposite situation the initial water isotopic
composition is close to 0 ‰ (Frew et al., 2000) and the distillation pathway is longer which leads to lower values of
precipitation $\delta^{18}O$.
We explored the links between the ice core parameters ($\delta^{18}O$, accumulation rate) with the NCP index and found no
significant correlation in winter, or in summer despite the significant correlation between the NCP and local temperature and
precipitation. A possible explanation may be that the NCP pattern only affects low elevation regional climate but not high
elevation climate.
No significant correlation was identified between deuterium excess and indices of large scale modes of variability. So far, no
regional or large-scale climate signal could be identified in Elbrus deuterium excess. Further investigations using back
trajectories and diagnoses of moisture source and evaporation characteristics will be needed to explore further the drivers of
this second-order isotopic parameter.

**4 Conclusion**

We found no persistent link between ice cores $\delta^{18}O$ and temperature on an interannual scale, a common feature emerging
from non-polar ice cores (e.g. Mariani et al., 2014). This finding is not an artefact of high elevation versus low elevation
difference, because the variability of the regional temperature stack used for this comparison is in good agreement with the
variability of the temperature at the drilling site as observed by the local AWS.
Our ice core records depict large decadal variations in $\delta^{18}O$ with high and variable values in the late 19[th]-early 20[th] centuries,
lower and more stable values in the 1940s-1960s, followed by a step increase in the 1970s. No unusual recent change is
detected in the isotopic composition or in the accumulation rate record, in contrast with the observed warming trend from
regional meteorological data. The accumulation rate appears significantly related to the NAO index coherently with the
earlier results for the Djankuat glacier (Shahgedanova et al. 2005).
Based on regional meteorological information and trajectory analyses, the main moisture source is situated not far from the
drilling site in the warm season, and consists of evaporation from the Black Sea and continental evapotranspiration. Changes
in regional temperature during the warm season may affect the initial vapour isotopic composition as well as the atmospheric
distillation processes, including convective activity, in a complex way. This may explain the significant, albeit non
persistent, correlation of summer $\delta^{18}O$ and temperature. Cold season moisture sources appear more variable geographically,
with potential contributions from the North Atlantic to the Mediterranean regions. Changes in moisture origin appear to
dominate in regional temperature-driven distillation processes. As a result, the isotopic composition of the ice cores appears
mostly related to characteristics of large–scale atmosphere circulation such as the NAO index. The changes in moisture
origin also influence the deuterium excess parameter, which does not have any prominent seasonal variations.
Our data can be used in atmospheric models equipped with water stable isotopes, for instance to assess their ability to
resolve NAO–water isotope relationships (Langebroek et al., 2011, Casado et al., 2014). The accumulation rate at the drilling
site is significantly correlated with the precipitation rate and gives information about precipitation variability before the
beginning of meteorological observations.

**Acknowledgements**

The research was supported by the RFBR grants 14-05-31102 mol_a and 17-05-00771 a. The analytical procedure ensuring a
high accuracy of isotope data obtained at CERL was elaborated with financial support from the Russian Science Foundation,
grant 14-27-00030. The study of dust layers was conducted with the support of RFBR grant 14-05-00137. The measurement
of the samples in IAEA was conducted according to research contracts 16184\R0, and 16795. This research work was
conducted in the framework of the International Associated Laboratory (LIA) "Climate and Environments from Ice
Archives" 2012–2016, linking several Russian and French laboratories and institutes. We thank Obbe Tuinenburg and Jean-
Louis Bonne for the back trajectory calculations. We thank Alice Lagnado for improving the English. We are grateful to four
anonymous reviewers and the Editor Professor Hou Shugui for their comments, which helped to improve the paper.

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

598  .

**Figures**

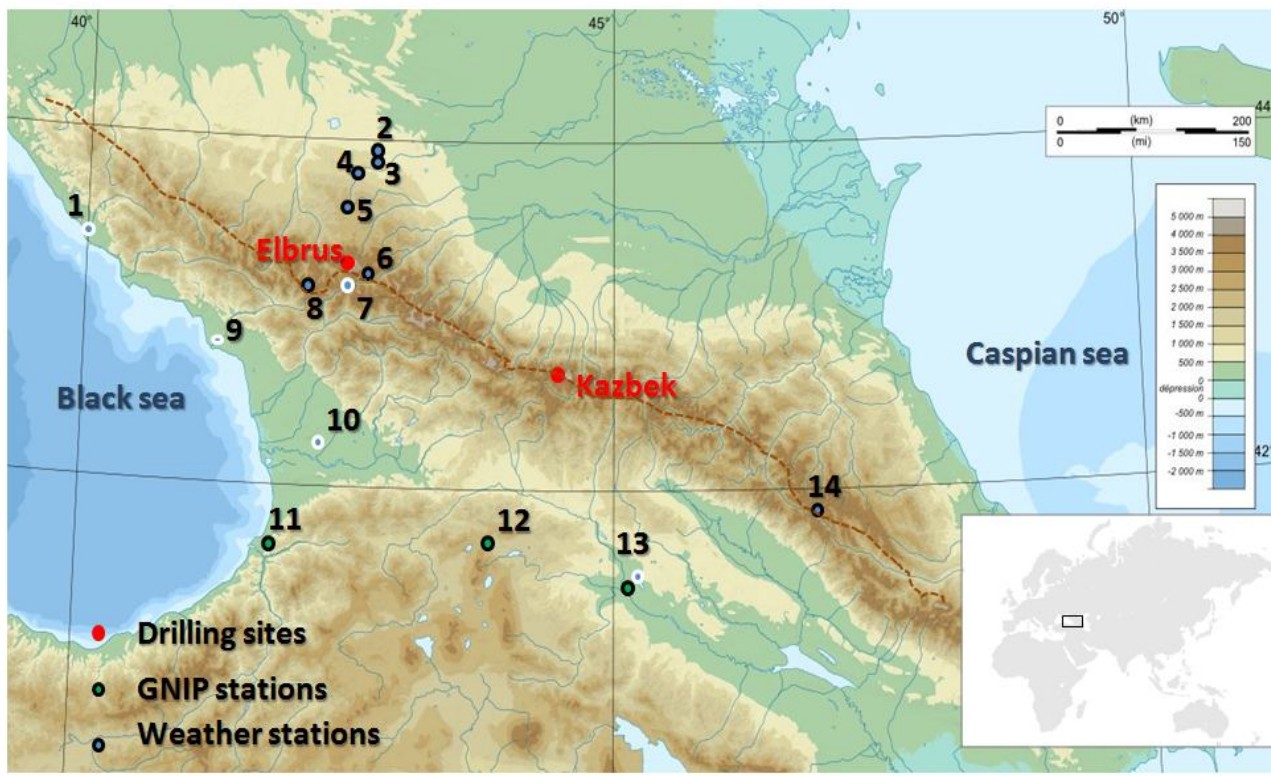

**Fig. 1: Map showing the region around Elbrus (black rectangle in the world's map in the lower right corner), with shading**
**indicating elevation (m above sea level). Drilling sites are indicated with red filled circles, GNIP stations as green filled circles, and**
**meteorological stations as blue dots. Stations situated to the south of the Main Caucasus Ridge according to the precipitation cycle**
**pattern are shown using a blue dot with white outside circle and the stations situated to the north are displayed with black outside**
**circle (see text for details). The brown dotted line shows the border between two types of precipitation seasonal cycles. The number**
**of the various stations refers to Table 1 for their detailed description.**

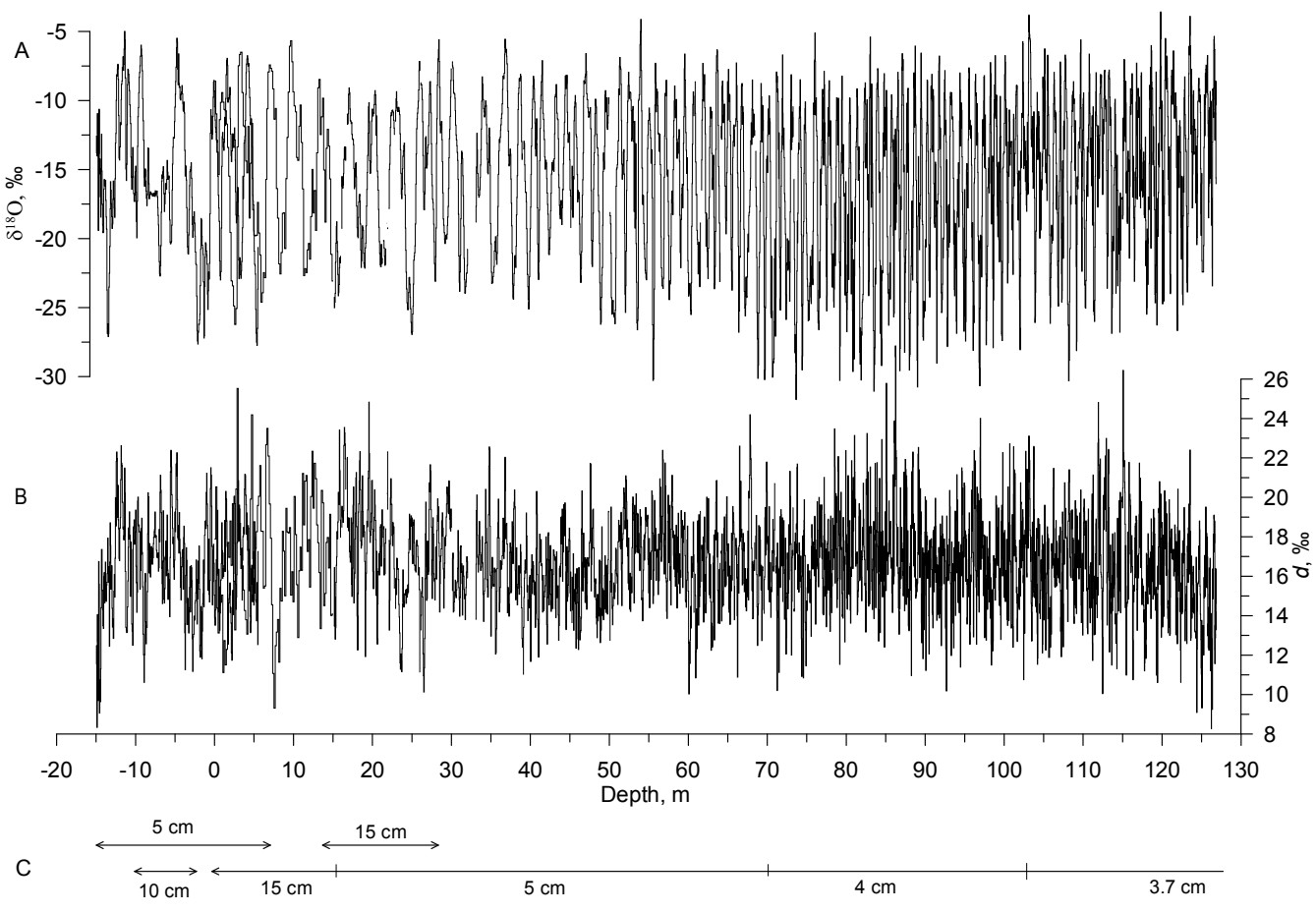

**Fig. 2. Vertical profile of $\delta^{18}O$ (A), deuterium excess (B), and the number of the ice core as well as sampling resolution (C). 0 m**
**depth corresponds to the surface of 2009.**

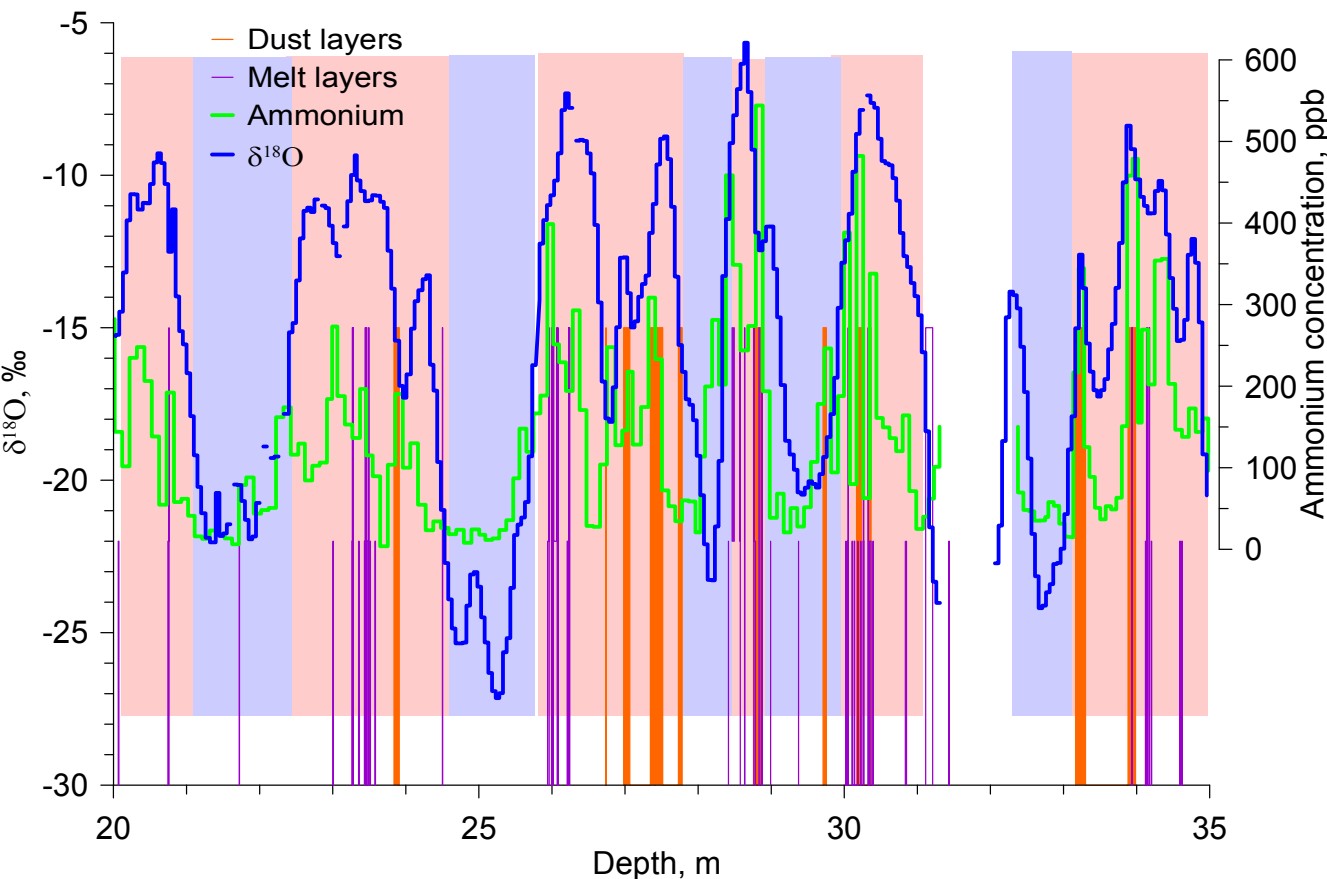

Fig. 3: Illustration of the scheme used to identify warm and cold half-years (respectively indicated by the light red and light blue
shaded areas) based on the deviation of the mean δ18O values from the long-term average value. The purple lines depict the melt
layers observed in the core, dust layers are shown in orange, and the ammonium concentration graph (Mikhalenko et al., 2015) is
in green.

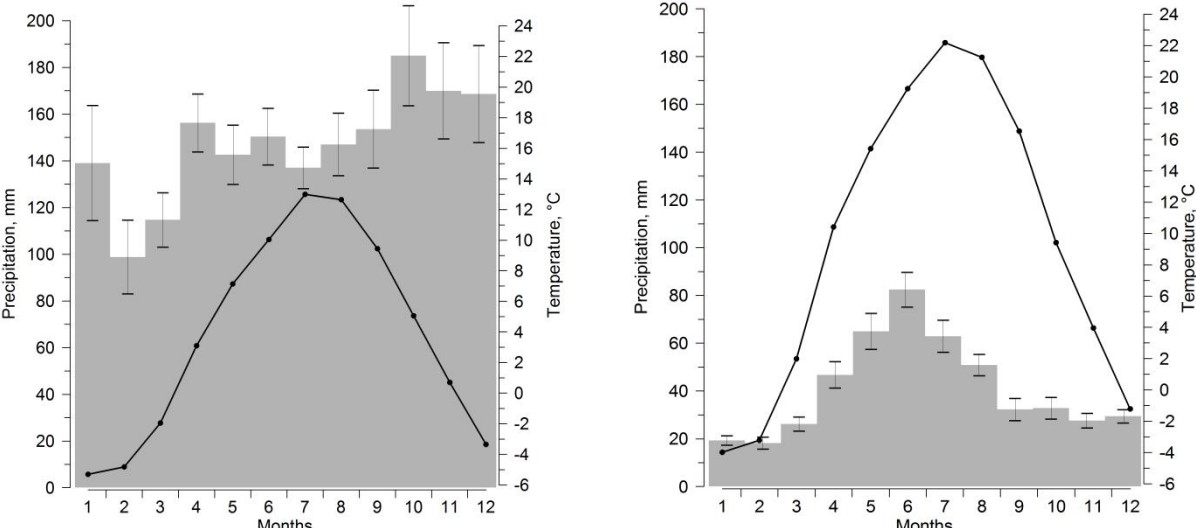

**Fig. 4: Average seasonal cycle of temperature (black dots and line) and precipitation (grey bars) calculated over 1966-1990 period,**
**a) for the Klukhorsky Pereval station (illustrating the lack of a distinct seasonal cycle in precipitation south of the Caucasus) and**
**b) for the Mineralnye Vody station (illustrating the clear seasonal cycle in precipitation seen in stations north of the Caucasus).**
**Error bars (SEM) are shown for the interannual standard deviation of the monthly precipitation rate while the same error bars**
**for the temperature are dimensionless at the scale of the graph.**

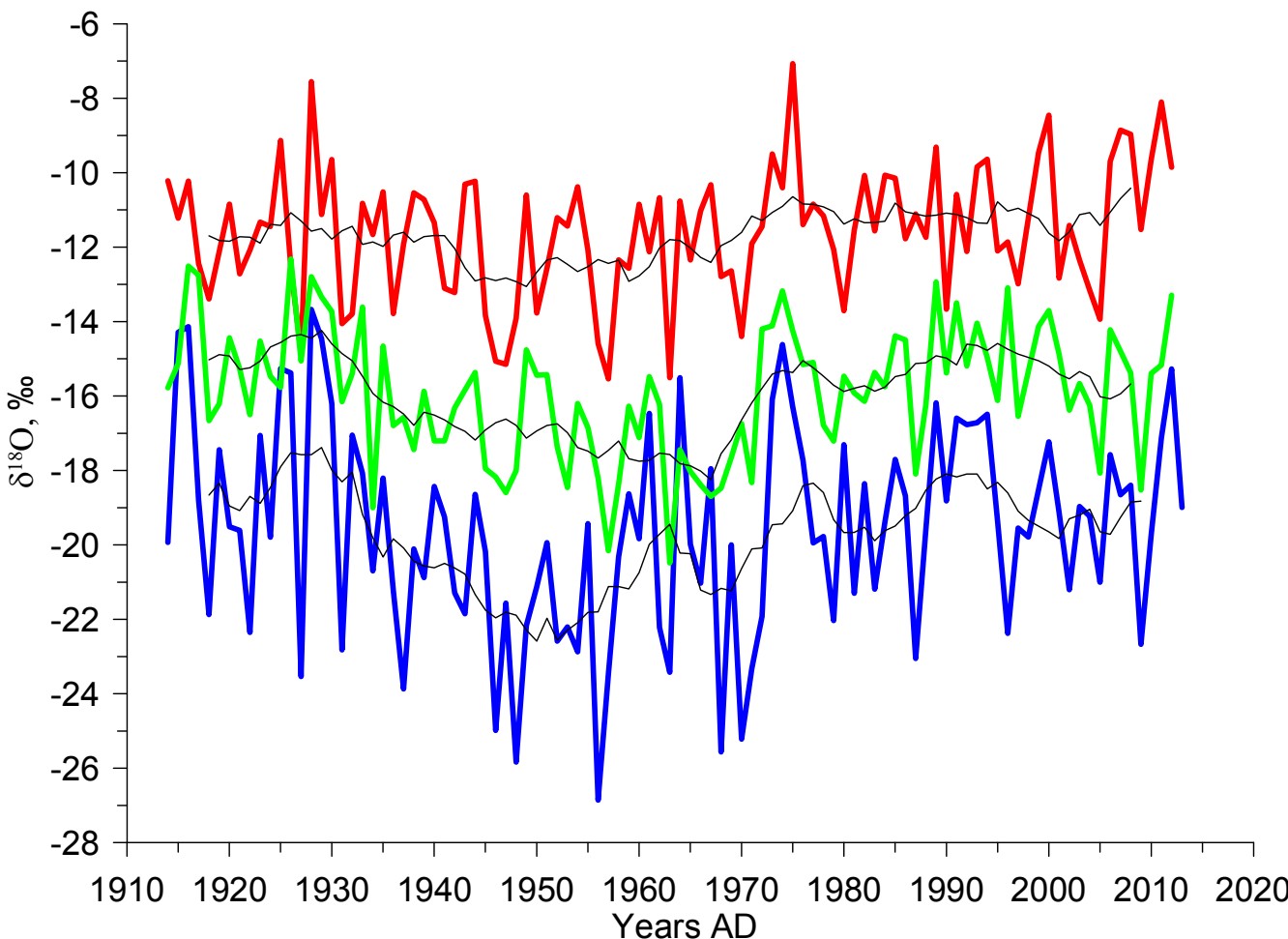

Fig. 5: Annual variations of δ¹⁸O in warm season (red line), in cold season (blue line), and annual means (green line). Thin black
lines show 10-year running means of these parameters.

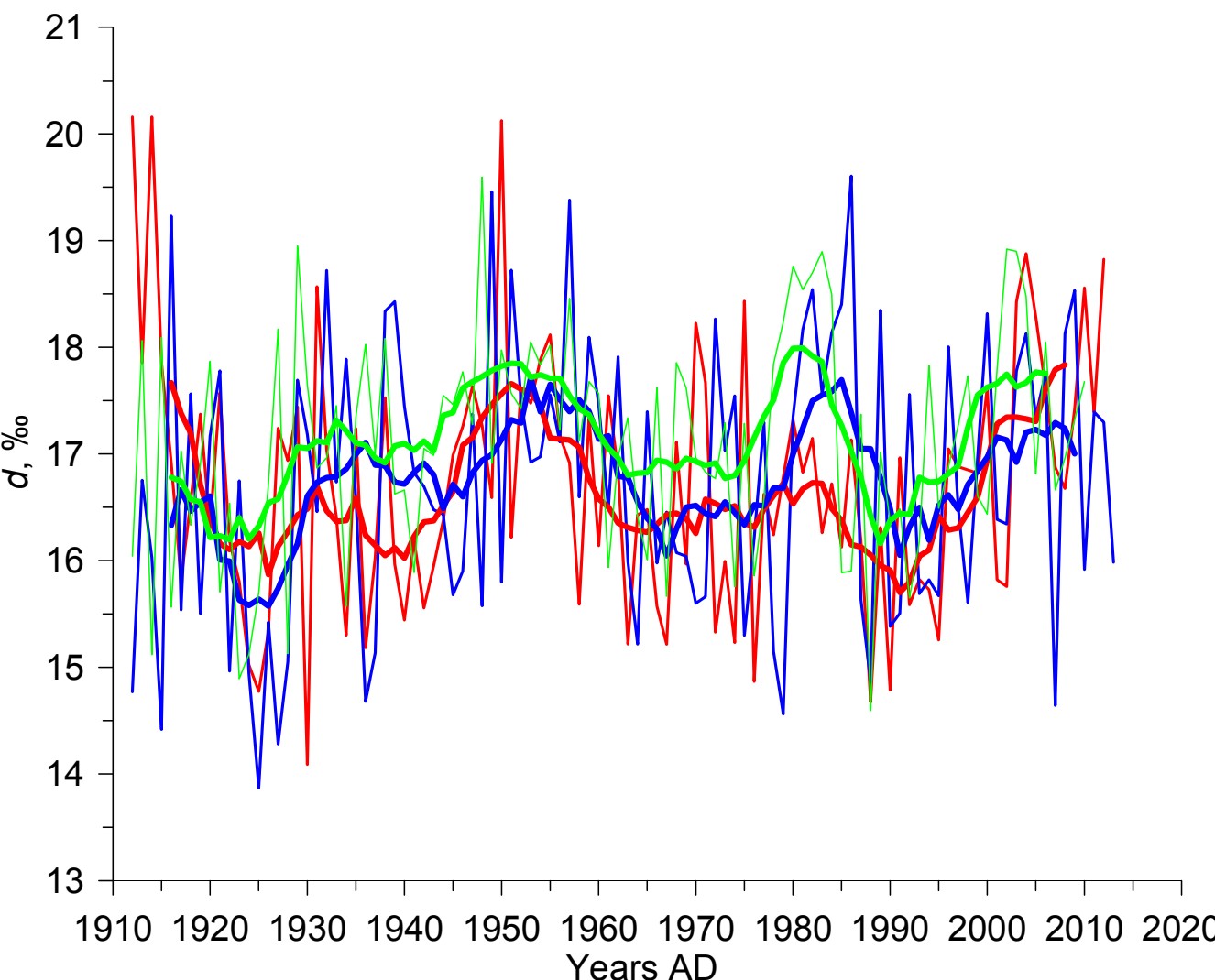

**Fig. 6: Annual variations of deuterium excess in warm season (red line), in cold season (blue line), and mean annual values (green**
**line). Thick lines show the 10-year smoothed values and the thin ones display the raw values.**

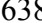

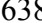

**Fig. 7: Monthly δ¹⁸O (blue line), *d* (green line) and air temperature (red line) data at Bakuriani GNIP station in 2009 (see Table 1
for information on station and Fig. 1 for its location). Note that there is no clear seasonal cycle in deuterium excess, in contrast
with δ¹⁸O showing maximum values in summer and minimum values in winter.**

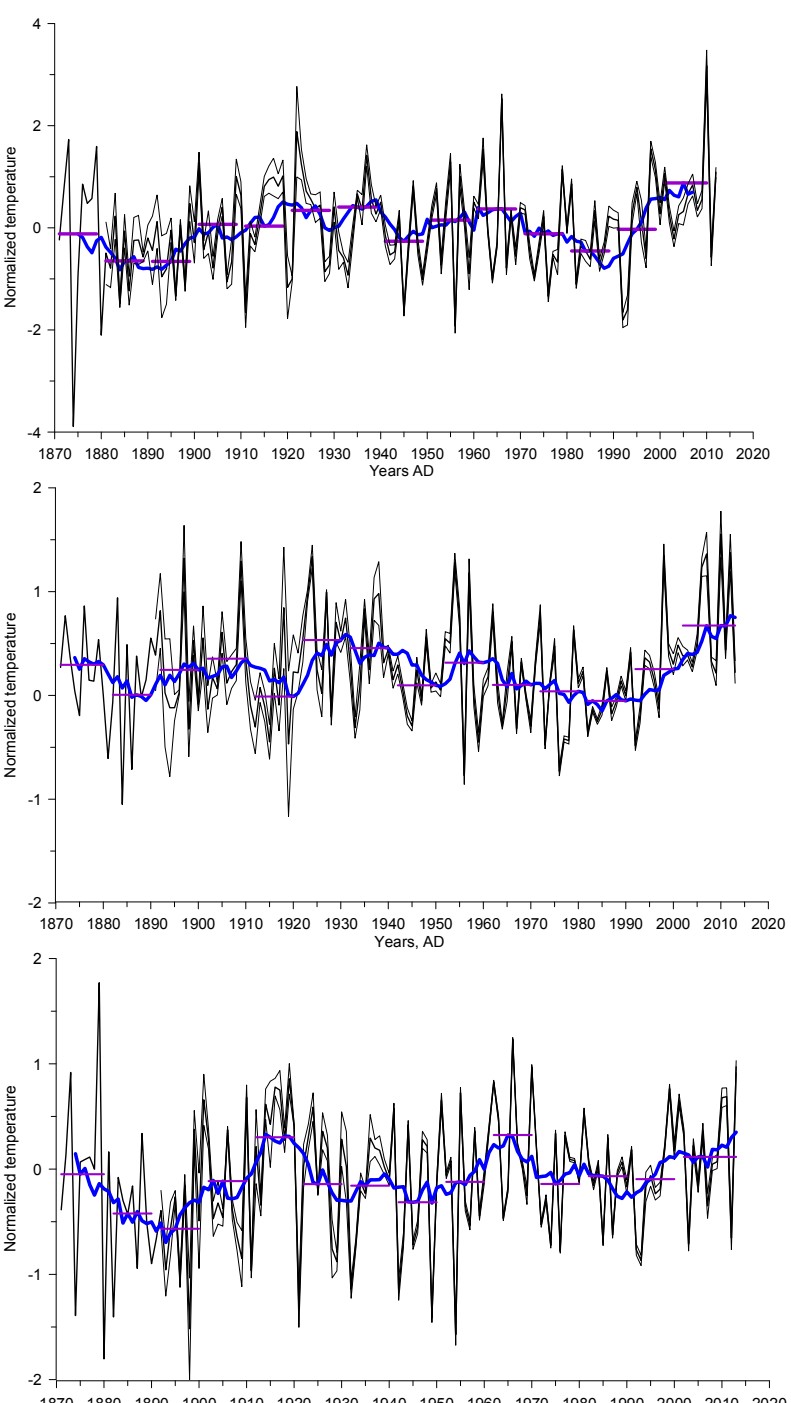



**Fig. 8: Normalized regional temperature record based on meteorological data, with respect to the reference period 1966-1990,**
**expressed as annual anomalies (°C). The thin lines illustrate the standard deviation across the individual records after accounting**
**for the lapse rate from Fig. S3, the blue line shows a 10 year running mean and the horizontal purple line demonstrates the**
**decadal mean value. The upper panel shows the annual means, the middle panel shows the warm season, and the lower panel**
**shows the cold season**

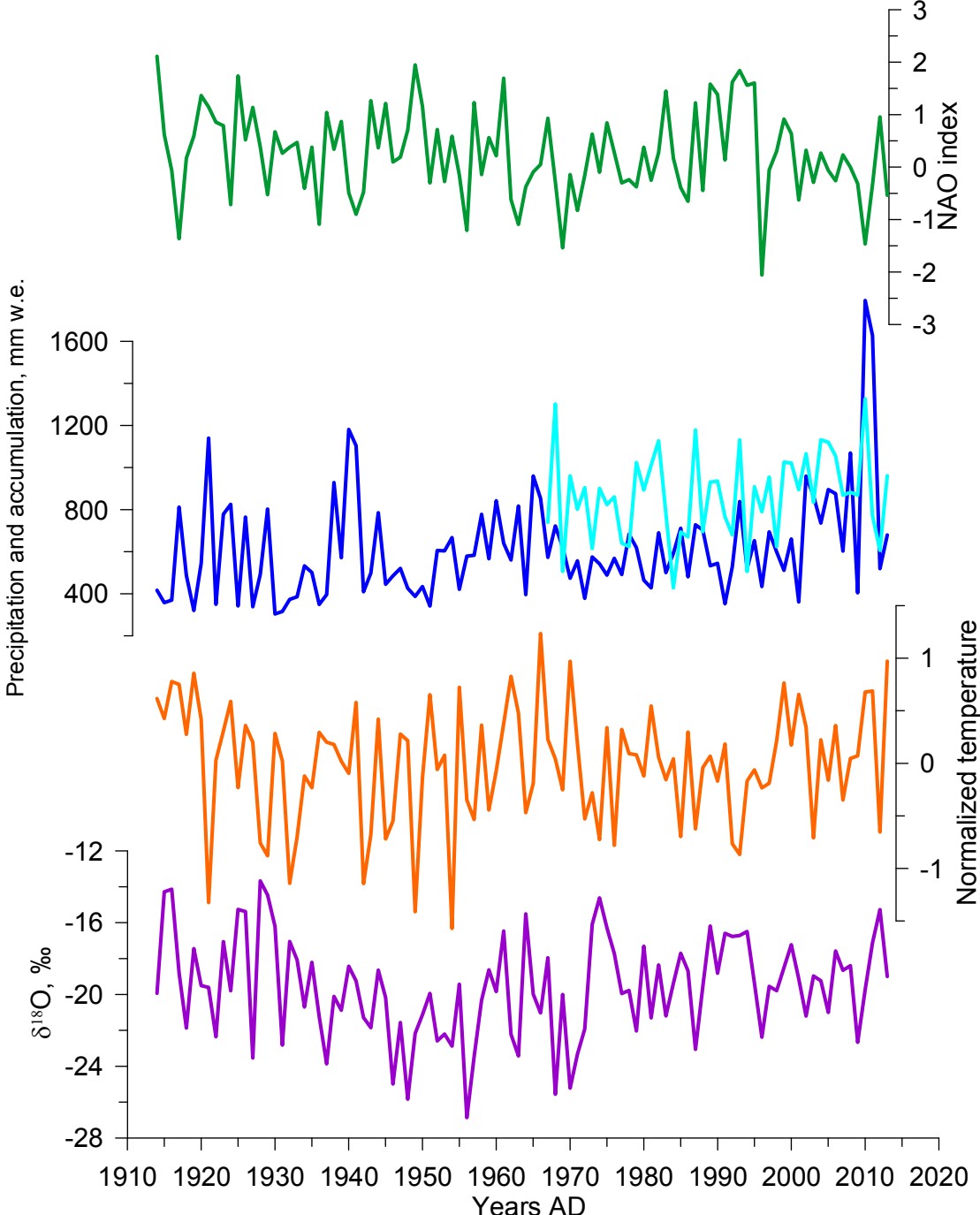

**Fig. 9: Comparison of the ice core record with instrumental regional climate information, for the cold season: δ<sup>18</sup>O composite**
**(purple), temperature at the drilling site calculated from the lapse rate (brown), precipitation at the Klukhorsky Pereval station**
**(light blue) as well as the ice core accumulation estimate (dark blue) and NAO index(green).**

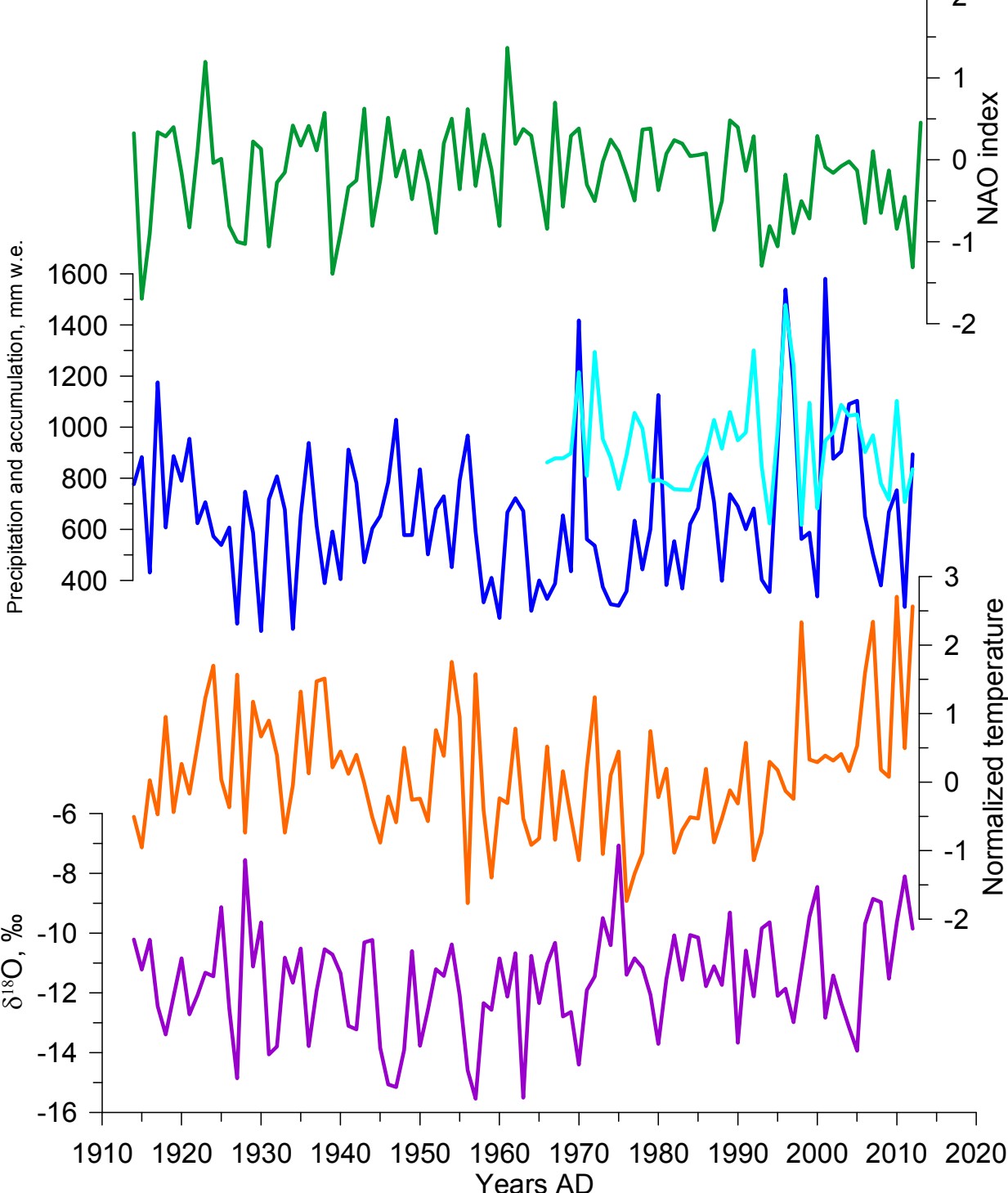

**Fig. 10: Same as fig. 9 but for the warm season.**
**Table 1: Description of meteorological and instrumental data used in the paper**

| Data type | Number on map (Fig. 1) | Location/Name | Altitude a.s.l. | Time span | Data source |
|---|---|---|---|---|---|
| Meteorological observations (temperature, precipitation rate) with daily resolution | 1 | Sochi | 57 m | 1871-present | www.meteo.ru |
| | 2 | Mineralnye Vody | 315 m | 1938-present | |
| | 3 | Kislovodsk | 943 m | 1940-present | |
| | 4 | Pyatigorsk | 538 m | 1891-1997 | |
| | 5 | Shadzhatmaz | 2070 m | 1959-present | |
| | 6 | Terskol | 2133 m | 1951-2005 | |
| | 7 | Klukhorsky Pereval | 2037 m | 1959-present | |
| | 8 | Teberda | 1550 m | 1956-2005 | |
| | 9 | Sukhumi | 75 m | 1904-1988 | |
| | 10 | Samtredia | 24 m | 1936-1992 | |
| | 13 | Tbilisi | 448 m | 1881-1992 | |
| | 14 | Sulak | 2927 m | 1930-present | |
| | 15 | Mestia | 1417 m | 1930-1991 | |
| GNIP data | 11 | Batumi | 32 m | 1980-1990 | http://www-naweb.iaea.org/napc/ih/IHS_resources_gnip.html |
| | 12 | Bakuriani | 1700 m | 2008-2009 | |
| | 13 | Tbilisi | 448 m | 2008-2009 | |
| Circulation indices | n/a | NAO | n/a | 1821-present | Vinter et al., 2009 https:\\crudata.uea.ac.uk\~timo\datapages\naoi.htm |
| | | | n/a | 1950-present | http:\\www.cpc.ncep.noaa.gov\products\precip\CWlink\ |
| | n/a | NCP | n/a | 1948-present | |
| | n/a | AO | n/a | 1950-present | |
| Reanalysis daily temperature | n/a | NCEP | 500 mb level | 1948-present | http://www.esrl.noaa.gov/psd/data/gridded/data.ncep.reanalysis.html Kalnay et al., 1996 |
| Back trajectories | n/a | Flexpart | n/a | 2002-2009 | Forster et al., 2007, Stohl et al., 2009 |
| | n/a | Hysplit | n/a | 1948-present | Draxler, 1999, Stein et al., 2015, Rolph, 2016 |
| | n/a | LMDZiso | n/a | n/a | Risi et al., 2010 |


**Table 2: Correlation coefficients between meteorological data and indices of large-scale modes of variability (statistically**
**significant coefficients at $p < 0.05$ are highlighted in bold). The period of calculation and number of data points (n) for each**
**coefficient are shown in brackets.**

| Annual mean | Temperature | P south* | P north* |
|---|---|---|---|
| | | | |
| NAO | **-0.24** (1914-2013, n=100) | -0.24 (1966-2013, n=48) | -0.03 (1966-2013, n=48) |
| AO | **-0.34** (1950-2013, n=64) | -0.06 (1966-2013, n=48) | 0.02 (1966-2013, n=48) |
| NCP | **-0.55** (1948-2013, n=66) | 0.26 (1966-2013, n=48) | 0.26 (1966-2013, n=48) |
| | | | |
| Warm season | | | |
| NAO | **-0.47** (1914-2013, n=100) | 0.23 (1966-2013, n=48) | 0.03 (1966-2013, n=48) |
| AO | -0.11 (1950-2013, n=64) | 0.08 (1966-2013, n=48) | 0.14 (1966-2013, n=48) |
| NCP | **-0.50** (1948-2013, n=66) | **0.34** (1966-2013, n=48) | **0.34** (1966-2013, n=48) |
| | | | |
| Cold season | | | |
| NAO | **-0.41** (1914-2013, n=100) | 0.04 (1966-2013, n=48) | 0.26 (1966-2013, n=48) |
| AO | **-0.40** (1950-2013, n=64) | 0.14 (1966-2013, n=48) | **0.37** (1966-2013, n=48) |
| NCP | **-0.77** (1948-2013, n=66) | 0.25 (1966-2013, n=48) | **0.33** (1966-2013, n=48) |

*P south – precipitation rate at the weather stations to the South from the Caucasus, P north – precipitation rate at the
weather stations to the North from the Caucasus.

**Table 3: Mean characteristics of the Elbrus ice core records, calculated for the period from 1914 to 2013.**

| Annual means | $\delta^{18}O$, ‰ | $\delta D$, ‰ | $d$, ‰ | Accumulation rate (m w.e./year) |
|---|---|---|---|---|
| Mean | −15.90 | −110.10 | 17.11 | 1,29 |
| Standard deviation | 1.76 | 14.03 | 1.02 | 0.44 |
| **Cold season** | | | | |
| Mean | −19.61 | −140.11 | 16.59 | 0.71 |
| Standard deviation | 2.81 | 22.54 | 2.11 | 0.36 |
| **Warm season** | | | | |
| Mean | −11.58 | −75.97 | 16.69 | 0.65 |
| Standard deviation | 1.75 | 13.98 | 1.14 | 0.27 |


Table 4. Correlation coefficients between ice core data, meteorological data and indices of large-scale modes of variability (statistically significant coefficients at p < 0.05 are highlighted in bold). The period of calculation and number of data points (n) for each coefficient is shown in brackets.

| Annual means | $\delta^{18}$O | Accumulation | d | NAO | AO | NCP |
|---|---|---|---|---|---|---|
| T. °C | −0.01 (1914-2013, n=100) | 0.16 (1914-2013, n=100) | 0.00 (1914-2013, n=100) | **−0.24** (1914-2013, n=100) | **−0.34** (1950-2013, n=64) | **−0.55** (1948-2013, n=66) |
| P north* | **−0.30** (1966-2013, n=48) | **0.36** (1966-2013, n=48) | 0.17 (1966-2013, n=48) | −0.03 (1966-2013, n=48) | −0.03 (1966-2013, n=48) | 0.27 (1966-2013, n=48) |
| P south* | 0.06 (1966-2013, n=48) | **0.52** (1966-2013, n=48) | 0.07 (1966-2013, n=48) | -0.24 (1966-2013, n=48) | −0.06 (1966-2013, n=48) | 0.18 (1966-2013, n=48) |
| $\delta^{18}$O | | | **−0.20** (1914-2013, n=100) | −0.06 (1914-2013, n=100) | 0.07 (1914-2013, n=100) | **0.41** (1950-2013, n=64) | 0.11 (1948-2013, n=66) |
| Accumulation | | | | **0.21** (1914-2013, n=100) | **−0.29** (1914-2013, n=100) | **−0.29** (1950-2013, n=64) | −0.03 (1948-2013, n=66) |
| d | | | | | −0.08 (1914-2013, n=100) | **−0.26** (1950-2013, n=64) | −0.14 (1948-2013, n=66) |
| Warm season | $\delta^{18}$O | Accumulation | d | NAO | AO | NCP |
| T. °C | 0.13 (1914-2013, n=100) | −0.04 (1914-2013, n=100) | **0.20** (1914-2013, n=100) | −0.02 (1914-2013, n=100) | −0.10 (1950-2013, n=64) | **−0.51** (1948-2013, n=66) |
| P north* | 0.01 (1966-2013, n=48) | 0.16 (1966-2013, n=48) | 0.09 (1966-2013, n=48) | 0.13 (1966-2013, n=48) | −0.14 (1966-2013, n=48) | 0.18 (1966-2013, n=48) |
| P south* | −0.27 (1966-2013, n=48) | **0.49** (1966-2013, n=48) | −0.02 (1966-2013, n=48) | −0.01 (1966-2013, n=48) | 0.07 (1966-2013, n=48) | **0.34** (1966-2013, n=48) |
| $\delta^{18}$O | | | **−0.42** (1914-2013, n=100) | −0.05 (1914-2013, n=100) | −0.08 (1914-2013, n=100) | 0.16 (1950-2013, n=64) | 0.00 (1948-2013, n=66) |
| Accumulation | | | | **0.31** (1914-2013, n=100) | 0.00 (1914-2013, n=100) | 0.09 (1950-2013, n=64) | 0.00 (1948-2013, n=66) |
| d | | | | | 0.00 (1914-2013, n=100) | −0.01 (1950-2013, n=64) | −0.14 (1948-2013, n=66) |
| Cold season | $\delta^{18}$O | Accumulation | d | NAO | AO | NCP |
| T. °C | −0.09 (1914-2013, n=100) | 0.11 (1914-2013, n=100) | −0.15 (1914-2013, n=100) | **−0.30** (1914-2013, n=100) | **−0.45** (1950-2013, n=64) | **−0.79** (1948-2013, n=66) |
| P north* | 0.20 (1966-2013, n=48) | 0.21 (1966-2013, n=48) | −0.12 (1966-2013, n=48) | **0.51** (1966-2013, n=48) | **0.37** (1966-2013, n=48) | 0.23 (1966-2013, n=48) |
| P south* | **−0.30** (1966-2013, n=48) | **0.37** (1966-2013, n=48) | −0.13 (1966-2013, n=48) | 0.26 (1966-2013, n=48) | 0.14 (1966-2013, n=48) | 0.25 (1966-2013, n=48) |
| $\delta^{18}$O | | | 0.05 (1914-2013, n=100) | 0.02 (1914-2013, n=100) | **0.41** (1914-2013, n=100) | **0.41** (1950-2013, n=64) | 0.19 (1948-2013, n=66) |
| Accumulation | | | | 0.07 (1914-2013, n=100) | −0.18 (1914-2013, n=100) | −0.15 (1950-2013, n=64) | 0.18 (1948-2013, n=66) |
| d | | | | | −0.06 (1914-2013, n=100) | −0.01 (1950-2013, n=64) | 0.11 (1948-2013, n=66) |

*P south – precipitation rate at the weather stations to the South from the Caucasus, P north – precipitation rate at the weather stations to the North from the Caucasus.