# Peer review of "Large-scale drivers of Caucasus climate variability in meteorological 1 records and Mt Elbrus ice cores 2"

_Climate of the Past, 2016_

## Referee Comment (RC1) · Anonymous Referee #1 · 29 Jul 2016

In this study a 100 year record of water stable isotopes from an ice core collected at Mt. Elbrus in the Caucasus is investigated using meteorological data, reanalysis temperatures and atmospheric circulation indices. The main conclusion is that in summer the isotopic composition is influenced by local temperature whereas in winter atmospheric circulation is the main driver.

Generally the data set presented is very valuable and deserves publication. It is from a region bordering Europe and Asia and although the highest mountain of Europe is located in the Caucasus there are no high-elevation meteorological data available. The presented ice core is also the first from the Caucasus reaching bedrock and collected at an altitude with limited surface melt in summer. I image the effort involved in collecting

this ice core and congratulate the authors for this achievement. The stable isotope records show impressively clear seasonal variations, allowing for dating by annual layer counting. However, the annual d18O values did not reveal a relation with regional temperature. The authors attempt to find an explanation for that, but unfortunately this is not convincing.

The most critical point is separating the record into a warm a cold season part. This is conducted by implementing a threshold (average d18O value of -15.5‰ for the entire record), thereby inherently presuming a d18O-temperature relationship and the absence of a trend. This introduces a circular argument when examining the temperature dependence of the resulting warm and cold season record. Nevertheless, no significant correlation with local temperature was found for the entire time period. Only when reducing the data set to the period 1984 to 2013 a significant correlation (r=0.52) was observed, implying that in this period local temperature explains 27% of the d18O variance. This is a weak relationship and not sufficient to draw the conclusion that the summer season isotopic composition depends on local temperature. Similar for accumulation: The highest correlation coefficient r of 0.44 was observed between warm or cold season accumulations and precipitation in the south for the period 1966-2013. Again precipitation explains only less than 20% of the accumulation variability, insufficient to reconstruct precipitation back to 1914 as was proposed.

The missing relation with local temperature is easily explained by the fact that the inter-annual variability of temperature is low, whereas the mean seasonal amplitude of d18O is high (20‰. A slight seasonal shift in precipitation or accumulation from year-to-year therefore affects the mean d18O significantly. There is no need to evoke post-depositional processes (which of course add further biases) or other than local controls. Without clear seasonal marker in the ice core this limitation cannot be overcome. This situation is particularly unfavourable at sites with no distinct cycle of precipitation as assumed for the Elbrus ice core. In cases with a strong seasonal cycle, this normally persists even if slight shifts occur from year-to-year. Examples are the Colle Gnifetti in

the Alps, where the saddle location induces an accumulation with strong seasonality biased towards summer, due to snow erosion in winter. There, for decadal scales, the isotope variability correlated with the temperature record at around r=0.65 (Bohleber et al., 2013). A similar situation was shown for Belukha glacier in the Altai, a region where summer precipitation dominates. 10-year averages of d18O and MAR–NOV temperatures were significantly correlated (r=0.83, p < 0.001) (Eichler et al., 2009). In both cases there were pronounced trends in temperatures and d18O. This is obviously absent in the Elbrus region over the last 100 years. So, the question is if you could investigate a longer time period (potentially showing a trend in temperature) and longer-term averages to smooth the effect of year-to-year shifts in precipitation/accumulation.

One other point is the dating uncertainty and how you deal with that for correlation analysis with meteorological data. You specify +/- 1 year uncertainty, whereas the first publication on this ice core (Mikhalenko et al., 2015) shows a 2-years difference be-tween annual layer counting of the stable isotope signal and the chemical stratigraphy at 106.7 m. What is correct and how to you consider this in the correlation analysis?

Obviously this is not the first publication about that ice core which is not a problem if you present other data or new analyses. Here this is not so clear and you should state it and reference it where results were already presented before. Examples are the diffusion of stable isotopes, the AWS data from the ice core site, the overlap with the shallow cores, the precision of the stable isotope analysis (0.06‰ for d18O here and 0.07‰ in Mikhalenko et al. (2015)).

I wonder how the entire stable isotope record looks like. In the manuscript only the part down to 126 m out of 182 m is shown, whereas it is stated that the entire core was analysed. Why do you not focus on a longer period, for example back to 1815, since the Tambora volcanic layer gives a nice time marker, detected in most of the ice core records.

In the introduction you state that water stable isotopes are more sensitive to distortion

because of seasonality than aerosol concentrations, which is not correct. The seasonality of aerosol-related species and the isotope signal are comparable, but the anthropogenic aerosol trend exceeds by far any temperature-driven water isotope increase during the Holocene (Wagenbach et al., 2012).

Explain why there are gaps in the data (fig. 2 and 3) and how you treated them for calculating annual averages.

Table 4: Include number of points n or time period for correlation analysis when they are different for the different parameters as for temperature and precipitation.

Bohleber, P., Wagenbach, D., Schoner, W., and Bohm, R.: To what extent do water isotope records from low accumulation Alpine ice cores reproduce instrumental temperature series?, Tellus Series B-Chemical and Physical Meteorology, 65, 17, 2013.

Eichler, A., Olivier, S., Henderson, K., Laube, A., Beer, J., Gaeggeler, H. W., Papina, T., and Schwikowski, M.: Temperature Changes in the Altai Are Driven by Solar and Anthropogenic Forcing, Chimia, 63, 889-889, 2009.

Mikhalenko, V., Sokratov, S., Kutuzov, S., Ginot, P., Legrand, M., Preunkert, S., Lavrentiev, I., Kozachek, A., Ekaykin, A., Faïn, X., Lim, S., Schotterer, U., Lipenkov, V., and Toropov, P.: Investigation of a deep ice core from the Elbrus western plateau, the Caucasus, Russia, The Cryosphere, 9, 2253-2270, 2015.

Wagenbach, D., Bohleber, P., and Preunkert, S.: COLD, ALPINE ICE BODIES REVISITED: WHAT MAY WE LEARN FROM THEIR IMPURITY AND ISOTOPE CONTENT?, Geogr. Ann. Ser. A-Phys. Geogr., 94A, 245-263, 2012.
* * *

---

## Referee Comment (RC2) · Anonymous Referee #2 · 27 Sep 2016

This study is an investigation of the isotopic composition of an alpine ice core from Mt. Elbrus, Caucassus. Kozachek et al. compares the seasonally divided ice core data to meteorological observations and finds the summer d18O to be related to temperature while the winter d18O to related to the main modes of atmospheric circulation.

While the authors presents some interesting analysis and the data is publication-worthy I find that there are some major drawbacks on the writing and the treatment of the data. Like Referee #1 I am concerned with the division of the data into seasons which is simply done by dividing the data around the long term mean value. At times I also find the writing to be imprecise, and a lack of citations of relevant previous work on related subjects (see detailed comments below).

Major comments.

Seasonal d18O data. The division of the data as shown in Fig 3 is largely unmotivated, except that there appears to be an annual cycle in the data. How is the distribution of seasonal accumulation? We don't know and it seems the authors have not investigated this. I suggest that a similar approach as Vinther et al. (2010) is made. I.e. investigating the proportion of the yearly accumulation to be assigned to either summer or winter depending on the coherency with meteorological observations, be it either temperature or circulation indecies. I think that before a properly motivated division of the seasons is made the effort of discussing the outcome of the analysis is not really relevant.

Detailed comments.

In the introduction in general I miss a stronger representation of similar work done for Greenland although Greenland is mentioned. Many of the research questions are similar as well as the connection to atmospheric circulation patterns. See e.g. Vinther et al. 2003, Vinther et al. 2010 and Ortega et al. 2014.

L57-63 here a lot of detailed processes are mention, but there are no reference to literature. Why not refer to the early isotope work by Willi Dansgaard and e.g Persson et al. 2011 on intermittency of snowfall.

L169-176 I can't follow this section easily. I suppose the point you want to make is that you think diffusion has little influence on the isotope values. Did you calculate the variation of amplitude of the d18O annual cycle from top to bottom? It might "look" like there is no decease in amplitude, but what are the numbers? Another way to test if diffusion plays a role is the d-excess. Since the diffusivity of HDO and H2-18O is different there will be a phase change of d-exess with the diffusion often shifting the d-excess peak earlier in the year (depending on the annual cycle of the d-excess).

References.

Persson A, Langen PL, Ditlevsen P, Vinther BM (2011) The influence of precipitation

weighting on interannual variability of stable water isotopes in Greenland. J Geophys Res 116:D20120. doi:10. 1029/2010JD015517

Ortega, P., Swingedouw, D., Masson-Delmotte, V. et al. Clim Dyn (2014) 43: 2585. doi:10.1007/s00382-014-2074-z

Vinther, B., S. Johnsen, K. Andersen, H. Clausen, and A. Hansen (2003), NAO signal recorded in the stable isotopes of Greenland ice cores, Geophys. Res. Lett., 30(7), 1387, doi:10.1029/2002GL016193.

Vinther, B., P. Jones, K. Briffa, H. Clausen, K. Andersen, D. Dahl‐-Jensen, and S. Johnsen (2010), Climatic signals in multiple highly resolved stable isotope records from Greenland, Quat. Sci. Rev., 29(3‐4), 522–538, doi:10.1016/j.quascirev.2009.11.002.
* * *

---

## Referee Comment (RC3) · Anonymous Referee #3 · 3 Oct 2016

Study on the drivers of climate variability in a region is very important for understanding climate change and its prediction. Based on the meteorological data and ice core records, this paper discussed the impacts of NAO, AO and NCP on climate change in the Caucasus, and found that in the summer season the isotopic composition in the Elbrus ice core depends on the local temperature, while in winter, the atmospheric circulation is the predominant driver of the ice core isotopic composition, and the ice core isotopic composition appears mostly related to characteristics of large-scale atmospheric circulations such as the NAO. However, there are some issues in the paper which should be clarified. 1. If possible, it would be better to draw a dividing line in Fig.1 to separate the regions with and without a distinct seasonal variation of precipitation. This can help readers to understand some discussions in the paper. 2. The dating is very important for the ice core study. In the section of dating, i.e. 2.1.4, authors used the mean value of the $\delta18O$ of the whole dataset (-15.5 ‰ as a threshold to separate between the warm and cold seasons. This suggestion should be verified and/or confirmed by the data of $\delta18O$ in precipitation at the GNIP stations around the ice core drilling site. Another way to test the effectiveness of the division of seasons in ice core is to discern if there is a consistency between the ratio of warm season accumulation rate to cold season accumulation rate (in table 3) and that of precipitation at the adjacent meteorological stations (this method was used by Wang et al (2002, Annals of Glaciology, Vol.35, 273-277) in a Himalayan ice core). Authors also mentioned that the other parameters with seasonal variational characteristics, such as dust and ammonium concentrations, were used to identify the warm/cold season in the ice core profile. It would be better to display the variations of these parameters in the Fig. 3. 3. Authors calculated the correlation between temperature and $\delta18O$ in the Lines 329-332 of the text using the 11-year running means for the different periods, and found that the correlations changes with time. If possible, authors can do this by a sliding window method used by Wang et al. (2003, Geophysical Research Letters. Vol.30, No.22, doi: 10.1029/2003GL018188) in a Tibetan ice core. Another issue is that the data series used in the paper ended in 2013, why their 11-year running means also ended in 2013 (shown in Fig. 11)? 4. The significance test in the paper should be paid much attention, especially for the datasets of 11-year and 20-year running means. The degree of freedom can be reduced sharply for the running mean datasets. For example, as for the 11-year mean data sets over the period of 1994-2013, their degree of freedom is only 2 (20/11 is about 2). 5. In the paragraph, Lines 343-346, authors should present the results of the seasonal cycle of precipitation isotopic composition calculated by using the LMDZiso model, and compare that with the ice core record in one chart. 6. When discussing the variations of $\delta18O$ in precipitation in lines 362-365, the continental effect should be considered. 7. In Tables 2 and 4, the period of calculation should be presented. 8. Line 321, "in the Alps by (Bohleber et al., 2013)" should

be "in the Alps by Bohleber et al. (2013)". 9. Line 327, "the methods described by (Bohleber et al., 2013)" should be "the methods described by Bohleber et al. (2013)".

---

## Editor Comment (EC1) · S. Hou (Editor) · 1 Nov 2016

Dear Authors, I would like invite you to submit a revised version taking into account the detailed comments of the referees. Best regards, Hou

---

## Author Response (AR1)

We would like to thank three anonymous reviewers for their comments. The answers to the questions
raised in the reviews provided below. The comments of reviewers are highlighted in *italic* and the
corrections in the paper are in **bold**. Figures that are used for the answers only are inserted into the text;
updated figures from the paper and the supplement are at the end of the document.
**Reviewer 1**
*The most critical point is separating the record into a warm a cold season part. This is conducted by*
*implementing a threshold (average d18O value of -15.5‰ for the entire record), thereby inherently*
*presuming a d18O-temperature relationship and the absence of a trend.*
We agree that the dating section should be revised to make the dating procedure clearer as was pointed
by all the three reviewers. However, we think that the proposed method of dating when the border
between warm and cold seasons is the 100-years mean value is the best one for this very ice core.
Accumulation at the drilling site has been investigated sporadically (see review in Mikhalenko et al.,
2015). We cannot use the meteorological observations from the nearest weather stations as these
stations situated at sufficiently lower elevation and belong to two different groups as discussed in
section 3.1. The ice core is the only source for the information about the seasonal cycle of this
parameter.
We think that the annual cycle of the isotopic composition is influenced by local temperature while
interannual variations depend on the other factors. In order to better illustrate the dating methodology
we will add the ammonium concentration and dust concentration profiles to Fig. 3. Layers with the high
dust concentration have been precisely dated by Kutuzov et al. (2013) for the 2012 ice core. Their
results show that the separation of the core into a warm and cold season part using the average value of
$\delta18O$ is appropriate for this drilling site at least for the period from 2009 till 2012 that was investigated
in the paper. Also, to show correlation between temperature and isotopic composition on annual scale,
we will add temperature data to the GNIP data graphs on Fig. 7.
As for the linear trend that can cause errors in the dating, we also tried separation into warm and cold
seasons using linear trend of $\delta18O$. The result is shown on Fig. below. The difference between this
method of separation and ours is about 0.5 per mil which is comparable with the $\delta18O$ measurement
precision and is negligible given the high accumulation rate at the drilling site.

[Figure]

Fig. Vertical profile of δ18O with the linear trend.

**2.1.4 Dating**

**The chronology is based on the identification of annual layers. These are prominent in $\delta^{18}O$ with the average seasonal amplitude of 20 ‰. As there is no trend in the $\delta^{18}O$ record. we used the mean value of the $\delta^{18}O$ of the whole dataset (-15.5 ‰) as a threshold to separate between the warm and cold seasons. For equivocal situations, we also used additional data: melt layers and dust layers (used to identify the warm season) (Kutuzov et al., 2013) as well as ammonium and succinic acid concentration data that also have seasonal variations (Mikhalenko et al., 2015). Layers with the high dust concentration have been precisely dated by Kutuzov et al. (2013) for the 2012 ice core. Their results show that the separation of the core into a warm and cold season part using the average value of $\delta^{18}O$ is appropriate for this drilling site at least for the period from 2009 till 2012 that was investigated by Kutuzov et al. (2013).**

*So, the question is if you could investigate a longer time period (potentially showing a trend in temperature) and longer term averages to smooth the effect of year-to-year shifts in precipitation/accumulation.*

Investigation of a sufficiently longer period is not feasible because of huge dating uncertainties. We will discuss them elsewhere.
Also, we tried 3-, 5-, and 7-years running means for the correlation analysis but obtained the same result. We can add these results as well.

**At the bottom part of the core the isotopic composition cycles are less prominent and cannot be used for dating, consequently the dating uncertainty is sufficiently higher. The isotopic composition of that part of the core will be discussed elsewhere.**

**We also repeated our linear correlation analysis using precipitation weighted temperature, and obtained the same results. We didn't find any statistically significant correlations when compared 3-, 5-, 7-years running means of these parameters.**

*One other point is the dating uncertainty and how you deal with that for correlation analysis with meteorological data. You specify +/- 1 year uncertainty, whereas the first publication on this ice core (Mikhalenko et al., 2015) shows a 2-years difference between annual layer counting of the stable isotope signal and the chemical stratigraphy at 106.7 m. What is correct and how to you consider this in the correlation analysis?*

We agree with the comment and will correct the uncertainty as stated by Mikhalenko et al (2015). In the correlation analysis we used the dating obtained using the isotopic composition annual cycles counting. We will add this point to the text of the paper.

**The discrepancy between two independent chronologies is 2 years at a depth of 126 m. We used the dating based on the isotopic composition data in this paper. This dating is also best fit for the correlation analysis with the meteorological data.**

*Obviously this is not the first publication about that ice core which is not a problem if you present other data or new analyses. Here this is not so clear and you should state it and reference it where results were already presented before. Examples are the diffusion of stable isotopes, the AWS data from the ice core site, the overlap with the shallow cores, the precision of the stable isotope analysis (0.06‰ for d18O here and 0.07‰ in Mikhalenko et al. (2015)).*

The paper of Mikhalenko et al (2015) presented the ice core and the analysis done. Now we are discussing the isotopic profile of the core. Of course, some replications are inevitable in this case. We will add this point to the discussion section as well as citations of Mikhalenko et al (2015) to the data and methods section. The analytical precision is slightly different as now we are discussing a bigger part of the core.

**The methods of the isotopic measurements have been partially discussed in (Mikhalenko et al., 2015).**

**Our calculation showed that the seasonal amplitude of $\delta^{18}O$ variations could be 10-20% less because of the diffusion (Mikhalenko et al., 2015).**

**At our drilling site, an automatic weather station (AWS) provided in situ measurements for the period from August 2007 till January 2008. The day to day variations of temperature at low elevation weather stations and at the AWS are coherent for the whole period of the AWS work (Mikhalenko et al., 2015).**

*I wonder how the entire stable isotope record looks like. In the manuscript only the part down to 126 m out of 182 m is shown, whereas it is stated that the entire core was analysed. Why do you not focus on a longer period, for example back to 1815, since the Tambora volcanic layer gives a nice time marker, detected in most of the ice core records.*

We are not going to discuss the bottom part of the core as the isotopic cycle is less prominent there and cannot be used for the dating purpose. The dating using the volcanic layers at Elbrus is complicated as Elbrus is a volcano itself. The dating of the bottom part of the core and the properties of this part like isotopic composition, chemical composition, and dust concentration will be discussed elsewhere. We focused on 100 years period because it is covered by weather observations in the region and we can obtain ice core data with annual resolution.

**Hereafter, we focus our analysis on one century, from 1914 till 2013, which corresponds to the upper 126 m of the core. This period has been chosen because of relatively small dating uncertainty and the availability of other records such as local meteorological observations. At the bottom part of the core the isotopic composition cycles are less prominent and cannot be used for dating, consequently the dating uncertainty is sufficiently higher. The isotopic composition of that part of the core will be discussed elsewhere.**

*In the introduction you state that water stable isotopes are more sensitive to distortion because of seasonality than aerosol concentrations, which is not correct. The seasonality of aerosol-related species and the isotope signal are comparable, but the anthropogenic aerosol trend exceeds by far any temperature-driven water isotope increase during the Holocene (Wagenbach et al., 2012).*

We removed this statement as it is unimportant for the further discussion.

*Explain why there are gaps in the data (fig. 2 and 3) and how you treated them for calculating annual*
*averages.*

The gaps came from the technical problems during the drilling operations and the analysis process. The
drilling problems are thoroughly described in (Mikhalenko et al., 2015). We used the values from the
duplicate core obtained in 2004 for the gap between 31.3 and 32.1 m. In case of one sample missing we
considered its isotopic value to be the average between the two neighbor samples. We will add this
explanation to the paper.

**There some gaps in the isotopic composition data that came from the technical problems during**
**the drilling operations and the analysis process. The drilling problems are described in**
**(Mikhalenko et al., 2015). We used the values from the duplicate core obtained in 2004 for the gap**
**between 31.3 and 32.1 m. In case of one sample missing we considered its isotopic value to be the**
**average between the two neighbor samples.**

*Table 4: Include number of points n or time period for correlation analysis when they are different for*
*the different parameters as for temperature and precipitation.*

Ok, we will do this

**Table 2: Correlation coefficients between meteorological data and indices of large-scale modes of**
**variability (statistically significant coefficients at p < 0.05 are highlighted in bold). The period of**
**calculation for each coefficient is shown in brackets.**

| | SUMMER | | | WINTER | | |
|---|---|---|---|---|---|---|
| | **Temperature** | **P south*** | **P north*** | **Temperature** | **P south*** | **P north*** |
| **NAO** | **-0.47 (100)** | **0.23 (45)** | **-0.03 (45)** | **-0.41 (100)** | **0.04 (45)** | **0.26 (45)** |
| **AO** | **-0.11 (63)** | **0.08 (45)** | **-0.14 (45)** | **-0.40 (63)** | **0.14 (45)** | **0.37 (45)** |
| **AMO** | **0.24 (100)** | **0.01 (45)** | **-0.02 (45)** | **0.07 (100)** | **0.27 (45)** | **0.25 (45)** |
| **NCP** | **-0.50 (65)** | **0.34 (45)** | **0.18 (45)** | **-0.77 (65)** | **0.25 (45)** | **0.33 (45)** |

**\*P south – precipitation rate at the weather stations to the South from the Caucasus, P north –**
**precipitation rate at the weather stations to the North from the Caucasus.**

**Table 4. Correlation coefficients between ice core data, meteorological data and indices of large-**
**scale modes of variability (statistically significant coefficients at p < 0.05 are highlighted in bold).**
**The period of calculation for each coefficient is shown in brackets.**

| Summer | $\delta^{18}O$ | Accumulation | $d$ | NAO | AO | NCP |
|---|---|---|---|---|---|---|
| $T.$ °C | 0.13 (100) | 0.09 (100) | 0.21 (100) | –0.48 (100) | –0.10 (63) | –0.51 (65) |
| P north | 0.07 (45) | 0.24 (45) | 0.11 (45) | –0.03 (45) | –0.14 (45) | 0.18 (45) |
| P south | –0.12 (45) | 0.44 (45) | –0.04 (45) | 0.23 (45) | 0.08 (45) | 0.34 (45) |
| $\delta^{18}O$ | | –0.17 (100) | –0.11 (100) | 0.06 (100) | 0.23 (63) | –0.04 (65) |
| Accumulation | | | 0.27 (100) | –0.25 (100) | 0.05 (63) | 0.07 (65) |
| $d$ | | | | –0.17 (100) | 0.00 (63) | –0.18 (65) |
| Winter | $\delta^{18}O$ | Accumulation | $d$ | NAO | AO | NCP |
| $T.$ °C | –0.02 (100) | 0.31 (100) | –0.08 (100) | –0.42 (100) | –0.45 (63) | –0.79 (65) |
| P north | 0.25 (45) | 0.13 (45) | –0.01 (45) | 0.26 (45) | 0.37 (45) | 0.23 (45) |
| P south | –0.09 (45) | 0.44 (45) | –0.06 (45) | 0.04 (45) | 0.14 (45) | 0.25 (45) |
| $\delta^{18}O$ | | –0.05 (100) | –0.04 (100) | 0.42 (100) | 0.34 (63) | 0.08 (65) |
| Accumulation | | | 0.04 (100) | –0.34 (100) | –0.35 (63) | 0.05 (65) |
| $d$ | | | | 0.05 (100) | –0.09 (63) | 0.04 (65) |

**\*P south – precipitation rate at the weather stations to the South from the Caucasus, P north – precipitation rate at the weather stations to the North from the Caucasus.**

**Reviewer 2**

*Seasonal d18O data. The division of the data as shown in Fig 3 is largely unmotivated, except that there appears to be an annual cycle in the data. How is the distribution of seasonal accumulation? We don't know and it seems the authors have not investigated this. I suggest that a similar approach as Vinther et al. (2010) is made. I.e. investigating the proportion of the yearly accumulation to be assigned to either summer or winter depending on the coherency with meteorological observations, be it either temperature or circulation indecies. I think that before a properly motivated division of the seasons is made the effort of discussing the outcome of the analysis is not really relevant.*

We will broaden the dating section as pointed before. See the answer to the reviewer 1.

*In the introduction in general I miss a stronger representation of similar work done for Greenland although Greenland is mentioned. Many of the research questions are similar as well as the connection to atmospheric circulation patterns. See e.g. Vinther et al. 2003, Vinther et al. 2010 and Ortega et al. 2014.*

We will add this to the introduction.

**Connection of Greenland ice cores isotopic composition with the atmospheric circulation patterns was studied by Vinther et al. (2003 and 2010). The strong influence of the NAO pattern on the Greenland ice cores isotopic composition has been discovered and the possibility to use the ice cores data for the past NAO changes reconstruction was proved (Vinther et al., 2003). The authors also revealed the importance of the seasonally resolved ice cores records study rather than annual records as there are different factors governing formation of the isotopic composition of precipitation in warm and in cold seasons (Vinther et al., 2010).**

*L57-63 here a lot of detailed processes are mention, but there are no reference to literature. Why not refer to the early isotope work by Willi Dansgaard and e.g Persson et al. 2011 on intermittency of snowfall.*

We will add these references to the paper.

**Water stable isotope records are in mid to high latitudes physically related to condensation temperature through distillation processes (Dansgaard, 1964), but the climate signal is archived through the snowfall deposition and post-deposition processes. One important artefact lies in the intermittency of precipitation, and the covariance between condensation temperature and precipitation, which may bias the climate record towards one season, or towards one particular weather regime, challenging an interpretation in terms of annual mean temperature (Persson et al., 2011).**

*L169-176 I can't follow this section easily. I suppose the point you want to make is that you think diffusion has little influence on the isotope values. Did you calculate the variation of amplitude of the d18O annual cycle from top to bottom? It might "look" like there is no decease in amplitude, but what are the numbers? Another way to test if diffusion plays a role is the d-excess. Since the diffusivity of HDO and H2-18O is different there will be a phase change of d-excess with the diffusion often shifting the d-excess peak earlier in the year (depending on the annual cycle of the d-excess).*

Yes, exactly, we think that the diffusion of stable isotopes does not influence the isotopic profile
significantly at the part of the core that is discussed in the paper. We will add description of the
calculation procedure to the section. Investigation of d-excess in this case will not add any information
as the seasonal cycle of this parameter is not observed (Fig. 2).

**2.1.5 Diffusion of stable isotopes**

**We calculated the potential influense of diffusion on the stable isotopes record according to**
**(Johnsen, 2000) model. We used the following parameters for the calculation: Our calculation**
**showed that the seasonal amplitude of $\delta^{18}$O variations could be 10-20% less because of the**
**diffusion (Mikhalenko et al., 2015). If it was the case we would observe a decreasing of $\delta^{18}$O**
**maxima and increasing of minima with depth. Moreover we would find a positive correlation**
**between accumulation rate and seasonal amplitude of $\delta^{18}$O. These features have not been found in**
**the ice core data. The correlation coefficient between seasonal amplitude and accumulation rate is**
**-0.10 and is statistically insignificant. There is also no statistically significant trend in the seasonal**
**amplitude; the seasonal amplitude varies stochastically from 10 to 25 ‰. The maximum value**
**observed on 1984 and the minimum in 1925. We therefore consider that the diffusion does not**
**influence sufficiently the isotopic composition record in the upper 126 m of the ice core. At the**
**bottom part of the core (e.g. at a depth of 180 m) the annual cycle of $\delta^{18}$O should have an**
**amplitude of 4 ‰ which is detectable but the length of the cycle should be less then 1 cm. As the *d***
**annual cycle is not prominent we cannot used the method based on the discrepancy between the**
**$\delta^{18}$O and *d* cycles. Thus, for obtaining climatic information from the bottom part of the core very**
**high sampling resolution is required.**

**Reviewer 3**
*1. If possible, it would be better to draw a dividing line in Fig.1 to separate the regions with and*
*without a distinct seasonal variation of precipitation. This can help readers to understand some*
*discussions in the paper.*

Ok, we will add it

*2. The dating is very important for the ice core study. In the section of dating, i.e. 2.1.4, authors used*
*the mean value of the d18O of the whole dataset (-15.5 ‰ as a threshold to separate between the warm*
*and cold seasons. This suggestion should be verified and/or confirmed by the data of d18O in*
*precipitation at the GNIP stations around the ice core drilling site. Another way to test the effectiveness*
*of the division of seasons in ice core is to discern if there is a consistency between the ratio of warm*

*season accumulation rate to cold season accumulation rate (in table 3) and that of precipitation at the*
*adjacent meteorological stations (this method was used by Wang et al (2002, Annals of Glaciology,*
*Vol.35, 273-277) in a Himalayan ice core). Authors also mentioned that the other parameters with*
*seasonal variational characteristics, such as dust and ammonium concentrations, were used to identify*
*the warm/cold season in the ice core profile. It would be better to display the variations of these*
*parameters in the Fig. 3.*
The discussion of the dating methodology will be expanded. Also we will add the dust and ammonium
concentration profiles to the fig. 3. See the answer to the reviewer 1.
*3. Authors calculated the correlation between temperature and d18O in the Lines 329-332 of the text*
*using the 11-year running means for the different periods, and found that the correlations changes with*
*time. If possible, authors can do this by a sliding window method used by Wang et al. (2003,*
*Geophysical Research Letters. Vol.30, No.22, doi: 10.1029/2003GL018188) in a Tibetan ice core.*
*Another issue is that the data series used in the paper ended in 2013, why their 11-year running means*
*also ended in 2013 (shown in Fig. 11)?*
We will reconsider these calculations according to the reviewer's suggestions. We added the sliding
window correlation graph to the fig.11. We would like to note that the periods for comparison on fog.
11 refer to the periods in the unsmoothed data. So it seems that the running means ended in 2013.
Actually, the number of points is less than it would be if we used the full record.
**(see fig. 11 for the correlation plot and regression equations as well as for the sliding window**
**correlation plot). The 10-years sliding window correlation shows the same result, i.e. sharp**
**changes of the correlation between these parameters with predominant negative correlation.**
*4. The significance test in the paper should be paid much attention, especially for the datasets of 11-*
*year and 20-year running means. The degree of freedom can be reduced sharply for the running mean*
*datasets. For example, as for the 11-year mean data sets over the period of 1994-2013, their degree of*
*freedom is only 2 (20/11 is about 2).*
We agree with the comment and will broaden the discussion of the statistical methods used for the
calculations. In the example of the reviewer the degree of freedom ($N - 2n - 2$, where N is number of
data points and n – smoothing period) are $35 - 22 - 2 = 11$ for the period from 1914 to 1928 and from
1994 to 2013, and $65 - 22 - 2 = 41$. In this case the correlations discussed in the paper are still
statistically significant with $p<0.05$.

**2.3 Statistical methods**

**For the correlation analysis we used Pearson correlation coefficient. Statistical significance was estimated with the Student significance test. When compared running means records we calculated the degrees of freedom as N − 2n − 2, where N is number of data points and n − smoothing period.**

*5. In the paragraph, Lines 343-346, authors should present the results of the seasonal cycle of precipitation isotopic composition calculated by using the LMDZiso model, and compare that with the ice core record in one chart.*

Ok, we will add it

**Calculation of the seasonal cycle of precipitation isotopic composition using the LMDZiso model (Risi et al., 2010) do not correspond to the results obtained from the ice core in absolute values or in amplitude (Fig. S5). This can be explained by a complicated relief of the region that influences strongly the isotopic composition, but it is not taken into account in the model. Also in summer Elbrus is in a local convective precipitation system that is not included in the model.**

*6. When discussing the variations of δ18O in precipitation in lines 362-365, the continental effect should be considered.*

Ok, we will add the discussion of the continental recycling to the section

**It is also the continental recycling of moisture (Eltahir and Bras, 1996) that influences the water isotopic composition. Due to this process the δ18O values became lower while *d* values increase (Aemisegger et al., 2014) which is observed in our ice core data.**

*7. In Tables 2 and 4, the period of calculation should be presented.*

Ok, we will add it. See the answer to the reviewer 1.

*8. Line 321, "in the Alps by (Bohleber et al., 2013)" should be "in the Alps by Bohleber et al. (2013)".*

Ok, we will correct it

**Another research performed in the Alps by Bohleber et al. (2013)**

*9. Line 327, "the methods described by (Bohleber et al., 2013)" should be "the methods described by*
*Bohleber et al. (2013)".*

Ok, we will correct it

**the methods described by Bohleber et al. (2013) because of the relatively short and sparse original**
**datasets.**

[Figure]

**Fig. 1: Map showing the region around Elbrus (black rectangle in the world's map in the lower**
**right corner), with shading indicating elevation (m above sea level). Drilling sites are indicated**
**with red filled circles, GNIP stations as green filled circles, and meteorological stations as blue**
**dots. Stations situated to the south of the Main Caucasus Ridge according to the precipitation**
**cycle pattern are shown using a blue dot with white outside circle and the stations situated to the**
**north are displayed with black outside circle (see text for the details). The brown dotted line**

**shows the border between two types of precipitation seasonal cycles. The number of the various**
**stations refers to Table 1 for their detailed description.**

[Figure]

**Fig. 3: Illustration of the scheme used to identify warm and cold half-years (respectively indicated**
**by the light red and light blue shaded areas) based on the deviation of the mean δ¹⁸O values from**
**the long-term average value. The purple lines depict the melt layers observed in the core, dust**
**layers are shown in orange and ammonium concentration graph (Mikhalenko et al., 2015) is in**
**green.**

[Figure]

[Figure]

**Fig. 11. Correlation plot and regression lines for the 11-year running means of the annual local temperature and annual δ$^{18}$O (upper panel) and 10-years sliding window correlation coefficients of the same parameters .**

[Figure]

**Fig. S5. Comparison of the precipitation isotopic composition seasonal cycle at the Elbrus**
**Western Plateau: derived from the ice core data (blue line) and calculated in LMDZiso (Risi et al.,**
**2010) model (green line).**

[revised manuscript text omitted]

Отформатировано ... [1]
Отформатировано ... [2]
Отформатировано ... [3]
Отформатировано ... [4]
Отформатировано ... [5]
Отформатировано ... [6]
Отформатировано ... [7]
Отформатировано ... [8]
Отформатировано ... [9]
Отформатировано ... [10]
Отформатировано ... [11]
Отформатировано ... [12]
Отформатировано ... [13]
Отформатировано ... [14]
Отформатировано ... [15]
Отформатировано ... [16]
Отформатировано ... [17]
Отформатировано ... [18]
Отформатировано ... [19]
Отформатировано ... [20]
Отформатировано ... [21]
Отформатировано ... [22]
Отформатировано ... [23]
Отформатировано ... [24]
Отформатировано ... [25]
Отформатировано ... [26]
Отформатировано ... [27]
Отформатировано ... [28]
Отформатировано ... [29]
Отформатировано ... [30]
Отформатировано ... [31]
Отформатировано ... [32]
Отформатировано ... [33]
Отформатировано ... [34]
Отформатировано ... [35]
Отформатировано ... [36]
Отформатировано ... [37]
Отформатировано ... [38]
Отформатировано ... [39]
Отформатировано ... [40]
Отформатировано ... [41]
Отформатировано ... [42]
Отформатировано ... [43]
Отформатировано ... [44]
Отформатировано ... [45]
Отформатировано ... [46]
Отформатировано ... [47]
Отформатировано ... [48]
Отформатировано ... [49]
Отформатировано ... [50]

| Стр. 51: [1] Отформатировано | Anna | 14.11.2016 16:48:00 |
|---|---|---|

Шрифт: не полужирный

| Стр. 51: [1] Отформатировано | Anna | 14.11.2016 16:48:00 |
|---|---|---|

Шрифт: не полужирный

| Стр. 51: [2] Отформатировано | Anna | 14.11.2016 16:53:00 |
|---|---|---|

Английский (США)

| Стр. 51: [3] Отформатировано | Anna | 14.11.2016 16:54:00 |
|---|---|---|

Английский (США)

| Стр. 51: [4] Отформатировано | Anna | 14.11.2016 16:54:00 |
|---|---|---|

Английский (США)

| Стр. 51: [5] Отформатировано | Anna | 14.11.2016 16:52:00 |
|---|---|---|

Английский (США)

| Стр. 51: [6] Отформатировано | Anna | 14.11.2016 16:53:00 |
|---|---|---|

Английский (США)

| Стр. 51: [7] Отформатировано | Anna | 14.11.2016 16:53:00 |
|---|---|---|

Английский (США)

| Стр. 51: [8] Отформатировано | Anna | 14.11.2016 16:53:00 |
|---|---|---|

Английский (США)

| Стр. 51: [9] Отформатировано | Anna | 14.11.2016 16:54:00 |
|---|---|---|

Английский (США)

| Стр. 51: [10] Отформатировано | Anna | 14.11.2016 16:48:00 |
|---|---|---|

Шрифт: полужирный

| Стр. 51: [10] Отформатировано | Anna | 14.11.2016 16:48:00 |
|---|---|---|

Шрифт: полужирный

| Стр. 51: [11] Отформатировано | Anna | 14.11.2016 16:53:00 |
|---|---|---|

Английский (США)

| Стр. 51: [12] Отформатировано | Anna | 14.11.2016 16:53:00 |
|---|---|---|

Английский (США)

| Стр. 51: [13] Отформатировано | Anna | 14.11.2016 16:54:00 |
|---|---|---|

Английский (США)

| Стр. 51: [14] Отформатировано | Anna | 14.11.2016 16:54:00 |
|---|---|---|

Английский (США)

| Стр. 51: [15] Отформатировано | Anna | 14.11.2016 16:54:00 |
|---|---|---|

Английский (США)

| Стр. 51: [16] Отформатировано | Anna | 14.11.2016 16:53:00 |
|---|---|---|

Английский (США)

| Стр. 51: [17] Отформатировано | Anna | 14.11.2016 16:54:00 |
|---|---|---|

Английский (США)

| Стр. 51: [18] Отформатировано | Anna | 14.11.2016 16:54:00 |
|---|---|---|

Английский (США)

| Стр. 51: [19] Отформатировано | Anna | 14.11.2016 16:55:00 |
|---|---|---|

Английский (США)

| Стр. 51: [20] Отформатировано | Anna | 14.11.2016 16:54:00 |
|---|---|---|

Английский (США)

| | | |
|---|---|---|
| **Стр. 51: [21] Отформатировано** | **Anna** | **14.11.2016 16:54:00** |

Английский (США)

| | | |
|---|---|---|
| **Стр. 51: [22] Отформатировано** | **Anna** | **14.11.2016 16:55:00** |

Английский (США)

| | | |
|---|---|---|
| **Стр. 51: [23] Отформатировано** | **Anna** | **14.11.2016 16:55:00** |

Английский (США)

| | | |
|---|---|---|
| **Стр. 51: [24] Отформатировано** | **Anna** | **14.11.2016 16:55:00** |

Английский (США)

| | | |
|---|---|---|
| **Стр. 51: [25] Отформатировано** | **Anna** | **14.11.2016 16:55:00** |

Английский (США)

| | | |
|---|---|---|
| **Стр. 51: [26] Отформатировано** | **Anna** | **14.11.2016 16:56:00** |

Английский (США)

| | | |
|---|---|---|
| **Стр. 51: [27] Отформатировано** | **Anna** | **14.11.2016 16:56:00** |

Английский (США)

| | | |
|---|---|---|
| **Стр. 51: [28] Отформатировано** | **Anna** | **14.11.2016 16:56:00** |

Английский (США)

| | | |
|---|---|---|
| **Стр. 51: [29] Отформатировано** | **Anna** | **14.11.2016 16:55:00** |

Английский (США)

| | | |
|---|---|---|
| **Стр. 51: [30] Отформатировано** | **Anna** | **14.11.2016 16:55:00** |

Английский (США)

| | | |
|---|---|---|
| **Стр. 51: [31] Отформатировано** | **Anna** | **14.11.2016 16:55:00** |

Английский (США)

| | | |
|---|---|---|
| **Стр. 51: [32] Отформатировано** | **Anna** | **14.11.2016 16:56:00** |

Английский (США)

| | | |
|---|---|---|
| **Стр. 51: [33] Отформатировано** | **Anna** | **14.11.2016 16:56:00** |

Английский (США)

| | | |
|---|---|---|
| **Стр. 51: [34] Отформатировано** | **Anna** | **14.11.2016 16:56:00** |

Английский (США)

| | | |
|---|---|---|
| **Стр. 51: [35] Отформатировано** | **Anna** | **14.11.2016 16:55:00** |

Английский (США)

| | | |
|---|---|---|
| **Стр. 51: [36] Отформатировано** | **Anna** | **14.11.2016 16:55:00** |

Английский (США)

| | | |
|---|---|---|
| **Стр. 51: [37] Отформатировано** | **Anna** | **14.11.2016 16:55:00** |

Английский (США)

| | | |
|---|---|---|
| **Стр. 51: [38] Отформатировано** | **Anna** | **14.11.2016 16:56:00** |

Английский (США)

| | | |
|---|---|---|
| **Стр. 51: [39] Отформатировано** | **Anna** | **14.11.2016 16:56:00** |

Английский (США)

| | | |
|---|---|---|
| **Стр. 51: [40] Отформатировано** | **Anna** | **14.11.2016 16:57:00** |

Английский (США)

| | | |
|---|---|---|
| **Стр. 51: [41] Отформатировано** | **Anna** | **14.11.2016 16:55:00** |

Английский (США)

| | | |
|---|---|---|
| **Стр. 51: [42] Отформатировано** | **Anna** | **14.11.2016 16:55:00** |

Английский (США)

| | | |
|---|---|---|
| **Стр. 51: [43] Отформатировано** | **Anna** | **14.11.2016 16:56:00** |

Английский (США)

| | | |
|---|---|---|
| **Стр. 51: [44] Отформатировано** | **Anna** | **14.11.2016 16:56:00** |

Английский (США)

| | | |
|---|---|---|
| **Стр. 51: [45] Отформатировано** | **Anna** | **14.11.2016 16:57:00** |

Английский (США)

| | | |
|---|---|---|
| **Стр. 51: [46] Отформатировано** | **Anna** | **14.11.2016 16:55:00** |

Английский (США)

| | | |
|---|---|---|
| **Стр. 51: [47] Отформатировано** | **Anna** | **14.11.2016 16:56:00** |

Английский (США)

| | | |
|---|---|---|
| **Стр. 51: [48] Отформатировано** | **Anna** | **14.11.2016 16:56:00** |

Английский (США)

| | | |
|---|---|---|
| **Стр. 51: [49] Отформатировано** | **Anna** | **14.11.2016 16:57:00** |

Английский (США)

| | | |
|---|---|---|
| **Стр. 51: [50] Отформатировано** | **Anna** | **14.11.2016 16:56:00** |

Английский (США)

| | | |
|---|---|---|
| **Стр. 51: [51] Отформатировано** | **Anna** | **14.11.2016 16:56:00** |

Английский (США)

| | | |
|---|---|---|
| **Стр. 51: [52] Отформатировано** | **Anna** | **14.11.2016 16:57:00** |

Английский (США)

---

## Referee Report (RR1)

**General**

The study presents a 100 year record of water stable isotopes derived from combination of several alpine shallow cores and a deep ice core collected at Mt. Elbrus in the Caucasus. Thanks to the high annual net accumulation rate at the site, high temporal resolution could be achieved, allowing obtaining a seasonally resolved data set. Meteorological data, reanalysis temperatures, GNIP isotope data and isotope modeling results as well as atmospheric circulation indices are used to investigate the regional climate and for the discussion of the ice core record. The study concludes that for the ice core site the isotopic composition in summer is related to local temperature whereas in winter it is modulated mainly by large scale atmospheric circulation.

Clearly this is a very valuable data set from a region with a lack of high-elevation meteorological data and therefore deserves publication. The drilling location is characterized by limited surface melt and the ice core(s) analysed are of high quality. Both of which emphasizes the presented records with clear seasonal variations to be useful for their interpretation as climate proxies. Because of the clear seasonality, the dating by annual layer counting is very convincing. However, and here I have to largely repeat the criticism of all three referees who reviewed the manuscript after its first submission, the applied separation into seasonal data, the applied statistical methods (and the lack of some of them) together with the imprecise writing (partly also related to language) does not allow to convincingly support the conclusions made. In their reply, the authors did address the concerns being raised in the previous review but their argumentation, resulting in minor changes only (not major as requested) is not very convincing either. Until the still persisting main issues are solved it (still) makes not much sense to provide detailed comments regarding interpretation and conclusion of the final results as those may (or may not) change. Instead, I once again try to summarize the main concerns adding another level of details and ideas how they might be addressed. Hopefully this will provide help for improvement of this potentially valuable manuscript. In summary, the current manuscript still requires major revision.

Because the manuscript uploaded after the open discussion seems not to be the revised version (File Upload 22 Nov 2016), my review refers to the "track changes version" attached at the end of the Author's Response file (also the line numbering). In my review I will also discuss the Author's response to the *Referee comments (in italic)* made during the open discussion.

**Separation into seasonal data: **Main point of concern**.**

Only once this issue is properly solved, the points discussed later on should be addressed because some of the current results/values might change significantly (not necessarily though).

*Referee 1 wrote: "…is conducted by implementing a threshold (average d18O value of -15.5‰ for the entire record), thereby inherently presuming a d18O-temperature relationship and the absence of a trend. This introduces a circular argument when examining the temperature dependence of the resulting warm and cold season record."*
The main point here is the "circular argument". Even when the approach how the separation is performed (d18O threshold) may lead to seasonal separation in agreement with reality we cannot be sure if this is the case unless there is independent confirmation. More details will be provided in the following.

In their reply, the authors argue: "…, we think that the proposed method of dating when the border between warm and cold seasons is the 100-years mean value is the best one for this very ice core." and later on "We think that the annual cycle of the isotopic composition is influenced by local temperature while interannual variations depend on the other factors."
Well, the first point is not really an argument whereas the second point is an assumption. This assumption defines the outcome and thus the outcome cannot be interpreted as a climate signal (circular argument).
It is very likely that the annual cycle of the isotopic composition is influenced by local temperature as indicated by the clear seasonal variation in the signal. Interannual variations likely also depend on other factors but at least partly they may be influenced by temperature as well. How much these other factors and temperature have contributed in the past might have changed over time and is a focus of the study. By using the 100 yr mean the possibility of longterm trends in annual T (e.g. on a decadal scale) is neglected. Instead such changes observed in d18O are assigned either to changes in cold/warm season temperatures or changes in seasonal accumulation (or a combination of both). This is much less complex for annually resolved data. For those, changes in accumulation do not depend on any initial assumption but can be discussed directly. For annually resolved T a complication because of a potential shift in the seasonal p distribution remains. However, this can be estimated by either assigning the observed increase/decrease in accumulation fully to either of the seasons. What would the expected shift in cold/warm season d18O be (e.g. more acc in winter results in lower d18O for the cold season)? What does the reanalysis data suggest to which season the change in accumulation should be assigned? Can the observed decrease in winter minima (e.g. depth around 65-100 m depth) be explained by this or do they indeed suggest that winters during that period were indeed slightly colder? Could it be a result of increased sampling resolution for this

period (higher resolved winter data resulting in a less smoothed winter signal → lower minima)? This should be discussed and for this and the above reasons I strongly suggest including discussion of annually resolved data which currently is completely ignored. Also see comments in the following.

…further in the reply: "Accumulation at the drilling site has been investigated sporadically (see review in Mikhalenko et al., 2015). We cannot use the meteorological observations from the nearest weather stations as these stations situated at sufficiently lower elevation and belong to two different groups as discussed in section 3.1. The ice core is the only source for the information about the seasonal cycle of this parameter."
I agree about the meteorological data. However, there should be regional reanalysis (or modeling) data available which at least might give some indication of potential changes of the seasonal precipitation cycle. To which extent do those agree/support your observations in the ice core on an annual and seasonal scale (see comment above)? Please discuss.

…and further: "In order to better illustrate the dating methodology we will add the ammonium concentration and dust concentration profiles to Fig. 3. Layers with the high dust concentration have been precisely dated by Kutuzov et al. (2013) for the 2012 ice core. Their results show that the separation of the core into a warm and cold season part using the average value of δ18O is appropriate for this drilling site at least for the period from 2009 till 2012 that was investigated in the paper."
So the solution to avoid the circular agreement is presented right here by introducing the chemistry data as an independent parameter. Chemistry largely relates to seasonal transport (vertical, convection) and seasonal emission (e.g. NH4+). Accordingly it is obviously a good choice to be used for separation into seasons. For the data in the cited study as well as for the period shown in Fig. 3 for which the d18O mean is very similar to the 2009-2012 period and actually also very close to the 100 yr mean of -15.5 permil, this is convincing and suggests that the chemistry records could indeed be used for separation into seasons (or d18O if only this period was considered). So why not just use d18O entirely as suggested by the authors? First because of the circular argument and second because of the period 2009-2012 likely not being representative for the last 100 years. In other words, the approach using a mean d18O value is especially problematic for the depths (i.e. periods) where d18O has a strong trend and differs significantly from the mean (see manuscript Fig. 2 below with these regions marked in red).

[Figure]

To clarify once more the problem of circular agreement on can use the above figure for illustration: if for the separation into seasons a straight line through the data at the value of -15.5 per mil is drawn one can imagine what the outcome likely will be. For the red marked region (around 65 to 105 m depth) the warm season (denoted as summer in the manuscript) d18O values will roughly be similar as for the other depth intervals because the maxima do not vary much. The cold season d18O values on the other hand will become slightly lower (see minima). At the same time the accumulation for the warm period will become smaller whereas accumulation will become bigger for the cold period. Any outcome will thus be defined by our initial assumption and accordingly cannot be interpreted as the reflection of a climatic signal.

Not knowing anything in the first place, looking into annually resolved data (see comments before), may provide us some initial, record based information of what really might have happened during these periods. Was there an increase/decrease in annual precipitation? What is the estimated potential effect on d18O if this change is either fully related to an increase/decrease solely in one season? Could there be a potential effect due to the sampling resolution if the accumulation increased/decreased strongly during one of the seasons (e.g. more pronounced minima)? All of those possibilities could be discussed based on the available station or re-analysis data (Which assumptions are most likely based on those independent results?). As pointed out earlier, whereas a change in annual accumulation can directly be extracted from the ice core record (after corrected for potential thinning) a decrease in annual d18O might not be indicative of a decrease in annual temperatures since a shift to cold season precipitation could have this effect and the observation would accordingly be unrelated to temperature. Using the chemistry (especially NH4+ with main emission in the warm season) already could solve one part of the puzzle, namely the question if the observed increase/decrease in accumulation is related to the cold or the warm season (or both). With this information, one can already come up with a better estimate of the effect of precipitation shift on d18O. Only then, one might start discussing the seasonally resolved data.

*Reviewer 1 also wrote: "…the question is if you could investigate a longer time period (potentially showing a trend in temperature) and longerterm averages to smooth the effect of year-to-year shifts in precipitation/accumulation."*
The authors showed Fig.2 with a linear trend as a response. I do not think this is what the reviewer meant and accordingly the correlation analysis with 3-, 5-, and 7-years running means is out of context here (and should be deleted again). Instead the reviewer's idea seems to be to reduce the high frequency signal (i.e. sub-decadal variations). Such a strongly smoothed signal could then be used as the threshold instead of the 100 yr mean to distinguish between the warm/cold seasons (see again manuscript Fig. 2 above, hand-drawn green dotted line). I am not supporting this idea as one assumption would just be replaced by another one and the circular argument would still persist. With this approach the question would be what the variations in this low frequency signal are related to? Are those decadal T variations or changes in climatic pattern? Because d18O is used as the threshold parameter one could not distinguish the two. Again, an independent parameter such as the chemistry data to split into seasons should allow overcoming this problem.

**In summary:**

The annually resolved data should be investigated first and based on the thereby gained information one can start interpreting the seasonal data which has to be derived by splitting the years based on an independent parameter (e.g. chemistry). Also for the chemistry data a threshold should be defined to separate between cold/warm (or summer/winter) seasons (or at least to indicate the onset of the seasons because in some cases one might see levels below threshold mid-summer e.g. due to a dilution effect in a high precipitation event). Preferentially multiple species (e.g. $NH_4^+$, dust/$Ca^{2+}$) are used to overcome potentially unclear separation for some years. Be aware that since e.g. $NH_4^+$ likely has a trend due to an increase in anthropogenic emission, this trend (not necessarily linear) has to be removed first (or considered for the threshold).

To avoid the circular argument, the authors should use the chemistry data for separation. Another option would be the approach chosen by Mariani et al., 2014 where the record is simply divided into 12 equally spaced bins between peaks (mid-summer; or dips accordingly mid-winter). This approach however assumes equal distribution of annual precipitation. However, by selecting only summer (JJA) and winter months (DJF), the so introduced potential bias/error (if present at all considering the S precipitation pattern) is reduced. It could be estimated if assuming the N or S pattern instead (probably in the order of 10-20%). Another approach (heavily smoothed signal as the threshold instead of the 100 yr mean) was suggested by Reviewer 1, which however also has some caveats as pointed out before.

**Other major comments:**

Seasons and summer winter definition:

The terms summer and winter are used for the ice core data separated into two seasons (e.g. in the Abstract line 404). Since the year is thereby divided in two seasons only this can certainly not be correct. The authors do give a definition of summer (May-Oct) and winter (Nov-Apr) rather late in the manuscript. Nevertheless, this definition is very uncommon and certainly extremely confusing. I suggest sticking entirely to the term warm/cold season with this term being defined in the very beginning of the manuscript (indicate months belonging to the respective seasons).

Correlation:

Throughout the manuscript it is difficult to keep track in what resolution the correlation analysis were performed (annual, seasonal, multiannual/smoothed data?). With at least the numbers of years included in the Tables this has already been slightly improve in the new version. Still it is unclear. I thus suggest to include the time period (19xy – 20zx?) and number data points (n=?) instead. This information should also be given in the text.

Line 586 ff (new section 2.3 Statistical methods):

The calculation of the degree of freedom for the smoothed data set is not correct. The estimate of Reviewer 3 was much better. See e.g. Friston et al., 1994 and 1995; Worsley and Friston, 1995. I will try to give a more intuitive explanation of the results therein here:

Consider a series of n independent observations of a population of mean m. The variance of the population is given by $\sigma^2 = \frac{\sum(X_i - \mu)^2}{N}$ which is best estimated by the sample variance $s^2 = \frac{\sum(X_i - \bar{X})^2}{n-1}$. The denominator in this expression is the number of degrees of freedom of the sample, and is one less than the observations, since only n-1 points are needed to describe the sample, the "last" point being determined by the mean. Now if the data set is smoothed with e.g. a 3 point running filter such that $X_n = (X_{n-1} + X_n + X_{n+1})/3$. The sum of the square deviations from the mean of the data is now (on average) three times less than that of the unsmoothed data. This means that the population variance is now best approximated by $s^2 = \frac{\sum(X_i - \bar{X})^2}{(n-1)/3}$ and implying that the smoothed data set has (n-1)/3 degrees of freedom. Applied to your case with a 11 yr running mean this results for the period 1994-2013 with df=(10-1)/11=0.82 and for 1914-1928 with df1=(15-1)/11=1.27. Since you then combine these two data sets for the correlation analysis the degree of freedom for this combined set is df2= ((10-1)+(15-1))/11 = 2.09. Now for the correlation analysis this results with **dfTotal**≈2*df2-2= **2.2**. For the period 1929-1993 the dfTotal is 2*((65-1)/11)-2≈**9.6**.

As a consequence the significance levels of all correlations using smoothed data have to be reconsidered. Considering this, also the newly added panel in Fig. 11 does not make any sense as in the sliding window the number of data points is even further reduced. It should thus be removed as it does not contain any useful information.

2.1.5 Diffusion of stable isotopes
Line 565-566: "Moreover we would find a positive correlation between accumulation rate and seasonal amplitude of δ18O."
I do not see why you would expect this to be correlated under this assumption. I think what is meant is the actual layer thickness and not the accumulation rate?

3.1 Regional climate
Line 600-604: "Meteorological data depict large regional variations in the seasonal cycle of precipitation. To the south of the Caucasus, there is no distinct seasonal cycle (Fig. 4a), showing the climatology for the Klukhorsky Pereval station. In fact, the Klukhorsky Pereval station is situated north of the Main ridge, but in terms of the seasonal cycle of precipitation it undoubtedly belongs to the southern group. But we are nevertheless using this station as an example because of the uninterrupted record of temperature and precipitation for the 1966-1990 period."
The way it is written here the choice is not very convincing. Reading further on in the manuscript I agree that this seems to be the best choice. But this should become clear at this point already. Also, in the new Fig.1 this station seems to be S of the main ridge?

You are in the fortunate position to have station data both from the north (2-5) and the south (most relevant probably 9 and 10, maybe also 1) as well as high elevation station data for both sides (N: 6,8 and S; 7). As a further plus, the later 3 are in very close proximity to the drill site. I suggest to show the precipitation distribution for all station data (at least in the supplement) and to discuss the patterns according to the groups (N, S, high elevation with N and S indicated) with the final conclusion why this station was chosen. Also see next comment.

Line 606-609: "Moreover, the annual precipitation rate to the south of the Caucasus is much higher than to the north. For example, the typical annual precipitation rate to the north of the Caucasus at the altitude close to the sea level is 500 mm per year, while to the south of the Caucasus at the same altitude it is about 1500 mm. The amount of precipitation in the region is affected by the altitude and the distance from the sea shore."

Line 616-619: "For precipitation data, available in this region since 1966, we considered two different stacks (fig. S4), separating the stations with a distinct seasonal cycle from those where no seasonal cycle was identified for precipitation rates. We coherently used the reference period from 1966 to 1990 for normalization for both precipitation rate and temperature."

Presentation of
   a) Accumulation:

All this information is almost entirely lost in the way Fig. S4 is presented.

1) It is not indicated to which stations the purple lines belong.

2) Because being normalized the absolute values are not visible.

3) The effect of altitude and distance from the sea is not visible since only the stacked record is shown and shown as normalized values.

I suggest following the example in Fig. 8 of Mariani et al. 2014, including all station data on the absolute scale and the altitude indicated behind the station name (one could even think of an additional scatter plot to show the effect of altitude and distance from the sea, respectively). By doing so, the reader is immediately able to visually see all statements made with the additional information about the amplitude of the variations and correlation (visual) between the stations. Since you also discuss seasonal data it would make sense to do provide figures for annual and seasonal values (if the fig does not get too complex, maybe they can be combined).

   b) Temperature:

The above generally also applies to the temperature data sets and its presentation in your Fig. 8 (show all stations, not normalized). Again, in annual and seasonal resolution. Considering my previous comments highlighting the importance of discussing annual values first a panel should be added to Fig. 5 for annual resolution.

Discussion of
   a) Temperature:

In the manuscript the stacked record is then used for further discussion. This assumes that all stations show very similar patterns for the respective region (N or S). Indicated by the standard deviation in Fig. 8, this assumption seems reasonable for the temperature. But also here valuable information is lost by doing so. For example, by using normalized values in Fig. 11, the

information of the slope is lost, which is an important value as it is indicative for the relation between d18O and °C. The slope should be around 0.6 (or in the range of maybe 0.4-0.8). Currently a negative slope is found which is however another issue (see comment later).
I suggest to use the high elevation stations only (one of them should be enough) and correct the T for the laps rate to the altitude of the drill site in order to get the most reliable d18O/T relationship (i.e. slope).

b) Precipitation:

For precipitation the variation between the different stations might be larger. Currently this cannot be assessed with the information provided but will become visible with the suggested changes for presentation of the data.
The information lost if using the stacked and normalized data is the amplitude of variability (both inter-annual and seasonal). Also, the elevation effect in total precipitation should be visible between station and ice core data. If not, it should be discussed.
I suggest to also here using the high elevation stations only instead of the stacked record which in fact likely is not representative for the drill site (too much weight is given to the low elevation stations and the N stations). As pointed out in the manuscript Klukhorski Pereval station (based on the current evaluation with r = 0.65 for both seasons) seems to be the best choice (at least for the current evaluation).

Correlation coefficients for annual resolution should be included in Table 4.

Line 652-654: "As an example we show the seasonal cycle of δ18O and d for Bakuriani station in 2009 (fig. 7). This station is the only one in the region for which the whole uninterrupted dataset for one annual cycle is available. The seasonal amplitude of δ18O is about 10 ‰."
In the revised version the T profile is added to Fig. 7. A quick and dirty calculation based on indicated y-axis-range for d18O (-2 to -18) and T (25 to -5) results in a slope of around 0.6 indicative for the d18O/T dependence. This value is as expected. Please re-calculate more carefully based on the data. How does the dependence change if precipitation weighted T is used instead (if available use daily T and p data for the weighting)? The correlation should improve since d18O can only be recorded if precipitation occurs.

3.2 Ice core records
Line 681-684: "Different patterns of inter-annual to multi-decadal variations appear in the instrumental temperature data (see section 3.1) and ice core δ18O records (Fig 5) emerge for winter versus summer. Consequently, we do not investigate annual mean results, and focus on each season."
I do not understand the statement in the first sentence probably because of language. In any case, the motivation to not use annual data is not convincing at all based on the presented data and for several reasons explained earlier. Based on what assumption can you assume that annual data cannot be compared to meteorological data but seasonal data can? It might be that this will be the outcome of the evaluation of the annual data I proposed earlier but until this is discussed and shown properly such an assumption is pure speculation.

The current splitting of the ice core data contains a large uncertainty by itself. Any finding might thus just be a coincidence. By using the annual data first this additional uncertainty is removed which opposite to the authors argumentation above strongly suggests to investigate the annual results first.

In any case, as suggested before, please add results for the annual resolved data to Table 4 and a panel with annual resolution d18O data to Figure 5. In the current version, the annual data in Fig. 8 cannot be compared anywhere with the annual ice core data.

3.3 Comparison of ice core records with regional meteorological data

Line 714-717: "We found no significant correlation between the ice core δ18O record and regional temperature, neither with the reanalysis data, nor with the observation data, when using the whole period. A significant correlation (r = 0.52, p<0.05) emerges for summer data, when calculated for the period since 1984. The slope for this period is 0.25 per mille per °C. We also repeated our linear correlation analysis using precipitation weighted temperature, and obtained the same results."

The value of 0.25 per mil /°C is very surprising regarding the fact that reasonable correlation was found. It is also a little bit surprising that precipitation weighting did not change the slope (although if no seasonal pattern in p exists this seems not unreasonable).

What data resolution has been used for the precipitation weighting of the temperature? Daily, weekly or monthly data (annual data would make no sense)?

Considering the fact no change was observed, I assume the seasonal distribution of p used for weighting was the one derived for the southern stations? From which station (I suggest to use Klukhorski Pereval station only because it shows highest correlation, see comments before)? How does the correlation and slope look like if the one from the N stations is used instead? How do the correlations and slope look like in this case for the annual and winter d18O record? Please redo the analysis accordingly for the entire period and for the 1984-2013 period.

Since precipitation data is shown only from 1966 I assume the precipitation weighting was only performed for this period? Or did you use the monthly distribution derived for the 1966-2013 period also for the period before, assuming it did not change much (if not done already this might be worth trying)? In any case, the information of what has been done is missing now. Please add.

Line 721-723: "Our results are comparable to those obtained in the Alps by Mariani et al. (2014): again, while the seasonal cycle of ice core δ18O appears related to that of temperature, this is not the case for inter-annual variations, driven by other factors such as changes in moisture sources."

It does not seem that the current results are comparable. See conclusion in the cited paper:
"1. The seasonal cycle of temperature is well-captured in both the Alpine ice cores. On a seasonal scale $\delta^{18}O$ is thus a valid temperature proxy explaining ~60% of the signal.
2. On an annual scale the high variability of precipitation, especially at high-altitude sites, might considerably bias the isotopic signal. For the glacier site with homogeneous distribution of

precipitation throughout the year the mean temperature signal is still partly preserved also on an annual scale. In the other case with strong intraseasonal precipitation variability, the annual mean of δ$^{18}$O was representative only for temperature during precipitation and not for annual mean temperature."

Line 733-735: "The regression analysis showed significant negative correlation between the two parameters. The regression equation for 11-year running means in the 1914-1928 and 1994-2013 differs from the same for the 1929-1993 (see fig. 11 for the correlation plot and regression equations as well as for the sliding window correlation plot).
Based on what criteria can these 2 periods (1914-1928/1994-2013 and 1929-1993) be separated? This seems rather subjective. If looking at the entire period, the correlation would be much worse and the negative slope would not be observed (i.e. both correlation and accordingly the negative slope would not be significant; which is actually also not the case now considering the issue with the correlation analysis of smoothed data pointed out before). Using p weighted data and a different approach for seasonal separation of the d18O (both discussed before) might lead to completely different results anyhow. So please reconsider once the reevaluation is done.
Line 735-737:  "The 10-years sliding window correlation…"
Remove (see discussion of correlation analysis).

Line 943 - New (and old) Fig. 3: Why is there a winter and a summer missing around 31 m? Or should the winter around 33 m cover this entire section from around 31-34?

**Minor comments:**

Abstract - line 403 ff: "In the summer season the isotopic composition depends on the local temperature..."
..and conclusion line 802 ff: "This may explain the significant albeit non persistent correlation of summer δ18O and temperature."
According to the main text this is only true for a certain period (1984-2013)? Please be precise or reconsider the statement.

Line 524-525 (& Fig. S2):
The overlap between the different cores does indeed look very good. Except for the lowermost 2-3 m of the 2013 core with the 2009 core (around 3-7 m depth in Fig. S2). Please comment.

Line 612-613: "The average regional lapse rate was calculated using the available meteorological data. It is minimum (replace with "lowest") in winter (2.3°C per 1000 m) and maximum (replace with "highest") (5.2 °C per 1000 m) in summer (Fig. S3)."
Is this similar for N and S? Are these numbers and Fig S3 for N and S combined or only for one of the 2 regions (or only one station?)?

Line 678-680: "We note that the shallow ice core from the Maili plateau of Kazbek shows the same mean values of δ18O as the Elbrus ice cores during their overlap period. This is a surprise, given the difference in elevation (500 m) and continentality (200 km distance)."
Is this really that much of a surprise? The continentally should make the d18O at Kazbek more negative whereas the lower elevation should make it more positive. In the sum, the two factors seem to cancel out. Can you give some estimates about the size of those two effects and if a 0 sum is reasonable? For the altitude effect, see e.g. Mariani et al., 2014 and references therein.

Line 774-777: "In order to explore the relationships of the Elbrus ice core datasets with the AMO, we used 20-year smoothed data."
I suggest removing this paragraph about AMO entirely. You do show it in Fig 9 and 10 and in some of the tables for comparison with the meteorological data. At this point it does not add anything but takes away from the main focus. Also, by using a 20 yr smoothed record the df is very low for the correlation analysis (<10, see earlier comment) and the result likely not significant anyhow.

Conclusion - Line 789-790: "We found no persistent link between ice cores δ18O and temperature, common feature emerging from non-polar ice cores (e.g. Mariani et al., 2014)."
This is not consistent whit what has been found in the Mariani et al, 2014 paper: See conclusion therein:
"1. The seasonal cycle of temperature is well-captured in both the Alpine ice cores. On a seasonal scale $\delta^{18}O$ is thus a valid temperature proxy explaining ~60% of the signal.
2. On an annual scale the high variability of precipitation, especially at high-altitude sites, might considerably bias the isotopic signal. For the glacier site with homogeneous distribution of precipitation throughout the year the mean temperature signal is still partly preserved also on an annual scale. In the other case with strong intraseasonal precipitation variability, the annual mean of $\delta^{18}O$ was representative only for temperature during precipitation and not for annual mean temperature."

Line 808-810: "The accumulation rate at the drilling site is highly correlated with the precipitation rate and gives information about precipitation variability before the beginning of meteorological observations."
In the current manuscript, the correlation is rather weak and should be changed to "…is significantly correlated…". However, with the current issues this result might change.

**Language:**

…needs to be improved in general and the writing has to be more precise.
Find some (rather randomly chosen) examples below.

Abstract - Line 396-397: Here, we report on the results of the water stable isotope composition from this ice core in comparison with results from shallow ice cores.

The report is not about the comparison between the ice core and the shallow cores (although the measurements at different labs and with different methods have been compared and the cores have been overlapped). The important part is that these datasets are combined and then the results are compared with the meteorological data etc (see line 25-27). Please reconsider this statement and/or reformulate.

Line 398-399: Dating has been performed for the upper 126 m of the deep core combined with shallow cores data.
Also here this is unclear. The records from the deep and shallow cores were combined and dating then performed on this combined dataset down to the ice core depth of 126 m (i.e. combined depth 126 m + xy m from the shallow cores).

Line 399:
The record covers 100 years but two centuries (21$^{st}$ and 20$^{th}$ century).

Introduction - Line 431 ff: "The authors explored the links between the ice cores isotopic composition, local climate and large-scale circulation patterns. They found that in mountain regions isotopic composition of the ice cores governed both by the local meteorological conditions and by the regional and global factors. However, ice core records are complex. For instance, even in areas without any seasonal melt, accumulation is the net effect of precipitation, sublimation, and wind erosion processes, and may significantly differ from precipitation."
The "However" in the 3$^{rd}$ sentence is misleading because what follows is what has been observed and discussed in these papers.
I suggest e.g.: "...global factors. These studies discussed the complexity of interpreting ice core records from high-altitude glaciers due to the potential bias from post-depositional processes and frequent changes in the origin of moisture sources. For instance, even in areas without any seasonal melt, accumulation is the net effect of precipitation, sublimation, and wind erosion processes, and may significantly differ from precipitation."
* * *
References:

Friston, K. J., Jezzard, P. and Turner, R. (1994) Analysis of Functional MRI Time-Series *Human Brain Mapping* **1,**153-171.

Friston, K. J., Holmes, A. P., Poline, J.-B., Grasby, P. J., Williams, S. C. R., Frackowiak, R. S. J. and Turner, R. (1995) Analysis of fMRI Time-Series Revisited *Neuroimage* **2,**45-53.

Worsley, K. J. and Friston, K. J. (1995) Analysis of fMRI Time-Series Revisited - Again *Neuroimage* **2,**173-181.

---

## Referee Report (RR2)

**General**

A 100 year record of water stable isotopes and accumulation derived from the combination of multiple alpine shallow cores and a deep ice core collected at Mt. Elbrus in the Caucasus is presented. The high annual net accumulation rate at the site allows for high temporal resolution and a seasonally resolved data set. Meteorological data, reanalysis temperatures, GNIP isotope data and isotope modeling results as well as atmospheric circulation indices are used to investigate the regional climate and to investigate the parameters recorded by the ice core. The study concludes that for the ice core site the isotopic composition in the warm season is related to local temperature for certain time periods whereas in in the cold season the atmospheric circulation is the main driver of modulation. The accumulation data is used to derive a reconstructed precipitation record for the Caucasus highlands for the time period prior to reliable observations.

The successful drilling and subsequent anaylsis of the presented ice core is already an impressive achievement on its own. The drilling location is characterized by limited surface melt and the ice cores and analysis performed are of high quality. The presented records with clear seasonal variations are certainly useful to gain further insight into the past climate and atmospheric conditions in the studied region which lacks of high-elevation meteorological data. In the current version of the manuscript, most of the issues raised previously in the review process were addressed and implemented. In particular, the approach to split the data into seasonal values is now much more convincing. Still, some issues remain which need more careful investigation and discussion. Addressed later on in more detail, this concerns in particular a) the lack of discussion regarding the dating uncertainty and its effect on the performed correlation analysis, b) the different conclusions drawn for the relation between T and precipitation and their respective ice core proxies which might be caused simply by the different length of the available time series of meteorological data and c) the choice in this version to use the altitude adjusted T (lapse rate corrected station data) for the correlation analysis which might have resulted due to a misunderstanding of a previous review request. Further, some of the figures presented in this version contain serious mistakes which also may or may not have occurred when performing the statistical analysis. This potentially may be a very serious issue and in any case is certainly very unfortunate to happen at this stage of review. Considering the above points, the interpretation and final conclusions drawn by the authors cannot be convincing. Also, the language still needs further improvement, which however is a minor issue.

Taking into consideration all the excellent work and big efforts already undertaken to receive the presented data, it would be a pity to reject this study for publication despite the still existing flaws. I therefore suggest once again major revisions but at the same time would like to urge the authors to invest additional effort and time to carefully reconsider their analysis and interpretation, also being open for potentially different final conclusions even when requiring rewriting substantial sections of the manuscript.

**More detailed major comments:**

Line numbering refers to the current revised version (version 4 I think).

**2.1.4 Dating:**

**Lines 161-162:** What is the estimated dating uncertainty at the bottom of the presented record?

The depth given here as 126 m is confusing because in fact 1914-2013 is contained in the 15 m covered by the shallow cores plus the 126 m covered by the deep core, thus around 140 m in total.

**Line 164:** Also here, 1914-2013 is not contained in 126 m. Please reformulate accordingly, e.g. "…which corresponds to the total of 140 m presented in this study (the 15 m covered by the shallow cores plus the 126 m covered by the deep ice core.".

Accordingly, please reconsider formulation also elsewhere in the manuscript.

**Line 164-165:** The formulation regarding the dating uncertainty ("…relatively small…") is extremely vague. Please indicate a number for the estimated dating uncertainty.

**Line 166:** Reformulate to "In the bottom part of the core the cycles in the isotopic composition are less prominent and dating becomes less reliable leading to a significant increase in uncertainty."

The threshold of exactly 126 m seems arbitrary. I assume the uncertainty already increased above compared to the top let's say 50 m or so. So the estimated dating uncertainty should definitely be indicated as a number somewhere (also see later comments).

**Line 173:** I do not understand what you mean by "stacked"? It is used later on in line 328 where it refers to the overlap of the various cores. This does not seem to be the same thing since here this refers to the entire record of which most is covered by the deep core only. Please explain and clarify accordingly in the manuscript.

**Line 182 and Figure 3:** To me it is not evident at all that "two seasons (one warm and one cold) are partially missing". If so, this would be a year with an exceptional low accumulation. So this certainly is one year of dating uncertainty.

Also **Line 183:** It is unclear what you mean by "we did not use these values for the correlation analysis"? If you have a gap, i.e. not data/value of course you cannot use it. Do you mean that for this missing year xy (if it really is one year considering the then very low accumulation…) you also did not include a value for the meteorological data? This seems trivial and I just hope you did not shift the age scale of the two records against each other when performing the analysis… Please re-check carefully.

**Figure 3:** Again, regarding the dating uncertainty to me it seems questionable if the minima in both d18O and Ammonium is really occurring in summer (double peak) or if this does not rather indicated another winter minima (with a rather high winter d18O). I think this is rather challenging to judge and cannot be decided without some uncertainty. The point is that this

should probably be assigned with another year of dating uncertainty. Together with the gap, this would make ± 2 years of uncertainty just for this section shown in Fig. 3. So the total uncertainty for the year 1914 (including the dating of > 90 additional annual layers) is very likely much higher than ±2 years.

**Line 329 and Fig. 2S:** The inter-core disagreement is indeed small. However, there is at least half a year of disagreement between the very bottom of the 2013 shallow core and the 2009 deep core (around 5 m depth in Fig. S2). This indicates that even in the top seven years (2007-2013) with 2 available absolute time markers (the drilling date of the 2012 and 2009 cores) there exists uncertainty in the dating. For the 93 years before with no absolute time markers available, the dating uncertainty will certainly be quite substantial and will definitely affect the correlation analysis particularly on an annual or seasonal scale. So when discussing the correlations found in Section 3.3. this should be addressed more carefully (a first step has been made by including 3, 5 and 7 yr running averages to the analysis).

**Line 187 / Figures 5, 6 and also 8, 9 and 10:**
I do not see a gap there for the missing year (or season) you discuss for Figure 3?
Please correct.

**3.1 Regional climate:**
**Lines 260-263 and line 270:**
According to the comment made in the previous revision it would be helpful to show the precipitation data for all the stations discussed (lines 260-263). As written in line 270 the authors intended to follow this suggestion but it seems they unfortunately have forgotten to actually include all the data in Fig. S4. Please add.

**Line 269:** It is unclear how the temperature for the drill site was calculated based on the determined lapse rate? Was the seasonal cycle in the lapse rate considered? Please clarify in the text.

Also, the authors followed the suggestions made in the previous review regarding the loss of information (namely the d18O/T relation) when only showing normalized T data. Unfortunately it seems a misunderstanding occurred. The reviewer's idea was this lapse rate adjusted temperature ("drill site T") to be used to determine the d18O/T relationship (i.e. in a way the calibration of d18O as a proxy for temperature). Whereas the correction for the lapse rate is a necessary step to do so, it is not required for the correlation analysis. This is where the misunderstanding happened. The authors now also used this adjusted T data for the correlation analysis (and accordingly also in the figures 8, 9 and 10). This was not suggested! In fact it does not make sense for the following reasons: Because the determined lapse rate certainly comes with an uncertainty (also a change in the rate over time cannot be excluded), an additional source of uncertainty will be introduced to the data set. This will bias the correlation analysis. To include such a bias is unnecessary because the d18O recorded in the ice core also reflects processes taking place on a larger regional scale such as evaporation temperature in the moisture source region, re-evaporation processes etc. and therefore a regional T (i.e. the

station average) is likely most representative for the potential T proxy recorded in the ice core (i.e. d18O). This is a different matter for the precipitation data for which the closest/high altitude stations are most relevant and the authors decision to only use those is a reasonable choice (precipitation and as well as accumulation may vary significantly within regional scale because of orography/altitude effects etc.).

In summary, for the correlation analysis the authors should absolutely stick to the averaged T including all stations as in the previous version (i.e. divided into N and S). It thereby does not matter if they are normalized or simply averaged as for the correlation the results will be the same. For the figures, I suggest to not show the normalized data.

**3.2 Ice core records:**

**Line 332-334:** In the authors response you wrote: "We calculated continental gradient and lapse rate for δ18O using the data from the GNIP stations in the region that are situated at the lower elevations and in our opinion one should be very cautious when using this data for the high elevations ice cores study. The lapse rate is -0.25 ‰/100 m and continental gradient is -0.85 ‰ /100 km. The mean value of δ18O for Kazbek ice core should be 1.25‰ more positive because of elevation difference and 1.7‰ more negative due to continentality factor."

I think the fact that these calculated effects actually match up with what is observed in the two ice cores is a very nice and interesting result. Please include this more detailed description and results given in the above answer to the manuscript.

**3.3 Comparison of ice core records with regional meteorological data:**

**Line 363-385 and Fig. 9 and 10:** In those figures the meteorological temperature data is shifted on the age scale by around 42 years! Shown is 1870-1970 instead of 1910-2013, see the combined figure created form the manuscript figs 8 & 9 and 8 & 10 included on the next page. The same mistake may have occurred when performing the statistical analysis (correlations). Please correct and check carefully!

[Figure]

Fig. 10: Same as fig. 9 but for the warm season.

[Figure]

Fig. 9: Comparison of the ice core record with instrumental regional climate information, for the cold season: δ¹⁸O composite (purple), temperature at the drilling site calculated from the lapse rate (brown), precipitation at the Klukhorskiy Pereval station (light blue) as well as the ice core accumulation estimate (dark blue) and NAO index(green).

**Line 386-390:** This is not very convincing. The problem is that you draw different conclusions for T-d18O and Precipitation-Accumulation relation which might only be caused by the difference in length of the available meteorological time series. In other words, a reasonable correlation was also found for warm season T with d18O for the younger part of the record. Still, the correlation is lost in the older section. How can you exclude the exact same thing is true for the precipitation data?

Also, layer thickness is not equal net accumulation! If precipitation is reconstructed from ice core derived accumulation data, one needs to account for layer thinning (Cuffey and Paterson, 2010). See for example in Mariani et al., 2014 ("The reconstructed net accumulation can be regarded as precipitation proxy, considering few caveats. (i) In order to account for thinning effects, such reconstructions require an accurate description of the glacier ice flow by means of physical models.").
Therefore, please address following the literature (e.g. Schwerzmann et al., 2006; Herren et al., 2013 or probably easiest Equations 1 both in Henderson et al, 2006 and Mariani et al., 2014 which is based on the Nye model).

**Line 376-378 and lines 432-433:** The conclusions and results of Mariani et al., 2014 are still not stated correctly.
In their response to the previous review the authors stated: We agree, that in (Mariani et al., 2014) the authors found strong link between temperature and δ18O on seasonal cycle scale. While on annual scale the signal is biased by other factors. Though they report correlation between δ18O and precipitation weighted temperature, this result is not useful for palaeoclimatology. Citation: "For such a glacier site, a paleotemperature reconstruction is not feasible."
When re-reading the study in question, the authors will realize that 2 separate ice cores are discussed therein: "We assume that at the Grenzgletscher the non-uniform snow deposition throughout the year is more pronounced than at Fiescherhorn (see Section 3.2.1), as it is generally the case in the Southern Alps compared to the Northern Alps (Frei and Schär, 1998; Eichler et al., 2004; Sodemann and Zubler, 2009)."
Obviously, those 2 ice cores are located in meteorologically significantly different regions. So whereas your above statement about the annual scale and the need for precipitation weighting is true for Grenzgletscher it is different for Fiescherhorn (no p weighting was necessary and performed for this core).
Because for the ice core in your study where you point out the relatively equal distribution of precipitation between the seasons, the conclusions/results you should cite are the ones related to Fiescherhorn. Accordingly the results/conclusion from Mariani et al. which you should consider are:
-   3.1.2 Annual scale: "The annual Fiescherhorn $\delta^{18}$O correlates significantly with the Jungfraujoch annual temperature (r=0.44, p<0.01, period 1961-2001). The resulting slope is (0.50±0.16)‰/°C which is consistent with the result based on the seasonal values."
-   Conclusions: "For a glacier site with homogeneously preserved accumulation throughout the year the mean temperature signal is partly preserved on annual scale."

The difference of your finding should be stated accordingly (or as it might change considering the previous comment it might turn out to still be the agreement). So please reformulate.

**Concerns regarding final results and conclusion.**
Out of curiosity after compiling the two figures shown further above, I created the two additional Figures A and B shown below. In this case, I adjusted the scales against each other the way they should be (the aforementioned 42 year shift). Also, following the comment made earlier (see 3.1 Regional climate – Line 269), I here used the earlier version of Fig. 8 (more precisely the normalized T data for the cold and warm period respectively). Assuming reasonable uncertainty in the dating (see comments regarding the dating above) I allowed the age scale of the T data to stretch until reaching a best fit (determined visually due to lack of the actual data which is admittedly a very crude method). For better visibility the d18O records from Fig. 9 and 10, respectively were copied, the background removed and these curves were directly overlaid with the according normalized T from the earlier version Fig. 8 (either warm or cold season T data). For a shift of 8 years, which does not seem unreasonable considering the potential dating uncertainty (i.e. an offset in dating by 8 years out of 100), a strikingly good agreement between T and d18O can visually be seen, particularly for the cold season for which some very characteristic features in the T record can also be found in the d18O record, e.g. between around 1908 and 1935 or 1990 to 2013 with ages referring to the T age scale (upper scale) (Figure A). Also the overall trend is in close agreement except maybe between a short period around 1965 -1970 (Figure A). For the warm season also some characteristic features in both T and d18O exist for the period around 1908-around 1930 and from around 2000-2013 (Figure B), although not as closely related as for the cold season. Also the trend for the warm season does not seem to agree as well as for the cold season. For the location and setting of the ice core site, a reasonable explanation why d18O might be more closely related to T during the cold season than during the warm season could be that in the cold season re-evaporation processes are reduced and transport from the source region is more direct (see manuscript supplement Fig. S1).

In any case, these findings completely disagree with the results and conclusions of the reviewed manuscript although it is based on the exact same data figures:
See for example line 28-30: "In the warm season (May - October) the isotopic composition depends on the local temperature, but the correlation is not persistent in time, while in cold season (November – April), the atmospheric circulation is the predominant driver of the ice core isotopic composition. "
Also line 367-368: "A significant correlation (r = 0.44, p<0.05) emerges for warm season data, when calculated for the period since 1984.", "line 372-373: We didn't find any statistically significant correlations when compared 3-, 5-, 7-years running means of these parameters."
or line 432: "We found no persistent link between ice cores δ18O and temperature on interannual scale…".
Even for the visually good agreement between T and d18O (see Figure A top panel), I would not expect a very high correlation because as stated somewhere earlier, even a 1 year offset can potentially destroy any correlation. However, on a multi-annual scale (3, 5 or 7 year running means) I would expect the correlation to be high.

I thus strongly suggest to carefully revisiting your dating and subsequent data analysis, evaluation and interpretation of your results. Previous publication of the dating can certainly not be a justification for not reconsidering. The potential finding that d18O does indeed reflect T and thus could be used as a T proxy would make this ice core archive certainly much more valuable.

[Figure]

Fig. 9: Comparison of the ice core record with instrumental regional climate information, for the cold season: δ18O composite (purple), temperature at the drilling site calculated from the lapse rate (brown), precipitation at the Klukhorskiy Pereval station (light blue) as well as the ice core accumulation estimate (dark blue) and NAO index(green).

**FIGURE A**

[Figure]

Fig. 10: Same as fig. 9 but for the warm season.

**FIGURE B**

**Minor comments:**

**Table 3 and 4:**
Some significant correlations are not in bold.

**Language (due to lack of time just one example for one of the newly written sections):**
Line 170: …we used ***a*** slightly…
Line 172: …ascribing ***minima*** in….and ***maxima***…
Line 174: ……using ***the*** criteria…described ***by***…
Line 177: …using the ***seasonal signal in the isotopic composition***…
Line 177-178: ***For the*** meteorological data we ***selected the period*** from November to April for the cold season and ***the*** period…
Line 179: There ***are*** some gaps…
…
…
* * *
References:

Cuffey, K., and Paterson, W.S.B.: Ice core studies, in: The physics of the glaciers, 4th ed., Elsevier, Butterworth-Heinemann, USA, 611-674, 2010.

Henderson, K., Laube, A., Gäggeler, H. W., Olivier, S., Papina, T., and Schwikowski, M.: Temporal variations of accumulation and temperature during the past two centuries from Belukha ice core, Siberian Altai, J. Geophys. Res., 111, D03104, doi:10.1029/2005JD005819, 2006.

Herren, P.-A., Eichler, A., Machguth, H., Papina, T., Tobler, L., Zapf, A. and Schwikowski, M.: The onset of Neoglaciation 6000 years ago in western Mongolia revealed by an ice core from the Tsambagarav mountain range, Quat. Sci. Rev., 69, 59–68, doi:10.1016/j.quascirev.2013.02.025, 2013.

Nye, J. F.: Correction factor for accumulation measured by the thickness of the annual layers in an ice sheet, J. Glaciol., 4, 785–788, 1963.

Schwerzmann, A., Funk M., Blatter H., Lüthi M., Schwikowski M., and Palmer A.: A method to reconstruct past accumulation rates in alpine firn regions: A study on Fiescherhorn, Swiss Alps, J. Geophys. Res., 111, F01014, doi:10.1029/2005JF000283, 2006.

---

## Referee Report (RR3)

This is the 4th review of the present study by Kozachek et al.: ***Large-scale drivers of Caucasus climate variability in meteorological records and Mt Elbrus ice cores***. I also reviewed the 2nd and 3rd version. The manuscript has constantly improved with each version. I appreciate the efforts by the authors to address the issues raised in the previous reviews. By now, almost all of these points are convincingly considered. Still, the splitting of the data into seasons seems not quite settled yet, a concern also raised by another reviewer. Further, there are a few explanations provided in the authors' response which did not find their way into the manuscript text although they seem important for clarity. Unfortunately, again the wrong dataset is plotted in one of the figure panels. Although such repeated, basic oversights do not help, I take the authors by their word, trusting that no comparable mistakes happened when evaluating the datasets. The language has improved significantly between the previous and this version, but nevertheless I encourage the authors to take advantage of the language editing offered by the journal. In conclusion, if the (generally) minor points in the comments below can be addressed, I think the manuscript could be accepted for publication.

**Detailed comments:**

**L178-181** This is still unclear. If the maximum values are always assigned to July and the minimum values are always assigned to January, how can in some occasions minimum values suddenly become assigned to summer and maximum values to winter? This makes no sense. The approach described above is similar to the one in Vinther et al. (2010) and the underlying assumption for such an approach (max=summer, min=winter) is 50% winter and summer accumulation. The boundary between summer and winter is then defined by the middle between these two extreme (depth scale in m w.e.). This approach to split the record into seasonal data (cold, warm) allows comparison with the meteorological data separated into the seasons in the way described in the manuscript.

However, the way I understand the approach described in the present version it seems the minima and maxima are always assigned to the middle of the respective season (cold and warm, respectively). But still I cannot imagine why minima/maxima suddenly should become assigned to winter/summer. In any case the describe approach (or my interpretation of its description) contains two additional assumptions: (1) the month of lowest (highest) T is always in the mid-season and (2) this has not changed over the investigated period. Both points and their potential consequences for the analysis due to the fact that both introduce additional uncertainty should in this case be discussed. With the station T data at hand it is easily possible to investigate these two assumptions. Therefore, e.g. plot the months with lowest/highest T against time. The results should show the months of most extreme temperatures to (1) lie in the middle of the respective season and (2) the months when these minima/maxima were observed did not change over time. Actually (1) has been shown to be valid by the data presented in manuscript Fig. 4.

Please explain your approach accordingly or adjust your methodology following the description in Vinther et al. (2010). The subsequent evaluation of the data (correlation analysis etc.) and its interpretation should then also be revised.

**L168-170** With the additional information about the two absolute time markers (1963 and 1912) now provided in L163-164, a lot about the dating was clarified including the reasoning behind the selection of the 100 year period. Clearly, this period was not selected because of the beauty of the number 100 as

suggested in the authors' response, but rather because at that depth the age scale is well defined by the 1912 time horizon. I thus suggest, adding this relevant information to the manuscript along the lines: "This period has been chosen because at this depth, the age scale is well defined by the time horizon found slightly below (Katmai 1912) resulting in a relatively small dating uncertainty of ±2 years, and because of the availability of other records such as local meteorological observations."

**L279-280 and Fig. 8** Is the uncertainty of the defined lapse rate (not indicated in Fig. S3) propagated? In any case, the upper panel (annual means) looks very much the same as in the previous version 4. Please check if the correct dataset is plotted and include the lines indicating the standard deviation across the individual records.

**L280-284 (and Fig. S4)** As suggested, Fig. S4 has now been updated so that it is now consistent with the manuscript text. When now also adding the information about the station altitude to the figure legend, it becomes visually obvious that precipitation variability and particularly for the cold season precipitation amount has a strong altitude effect. With this, the choice to only use the two high altitude stations for precipitation data is clarified. The according text should be added to the manuscript. Therefore, please add altitudes to the legends in Fig. S4 and adjust the text along the lines: "All the precipitation data available for this region since 1966 is shown in fig. S4. Because of the obvious altitude dependence of both precipitation variability and precipitation amount (particularly for the cold season) only the data from the two high altitude stations Klukhorskiy Pereval (???? m asl.) and Mineralnye Vody (???? m asl.) were used for the calculations here. The two stations are further representative for stations with and without a prominent seasonal cycle (Mineralnye Vody and Klukhorskiy Pereval, respectively)."

**L437-438** It is unclear for what season NAO is correlated with regional temperature. I suggest changing to: "For the cold season, the ice core d18O record shows…"

**Fig. 7** There are only 11 increments for the 12 months which is confusing. To be consistent within the manuscript and with the commonly used way for display, please adjust the x-axis scale similar to manuscript Fig. 4.

**Fig. 9** Even though the temperature record in this figure (and in Fig. 10) is now at least plotted on the correct age scale, it is still not correct. In the lowermost panel, not d18O is plotted but normalized temperature (most obvious by the y-axis scale). Please correct.

**Table 4 (and according sections in the text)** Although in this version, the normalized temperatures are again used for the correlation analysis as it was suggested, none of the correlation coefficients changed. I do not expect extreme changes but at least some considering the fact that in the previous version only the station data from Klukhorskiy Pereval and Mineralnye Vody was used.
Further, for the field in the upper left corner, annual means - T vs d18O – the value of 0.16 (n=100) is not significant and should not be bold.

**Language:**

When accepted for publication, please take advantage of the language editing service offered by the journal. One easy fix: in many cases δ18O should be replaced with $\delta^{18}O$.

---

## Author Response (AR2)

We would like to thank the reviewers for the thorough reviews and detailed comments. Here we provide answers to the questions raised in the reviews.

Comments of the reviewers are in blue, our answers are in green, and corrections in the paper text are in black.

Reviewer 1.

I urge the authors to carefully read section 3.3 in Vinther et al., 2010 and adopt a similar line of thinking – if not adopting a similar approach to dividing the seasons.

We changed the method for the seasons dividing following the recommended paper. The results of the new records analysis are discussed below at the answers to the reviewer 2.

I can't find any references of which data is used for NAO, AO and AMO. Please make sure to check all data for references and describe the climate indices in the data section, including definitions and data sources.

References for all the data sources had been presented in the table 1 in all the versions of the paper. We broadened the data section with the description of the indices. AMO index was excluded from the paper following the suggestion of the reviewer 2.

Circulation of the atmosphere influence sufficiently isotopic composition of the ice cores (Casado et al., 2013 and references therein). Atmospheric circulation quantitatively characterized by circulation indices. In this research we used three indices: NAO, AO, NCP that are widely used to characterize European climate (Jones et al., 2003, Thompson and Wallace, 2001, Brunetti et al., 2011 and references therein). Time span and references for the indices are presented in table 1.

NAO (North-Atlantic Oscillation) characterizes type of circulation in Europe, strength of Azores maximum and Icelandic minimum. Positive values of NAO index correspond to lower than usual value of atmospheric pressure in Iceland and higher that usual value of atmospheric pressure at Azores. Negative index correspond to less prominent centers of action in the Norrthern Hemisphere. Usually this index is calculated as difference of atmospheric pressure measured at Reykjavik and Lisbon, Ponta Delgada or Gibraltar. Here we used data from (Vinther et al., 2003 and https:\\crudata.uea.ac.uk\~timo\datapages\naoi.htm) that were calculated using data from Gibraltar station. Negative NAO leads to increase of precipitation rate in Southern Europe, positive NAO leads to increase of precipitation rate in Northern Europe (Hurrel, 1995, Jones et al., 2003, Vinther et al., 2003).

Arctic Oscillation index (AO) also is a characteristic of the Northern Hemisphere circulation. It is used to analyze climatic variability with periods longer that 10 years. It is calculated as EOF of 500 hPa surface. Negative valued correspond to high pressure at the Pole and cooling of Europe, while positive values correspond to low pressure at the Pole and drying of Mediterranean (Thompson and Wallace, 2001). We used AO data from NOAA (http:\\www.cpc.ncep.noaa.gov\products\precip\CWlink\).

NCP (North-Sea Caspian Pattern) index is less widely used, though it was proved that it is convenient to use it in Mediterranean climate studies (Kutiel et al., 1997; Brunetti et al., 2011). The index is calculated as normalized difference of geopotential heights between Caspian and Northern seas.
Positive values correspond to stronger meridional circulation in Europe and lower summer
temperatures, Negative values reflect strengthening of zonal circulation and higher summer
temperatures in Europe (Brunetti et al., 2011). We used NCP data from NOAA
(http:\\www.cpc.ncep.noaa.gov\products\precip\CWlink\).

Reviewer 2.

Separation into seasonal data: Main point of concern.
Only once this issue is properly solved, the points discussed later on should be addressed because some
of the current results/values might change significantly (not necessarily though).

We tried three seasons' separation methodologies: using the fixed value as it was in the previous
version of the paper, method used by Mariani et al., 2014 and by Vinther et al., 2010. The method of
Vinther et al was slightly changed in order to avoid ascribing minima to the warm season and maxima
to the cold season. But we stacked to having the minima and maxima in the middle of the corresponding
season as it is in accordance with meteorological data showing minimum temperatures in Jan-Feb and
maximum temperatures in Jul-Aug. Here we compare three versions of warm and cold seasons
interannual variations of δ18O and accumulation rate. Though the differences are sufficient (fig. A1 and
A2), none of the methods led to finding persistent correlation between δ18O and air temperature.

[Figure]

Fig. A1. Interannual variations of δ18O in cold and warm seasons using different dividing methods. Number in the legend refer to the different dividing methods: 1 – method of Vinther et al., 2010; 2 – method of the fixed threshold; 3 – method of Mariani et al., 2014.

[Figure]

Fig. 2A. The same as fig. 1A but for the accumulation rate.

In the current version of the paper we changed the separation method following the Vinther et al., 2010, also using the ammonium data as an independent marker according to criteria described in (Mikhalenko et al., 2015).
We added this point to the dating section. The fig. 3 was also changed.

2.1.4 Dating

The chronology is based on the identification of annual layers. These are prominent in $\delta 18O$ with the average seasonal amplitude of 20 ‰. For annual mean values we calculated averages of $\delta 18O$ from one minimum of this parameter to another one as well as from one maximum to another. As we found no significant differences between the records obtained with two ways of year allocation we use minimum to minimum dating as more common one. We compared annual layers counting performed independently using the seasonal cycles in the isotopic composition and the ammonium concentration. The discrepancy between two independent chronologies is 2 years at a depth of 126 m. We used the dating based on the isotopic composition data in this paper. This dating is also best fit for the correlation analysis with the meteorological data. Hereafter, we focus our analysis on one century, from 1914 till 2013, which corresponds to the upper 126 m of the core. This period has been chosen because of relatively small dating uncertainty and the availability of other records such as local meteorological observations. At the bottom part of the core the isotopic composition cycles are less prominent and cannot be used for dating, consequently the dating uncertainty is sufficiently higher. The isotopic composition of that part of the core will be discussed elsewhere. In meteorological data we used average values from January to December of each year for the comparison with annual means of ice cores parameter.

For warm and cold seasons allocation we used slightly adapted method from (Vinther et al., 2010). The original method requires ascribing of equal accumulation rate for warm and cold season of each year. We changed the borders between the seasons when needed in order to avoid ascribing minimum of $\delta 18O$ to the warm season and maximum to the cold season. We stacked to keeping the extreme values in the middle of the season as this is in coherence with meteorological data. We also used ammonium concentration as an independent marker, using criteria described on (Mikhalenko et al., 2015). For equivocal situations, we also used additional data: melt layers and dust layers (used to identify the warm season) (Kutuzov et al., 2013) as well as succinic acid concentration data that also have seasonal variations (Mikhalenko et al., 2015).

Figure 3 illustrates the identification of seasons using the isotopic composition seasonal cycle. In meteorological data we used period from November to April for the cold season and period from May to October for the warm season.

Following the suggestion of the reviewer we added annual mean data to all the sections in the paper. When studying the annual means we also tried two versions of the dating. We calculated mean values from minimum to minimum in the ice core data and from Jan to Dec in meteodata. And we did the same from maximum to maximum and from Jul to Jun. As we didn't find any difference in using these
records we present the dataset obtained by the min-min dating as more commonly used one.
**Other major comments:**
Seasons and summer winter definition:
The terms summer and winter are used for the ice core data separated into two seasons (e.g. in the
Abstract line 404). Since the year is thereby divided in two seasons only this can certainly not be
correct. The authors do give a definition of summer (May-Oct) and winter (Nov-Apr) rather late in the
manuscript. Nevertheless, this definition is very uncommon and certainly extremely confusing. I
suggest sticking entirely to the term warm/cold season with this term being defined in the very
beginning of the manuscript (indicate months belonging to the respective seasons).
We changed the terms summer/winter to the warm/cold seasons respectively. The definition of these
seasons is now given in the section 2.1.4.
Figure 3 illustrates the identification of seasons using the isotopic composition seasonal cycle. In
meteorological data we used period from November to April for the cold season and period from May
to October for the warm season.
**Correlation:**
Throughout the manuscript it is difficult to keep track in what resolution the correlation analysis were
performed (annual, seasonal, multiannual/smoothed data?). With at least the numbers of years included
in the Tables this has already been slightly improved in the new version. Still it is unclear. I thus suggest
to include the time period (19xy – 20zx?) and number data points (n=?) instead. This information
should also be given in the text.
We added the time period and the number of data points to the tables 2 and 4. This is also clarified in
the text. We removed the discussion of smoothed datasets from the paper as well as chapter 2.3
statistical methods.
Table 2: Correlation coefficients between meteorological data and indices of large-scale modes of
variability (statistically significant coefficients at $p < 0.05$ are highlighted in bold). The period of
calculation and number of data points (n) for each coefficient are shown in brackets.

| Annual mean | Temperature | P south* | P north* |
|---|---|---|---|
|  |  |  |  |
| NAO | -0.24 (1914-2013, n=100) | -0.24 (1966-2013, n=48) | -0.03 (1966-2013, n=48) |
| AO | -0.34 (1950-2013, n=64) | -0.06 (1966-2013, n=48) | 0.02 (1966-2013, n=48) |

| | | | |
|---|---|---|---|
| NCP | -0.55 (1948-2013, n=66) | 0.26 (1966-2013, n=48) | 0.26 (1966-2013, n=48) |
| | | | |
| Warm season | | | |
| NAO | -0.47 (1914-2013, n=100) | 0.23 (1966-2013, n=48) | 0.03 (1966-2013, n=48) |
| AO | -0.11 (1950-2013, n=64) | 0.08 (1966-2013, n=48) | 0.14 (1966-2013, n=48) |
| NCP | -0.50 (1948-2013, n=66) | 0.34 (1966-2013, n=48) | 0.34 (1966-2013, n=48) |
| | | | |
| Cold season | | | |
| NAO | -0.41 (1914-2013, n=100) | 0.04 (1966-2013, n=48) | 0.26 (1966-2013, n=48) |
| AO | -0.40 (1950-2013, n=64) | 0.14 (1966-2013, n=48) | 0.37 (1966-2013, n=48) |
| NCP | -0.77 (1948-2013, n=66) | 0.25 (1966-2013, n=48) | 0.33 (1966-2013, n=48) |

Table 4. Correlation coefficients between ice core data, meteorological data and indices of large-scale
modes of variability (statistically significant coefficients at $p < 0.05$ are highlighted in bold). The period
of calculation and number of data points (n) for each coefficient is shown in brackets.

| Annual means | $\delta^{18}O$ | Accumulation | $d$ | NAO | AO | NCP |
|---|---|---|---|---|---|---|
| $T.$ °C | -0.01 (1914-2013, n=100) | 0.16 (1914-2013, n=100) | 0.00 (1914-2013, n=100) | -0.24 (1914-2013, n=100) | -0.34 (1950-2013, n=64) | -0.55 (1948-2013, n=66) |
| P north* | -0.30 (1966-2013, n=48) | 0.36 (1966-2013, n=48) | 0.17 (1966-2013, n=48) | -0.03 (1966-2013, n=48) | -0.03 (1966-2013, n=48) | 0.27 (1966-2013, n=48) |
| P south* | 0.06 (1966-2013, n=48) | 0.52 (1966-2013, n=48) | 0.07 (1966-2013, n=48) | -0.24 (1966-2013, n=48) | -0.06 (1966-2013, n=48) | 0.18 (1966-2013, n=48) |
| $\delta^{18}O$ | | -0.20 (1914-2013, n=100) | -0.06 (1914-2013, n=100) | 0.07 (1914-2013, n=100) | 0.41 (1950-2013, n=64) | 0.11 (1948-2013, n=66) |
| Accumulation | | | 0.21 06 (1914-2013, n=100) | -0.29 (1914-2013, n=100) | -0.29 (1950-2013, n=64) | -0.03 (1948-2013, n=66) |

| | δ¹⁸O | Accumulation | d | NAO | AO | NCP |
|---|---|---|---|---|---|---|
| *d* | | | | -0.08 (1914-2013, n=100) | -0.26 (1950-2013, n=64) | -0.14 (1948-2013, n=66) |
| Warm season | δ¹⁸O | Accumulation | *d* | NAO | AO | NCP |
| *T.* °C | 0.13 (1914-2013, n=100) | -0.04 (1914-2013, n=100) | 0.20 (1914-2013, n=100) | -0.02 (1914-2013, n=100) | -0.10 (1950-2013, n=64) | -0.51 (1948-2013, n=66) |
| P north* | 0.01 (1966-2013, n=48) | 0.16 (1966-2013, n=48) | 0.09 (1966-2013, n=48) | 0.13 (1966-2013, n=48) | -0.14 (1966-2013, n=48) | 0.18 (1966-2013, n=48) |
| P south* | -0.27 (1966-2013, n=48) | 0.49 (1966-2013, n=48) | -0.02 (1966-2013, n=48) | -0.01 (1966-2013, n=48) | 0.07 (1966-2013, n=48) | 0.34 (1966-2013, n=48) |
| δ¹⁸O | | -0.42 (1914-2013, n=100) | -0.05 (1914-2013, n=100) | -0.08 (1914-2013, n=100) | 0.16 (1950-2013, n=64) | 0.00 (1948-2013, n=66) |
| Accumulation | | | 0.31 06 (1914-2013, n=100) | 0.00 (1914-2013, n=100) | 0.09 (1950-2013, n=64) | 0.00 (1948-2013, n=66) |
| *d* | | | | 0.00 (1914-2013, n=100) | -0.01 (1950-2013, n=64) | -0.14 (1948-2013, n=66) |
| Cold season | δ¹⁸O | Accumulation | *d* | NAO | AO | NCP |
| *T.* °C | -0.09 (1914-2013, n=100) | 0.11 (1914-2013, n=100) | -0.15 (1914-2013, n=100) | -0.30 (1914-2013, n=100) | -0.45 (1950-2013, n=64) | -0.79 (1948-2013, n=66) |
| P north* | 0.20 (1966-2013, n=48) | 0.21 (1966-2013, n=48) | -0.12 (1966-2013, n=48) | 0.51 (1966-2013, n=48) | 0.37 (1966-2013, n=48) | 0.23 (1966-2013, n=48) |
| P south* | -0.30(1966-2013, n=48) | 0.37 (1966-2013, n=48) | -0.13 (1966-2013, n=48) | 0.26 (1966-2013, n=48) | 0.14 (1966-2013, n=48) | 0.25 (1966-2013, n=48) |
| δ¹⁸O | | 0.05 (1914-2013, n=100) | 0.02 (1914-2013, n=100) | 0.41 (1914-2013, n=100) | 0.41 (1950-2013, n=64) | 0.19 (1948-2013, n=66) |
| Accumulation | | | 0.07(1914-2013, n=100) | -0.18 (1914-2013, n=100) | -0.15 (1950-2013, n=64) | 0.18 (1948-2013, n=66) |
| *d* | | | | -0.06 (1914-2013, n=100) | -0.01 (1950-2013, n=64) | 0.11 (1948-2013, n=66) |

*P south – precipitation rate at the weather stations to the South from the Caucasus, P north –
precipitation rate at the weather stations to the North from the Caucasus.

As a consequence the significance levels of all correlations using smoothed data have to be reconsidered. Considering this, also the newly added panel in Fig. 11 does not make any sense as in the sliding window the number of data points is even further reduced. It should thus be removed as it does not contain any useful information.

We thank the reviewer for providing the comprehensive explanation of the calculations. We removed all the discussion of smoothed datasets, including fig.11.

2.1.5 Diffusion of stable isotopes
Line 565-566: "Moreover we would find a positive correlation between accumulation rate and seasonal amplitude of $\delta18O$."
I do not see why you would expect this to be correlated under this assumption. I think what is meant is the actual layer thickness and not the accumulation rate?

Agree. We tried both: accumulation and actual layer thickness. In both cases no significant correlation was found.

Moreover we would find a positive correlation between layer thickness and seasonal amplitude of $\delta^{18}O$.

The way it is written here the choice is not very convincing. Reading further on in the manuscript I agree that this seems to be the best choice. But this should become clear at this point already. Also, in the new Fig.1 this station seems to be S of the main ridge? You are in the fortunate position to have station data both from the north (2-5) and the south (most relevant probably 9 and 10, maybe also 1) as well as high elevation station data for both sides (N: 6,8 and S; 7). As a further plus, the later 3 are in very close proximity to the drill site.
I suggest to show the precipitation distribution for all station data (at least in the supplement) and to discuss the patterns according to the groups (N, S, high elevation with N and S indicated) with the final conclusion why this station was chosen.

Kukhorsky Pereval station is situated S from the Elbrus but N from the Main Caucasus ridge. The detailed map is shown in Mikhalenko et al., 2015 (fig. 1 b). But in terms of precipitation annual cycle it belongs to the southern group. The brown line on fig. 1 shows the border between two types of precipitation cycle.
We do not find it useful to discuss the seasonal cycles at all the stations as it has already been performed by (Mikhalenko et al., 2015), we added the link to the text. We have chosen two stations for the calculations: Mineralnye Vody for the N stations and Klukhorskiy Pereval for the S stations because of their close position to the drilling site and because of uninterrupted record for the period of precipitation data availability (1966-2013)

Presentation of
a) Accumulation:

All this information is almost entirely lost in the way Fig. S4 is presented.
1) It is not indicated to which stations the purple lines belong.
2) Because being normalized the absolute values are not visible.
3) The effect of altitude and distance from the sea is not visible since only the stacked record is shown
and shown as normalized values.
I suggest following the example in Fig. 8 of Mariani et al. 2014, including all station data on the
absolute scale and the altitude indicated behind the station name (one could even think of an additional
scatter plot to show the effect of altitude and distance from the sea, respectively). By doing so, the
reader is immediately able to visually see all statements made with the additional information about the
amplitude of the variations and correlation (visual) between the stations. Since you also discuss seasonal
data it would make sense to do provide figures for annual and seasonal values (if the fig does not get too
complex, maybe they can be combined).
We changed the records of precipitation in the paper from normalized values to absolute ones. The
records for many of the stations are presented in Shahgedanova et al., 2014 and Tielidze et al., 2016.
We added these references to the text.
b) Temperature:
The above generally also applies to the temperature data sets and its presentation in your Fig. 8 (show
all stations, not normalized). Again, in annual and seasonal resolution. Considering my previous
comments highlighting the importance of discussing annual values first a panel should be added to Fig.
5 for annual resolution.
We changed the presentation of the temperature records to absolute values at the drilling site calculated
using the lapse rate.
Discussion of
a) Temperature:
In the manuscript the stacked record is then used for further discussion. This assumes that all stations
show very similar patterns for the respective region (N or S). Indicated by the standard deviation in Fig.
8, this assumption seems reasonable for the temperature. But also here valuable information is lost by
doing so. For example, by using normalized values in Fig. 11, the information of the slope is lost, which
is an important value as it is indicative for the relation between d18O and °C. The slope should be
around 0.6 (or in the range of maybe 0.4-0.8). Currently a negative slope is found which is however
another issue (see comment later).
I suggest to use the high elevation stations only (one of them should be enough) and correct the T for
the laps rate to the altitude of the drill site in order to get the most reliable d18O/T relationship (i.e.
slope).

We changed the presentation of the temperature records to absolute values at the drilling site calculated
using the lapse rate.
b) Precipitation:
For precipitation the variation between the different stations might be larger. Currently this cannot be
assessed with the information provided but will become visible with the suggested changes for
presentation of the data.
The information lost if using the stacked and normalized data is the amplitude of variability (both inter-
annual and seasonal). Also, the elevation effect in total precipitation should be visible between station
and ice core data. If not, it should be discussed.
I suggest to also here using the high elevation stations only instead of the stacked record which in fact
likely is not representative for the drill site (too much weight is given to the low elevation stations and
the N stations). As pointed out in the manuscript Klukhorski Pereval station (based on the current
evaluation with r = 0.65 for both seasons) seems to be the best choice (at least for the current
evaluation).
Correlation coefficients for annual resolution should be included in Table 4.
We changed the presentation to the absolute values from two stations: Klukhorskiy Pereval
(representative of the S stations) and Mineralnye Vody (representative of the N stations).
Line 652-654: "As an example we show the seasonal cycle of δ18O and d for Bakuriani station in 2009
(fig. 7). This station is the only one in the region for which the whole uninterrupted dataset for one
annual cycle is available. The seasonal amplitude of δ18O is about 10 ‰."
In the revised version the T profile is added to Fig. 7. A quick and dirty calculation based on indicated
y-axis-range for d18O (-2 to -18) and T (25 to -5) results in a slope of around 0.6 indicative for the
d18O/T dependence. This value is as expected. Please re-calculate more carefully based on the data.
How does the dependence change if precipitation weighted T is used instead (if available use daily T
and p data for the weighting)? The correlation should improve since d18O can only be recorded if
precipitation occurs.
We recalculated the slope using the data. The slope is 0.32 ‰/°C. Unfortunately, daily data are not
available for this station as well as for the other GNIP stations in the region.
The seasonal amplitude of $\delta^{18}O$ is about 17 ‰. The slope between δ18O and temperature is 0.32 ‰/°C.
3.2 Ice core records
Line 681-684: "Different patterns of inter-annual to multi-decadal variations appear in the instrumental
temperature data (see section 3.1) and ice core δ18O records (Fig 5) emerge for winter versus summer.
Consequently, we do not investigate annual mean results, and focus on each season."
I do not understand the statement in the first sentence probably because of language. In any case, the
motivation to not use annual data is not convincing at all based on the presented data and for several reasons explained earlier. Based on what assumption can you assume that annual data cannot be
compared to meteorological data but seasonal data can? It might be that this will be the outcome of the
evaluation of the annual data I proposed earlier but until this is discussed and shown properly such an
assumption is pure speculation. The current splitting of the ice core data contains a large uncertainty by
itself. Any finding might thus just be a coincidence. By using the annual data first this additional
uncertainty is removed which opposite to the authors argumentation above strongly suggests to
investigate the annual results first.
In any case, as suggested before, please add results for the annual resolved data to Table 4 and a panel
with annual resolution d18O data to Figure 5. In the current version, the annual data in Fig. 8 cannot be
compared anywhere with the annual ice core data.
We added the annual data as well
3.3 Comparison of ice core records with regional meteorological data
Line 714-717: "We found no significant correlation between the ice core $\delta 18O$ record and regional
temperature, neither with the reanalysis data, nor with the observation data, when using the whole
period. A significant correlation (r = 0.52, p<0.05) emerges for summer data, when calculated for the
period since 1984. The slope for this period is 0.25 per mille per °C. We also repeated our linear
correlation analysis using precipitation weighted temperature, and obtained the same results."
The value of 0.25 per mil /°C is very surprising regarding the fact that reasonable correlation was found.
It is also a little bit surprising that precipitation weighting did not change the slope (although if no
seasonal pattern in p exists this seems not unreasonable).
What data resolution has been used for the precipitation weighting of the temperature? Daily, weekly or
monthly data (annual data would make no sense)?
Considering the fact no change was observed, I assume the seasonal distribution of p used for weighting
was the one derived for the southern stations? From which station (I suggest to use Klukhorski Pereval
station only because it shows highest correlation, see comments before)?
How does the correlation and slope look like if the one from the N stations is used instead?
How do the correlations and slope look like in this case for the annual and winter d18O record? Please
redo the analysis accordingly for the entire period and for the 1984-2013 period.
Since precipitation data is shown only from 1966 I assume the precipitation weighting was only
performed for this period? Or did you use the monthly distribution derived for the 1966-2013 period
also for the period before, assuming it did not change much (if not done already this might be worth
trying)? In any case, the information of what has been done is missing now. Please add.
As the seasonal averages of $\delta 18O$ were changed, the new correlation coefficient is 0.13 for the 100-
years period for the warm season. Again, the correlation is much higher (r=0.44, p<0.05) if we take the
period from 1984 till 2013. The slope is 0.6 ‰/°C. No correlation found for the cold season or for the
annual means.
Calculation of precipitation weighted temperatures using precipitation didn't give any additional
correlation. For the precipitation weighting we used daily values of meteodata. We calculated this
parameter for two stations: Klukhorsky Pereval (representative of the S stations) and Mineral'nye Vody (representative of N stations). The main period of calculation is 1966-2013 as reliable precipitation data
is available for this period only. We also tried calculation for the longer period using "unreliable" data
that led to the same result.
We found no significant correlation between the ice core $\delta^{18}O$ record and regional temperature, neither
with the reanalysis data, nor with the observation data, when using the whole period. A significant
correlation ($r = 0.44$, $p<0.05$) emerges for warm season data, when calculated for the period since 1984.
The slope for this period is 0.6 per mille per °C. We also repeated our linear correlation analysis using
precipitation weighted temperature, and obtained the same results. The precipitation weighted
temperature was calculated using daily meteorological data. We used data from two stations:
Klukhorskiy Pereval (as a representative of southern stations) and Mineralnye Vody (as a representative
of the northern stations). We didn't find any statistically significant correlations when compared 3-, 5-,
7-years running means of these parameters.
Line 721-723: "Our results are comparable to those obtained in the Alps by Mariani et al. (2014): again,
while the seasonal cycle of ice core δ18O appears related to that of temperature, this is not the case for
inter-annual variations, driven by other factors such as changes in moisture sources."
It does not seem that the current results are comparable. See conclusion in the cited paper:
"1. The seasonal cycle of temperature is well-captured in both the Alpine ice cores. On a seasonal scale
δ18O is thus a valid temperature proxy explaining ~60% of the signal.
2. On an annual scale the high variability of precipitation, especially at high-altitude sites, might
considerably bias the isotopic signal. For the glacier site with homogeneous distribution of precipitation
throughout the year the mean temperature signal is still partly preserved also on an annual scale. In the
other case with strong intraseasonal precipitation variability, the annual mean of δ18O was
representative only for temperature during precipitation and not for annual mean temperature."
We agree, that in (Mariani et al., 2014) the authors found strong link between temperature and δ18O on
seasonal cycle scale. While on annual scale the signal is biased by other factors. Though they report
correlation between δ18O and precipitation weighted temperature, this result is not useful for
palaeoclimatology. Citation: "For such a glacier site, a paleotemperature reconstruction is not feasible."
We added that this finding is a feature of annual variability of δ18O.
We found no persistent link between ice cores δ18O and temperature on interannual scale
Line 733-735: "The regression analysis showed significant negative correlation between the two
parameters. The regression equation for 11-year running means in the 1914-1928 and 1994-2013 differs
from the same for the 1929-1993 (see fig. 11 for the correlation plot and regression equations as well as
for the sliding window correlation plot).
Based on what criteria can these 2 periods (1914-1928/1994-2013 and 1929-1993) be separated? This
seems rather subjective. If looking at the entire period, the correlation would be much worse and the
negative slope would not be observed (i.e. both correlation and accordingly the negative slope would
not be significant; which is actually also not the case now considering the issue with the correlation analysis of smoothed data pointed out before). Using p weighted data and a different approach for seasonal separation of the d18O (both discussed before) might lead to completely different results anyhow. So please reconsider once the reevaluation is done.

We entirely removed this paragraph as well as fig.11

Line 735-737: "The 10-years sliding window correlation…"
Remove (see discussion of correlation analysis).

Removed

Line 943 - New (and old) Fig. 3: Why is there a winter and a summer missing around 31 m? Or should the winter around 33 m cover this entire section from around 31-34?

There was a piece of the core lost during the drilling operations. This part is covered by the bottom part of the 2004 core where the sampling resolution was 50 cm. It is evident that two seasons (one warm and one cold) are missing but we removed these values from the correlation analysis because of large uncertainty of the seasonal values calculations in this case.

The drilling problems are described in (Mikhalenko et al., 2015). The biggest gap appears at the depth 31.3 and 32.1 m. There was a piece of the core lost during the drilling operations. This part is covered by the bottom part of the 2004 core where the sampling resolution was 50 cm. It is evident that two seasons (one warm and one cold) are partially missing. We didn't use these values for the correlation analysis because of large uncertainty of the seasonal values calculations in this case.

Abstract - line 403 ff: "In the summer season the isotopic composition depends on the local temperature..."
..and conclusion line 802 ff: "This may explain the significant albeit non persistent correlation of summer δ18O and temperature."
According to the main text this is only true for a certain period (1984-2013)? Please be precise or reconsider the statement.

Reformulated according to the newer calculations.

In the warm season the isotopic composition depends on the local temperature but the correlation is not persistent in time.

Line 524-525 (& Fig. S2):
The overlap between the different cores does indeed look very good. Except for the lowermost 2-3 m of the 2013 core with the 2009 core (around 3-7 m depth in Fig. S2). Please comment.

We explain this with the different sampling resolution (5 cm for 2013 core and 15 cm for 2009 core), this explanation is in the text.

Line 612-613: "The average regional lapse rate was calculated using the available meteorological data. It is minimum (replace with "lowest") in winter (2.3°C per 1000 m) and maximum (replace with "highest") (5.2 °C per 1000 m) in summer (Fig. S3)."
Is this similar for N and S? Are these numbers and Fig S3 for N and S combined or only for one of the 2 regions (or only one station?)?

Comments added

The average regional lapse rate was calculated using the available meteorological data, we used the data from all of the stations for the calculation. The lapse rate is lowest in winter (2.3°C per 1000 m) and highest (5.2 °C per 1000 m) in summer (Fig. S3).

Line 678-680: "We note that the shallow ice core from the Maili plateau of Kazbek shows the same mean values of δ18O as the Elbrus ice cores during their overlap period. This is a surprise, given the difference in elevation (500 m) and continentality (200 km distance)."
Is this really that much of a surprise? The continentally should make the d18O at Kazbek more negative whereas the lower elevation should make it more positive. In the sum, the two factors seem to cancel out. Can you give some estimates about the size of those two effects and if a 0 sum is reasonable? For the altitude effect, see e.g. Mariani et al., 2014 and references therein.

We calculated continental gradient and lapse rate for δ18O using the data from the GNIP stations in the region that are situated at the lower elevations and in our opinion one should be very cautious when using this data for the high elevations ice cores study. The lapse rate is -0.25 ‰/100 m and continental gradient is -0.85 ‰ /100 km. The mean value of δ18O for Kazbek ice core should be 1.25‰ more positive because of elevation difference and 1.7‰ more negative due to continentality factor. We removed the surprise from the text.

This is a result of a mutual compensation of δ18O increase due to lower elevation position (Kazbek drilling site is 500 m lower) and of δ18O decrease because of continentality effect (Kazbek is 200 km further from the sea).

Line 774-777: "In order to explore the relationships of the Elbrus ice core datasets with the AMO, we used 20-year smoothed data."
I suggest removing this paragraph about AMO entirely. You do show it in Fig 9 and 10 and in some of the tables for comparison with the meteorological data. At this point it does not add anything but takes away from the main focus. Also, by using a 20 yr smoothed record the df is very low for the correlation analysis (<10, see earlier comment) and the result likely not significant anyhow.

Removed

Conclusion - Line 789-790: "We found no persistent link between ice cores δ18O and temperature,
common feature emerging from non-polar ice cores (e.g. Mariani et al., 2014)."
This is not consistent whit what has been found in the Mariani et al, 2014 paper: See conclusion therein:
"1. The seasonal cycle of temperature is well-captured in both the Alpine ice cores. On a seasonal scale
δ18O is thus a valid temperature proxy explaining ~60% of the signal.
2. On an annual scale the high variability of precipitation, especially at high-altitude sites, might
considerably bias the isotopic signal. For the glacier site with homogeneous distribution of precipitation
throughout the year the mean temperature signal is still partly preserved also on an annual scale. In the
other case with strong intraseasonal precipitation variability, the annual mean of δ18O was
representative only for temperature during precipitation and not for annual mean temperature."

We agree, that in (Mariani et al., 2014) the authors found strong link between temperature and δ18O on
seasonal cycle scale. While on annual scale the signal is biased by other factors. Though they report
correlation between δ18O and precipitation weighted temperature, this result is not useful for
palaeoclimatology. Citation: "For such a glacier site, a paleotemperature reconstruction is not feasible."
We added that this finding is a feature of annual variability of δ18O.

We found no persistent link between ice cores δ18O and temperature on interannual scale

Line 808-810: "The accumulation rate at the drilling site is highly correlated with the precipitation rate
and gives information about precipitation variability before the beginning of meteorological
observations."
In the current manuscript, the correlation is rather weak and should be changed to "…is significantly
correlated…". However, with the current issues this result might change.

Changed

…drilling site is significantly correlated with the precipitation …

Language:
…needs to be improved in general and the writing has to be more precise.

The language of the corrected version has been checked by a native-speaker

Find some (rather randomly chosen) examples below.
Abstract - Line 396-397: Here, we report on the results of the water stable isotope composition from
this ice core in comparison with results from shallow ice cores. The report is not about the comparison
between the ice core and the shallow cores (although the measurements at different labs and with
different methods have been compared and the cores have been overlapped). The important part is that these datasets are combined and then the results are compared with the meteorological data etc (see line 25-27). Please reconsider this statement and/or reformulate.

Reformulated

Here, we report on the results of the water stable isotope composition from this ice core with additional data from the shallow cores.

Line 398-399: Dating has been performed for the upper 126 m of the deep core combined with shallow cores data.
Also here this is unclear. The records from the deep and shallow cores were combined and dating then performed on this combined dataset down to the ice core depth of 126 m (i.e. combined depth 126 m + xy m from the shallow cores).

Reformulated combined with 20 m from the shallow cores.

Line 399:
The record covers 100 years but two centuries (21st and 20th century).

Reformulated

The record covers 100 years from 2013 back to 1914

Introduction - Line 431 ff: "The authors explored the links between the ice cores isotopic composition, local climate and large-scale circulation patterns. They found that in mountain regions isotopic composition of the ice cores governed both by the local meteorological conditions and by the regional and global factors. However, ice core records are complex. For instance, even in areas without any seasonal melt, accumulation is the net effect of precipitation, sublimation, and wind erosion processes, and may significantly differ from precipitation."
The "However" in the 3rd sentence is misleading because what follows is what has been observed and discussed in these papers.

[revised manuscript text omitted]

Отформатированная таблица

| Стр. 63: [2] Отформатировано | Anna | 18.01.2017 2:58:00 |
| --- | --- | --- |

Английский (США)

| Стр. 63: [3] Отформатировано | Anna | 18.01.2017 2:35:00 |
| --- | --- | --- |

Шрифт: не курсив

| Стр. 63: [3] Отформатировано | Anna | 18.01.2017 2:35:00 |
| --- | --- | --- |

Шрифт: не курсив

| Стр. 63: [4] Отформатировано | Anna | 20.01.2017 2:56:00 |
| --- | --- | --- |

Шрифт: полужирный

| Стр. 63: [4] Отформатировано | Anna | 20.01.2017 2:56:00 |
| --- | --- | --- |

Шрифт: полужирный

| Стр. 63: [5] Отформатировано | Anna | 20.01.2017 2:56:00 |
| --- | --- | --- |

Шрифт: Times New Roman, 10 пт, полужирный, Цвет шрифта: Авто, Английский (США)

| Стр. 63: [5] Отформатировано | Anna | 20.01.2017 2:56:00 |
| --- | --- | --- |

Шрифт: Times New Roman, 10 пт, полужирный, Цвет шрифта: Авто, Английский (США)

| Стр. 63: [5] Отформатировано | Anna | 20.01.2017 2:56:00 |
| --- | --- | --- |

Шрифт: Times New Roman, 10 пт, полужирный, Цвет шрифта: Авто, Английский (США)

| Стр. 63: [5] Отформатировано | Anna | 20.01.2017 2:56:00 |
| --- | --- | --- |

Шрифт: Times New Roman, 10 пт, полужирный, Цвет шрифта: Авто, Английский (США)

| Стр. 63: [6] Отформатировано | Anna | 20.01.2017 2:57:00 |
| --- | --- | --- |

Шрифт: Times New Roman, 10 пт, полужирный

| Стр. 63: [6] Отформатировано | Anna | 20.01.2017 2:57:00 |
| --- | --- | --- |

Шрифт: Times New Roman, 10 пт, полужирный

| Стр. 63: [6] Отформатировано | Anna | 20.01.2017 2:57:00 |
| --- | --- | --- |

Шрифт: Times New Roman, 10 пт, полужирный

| Стр. 63: [7] Отформатировано | Anna | 18.01.2017 2:31:00 |
| --- | --- | --- |

Английский (США)

| Стр. 63: [8] Отформатировано | Anna | 18.01.2017 2:35:00 |
| --- | --- | --- |

Шрифт: не курсив

| Стр. 63: [9] Отформатировано | Anna | 18.01.2017 2:37:00 |
| --- | --- | --- |

Шрифт: Times New Roman, 10 пт, Цвет шрифта: Авто, Английский (США)

| Стр. 63: [10] Отформатировано | Anna | 18.01.2017 2:31:00 |
| --- | --- | --- |

Английский (США)

| Стр. 63: [11] Отформатировано | Anna | 20.01.2017 2:56:00 |
| --- | --- | --- |

Шрифт: полужирный

| Стр. 63: [12] Отформатировано | Anna | 18.01.2017 2:35:00 |
| --- | --- | --- |

Шрифт: не курсив

| Стр. 63: [13] Отформатировано | Anna | 18.01.2017 2:41:00 |
| --- | --- | --- |

Шрифт: Times New Roman, 10 пт

| Стр. 63: [14] Отформатировано | Anna | 18.01.2017 2:35:00 |
| --- | --- | --- |

Шрифт: не курсив

| Стр. 63: [15] Отформатировано | Anna | 20.01.2017 2:56:00 |
| --- | --- | --- |

Шрифт: полужирный

| Стр. 63: [16] Отформатировано | Anna | 18.01.2017 2:35:00 |
| --- | --- | --- |

Шрифт: не курсив

| Стр. 63: [17] Отформатировано | Anna | 20.01.2017 2:56:00 |
| --- | --- | --- |

Шрифт: полужирный

| Стр. 63: [18] Отформатировано | Anna | 20.01.2017 2:57:00 |
| --- | --- | --- |

Шрифт: полужирный

| Стр. 63: [19] Отформатировано | Anna | 18.01.2017 2:41:00 |
| --- | --- | --- |

Шрифт: Times New Roman, 10 пт

| Стр. 63: [20] Отформатировано | Anna | 18.01.2017 2:35:00 |
| --- | --- | --- |

Шрифт: не курсив

| Стр. 63: [21] Отформатировано | Anna | 20.01.2017 2:57:00 |
| --- | --- | --- |

Шрифт: не полужирный

| Стр. 63: [21] Отформатировано | Anna | 20.01.2017 2:57:00 |
| --- | --- | --- |

Шрифт: не полужирный

| Стр. 63: [21] Отформатировано | Anna | 20.01.2017 2:57:00 |
| --- | --- | --- |

Шрифт: не полужирный

| Стр. 63: [22] Отформатировано | Anna | 20.01.2017 2:57:00 |
| --- | --- | --- |

Шрифт: не полужирный

| Стр. 63: [23] Отформатировано | Anna | 20.01.2017 2:57:00 |
| --- | --- | --- |

Шрифт: не полужирный

| Стр. 63: [23] Отформатировано | Anna | 20.01.2017 2:57:00 |
| --- | --- | --- |

Шрифт: не полужирный

| Стр. 63: [23] Отформатировано | Anna | 20.01.2017 2:57:00 |
| --- | --- | --- |

Шрифт: не полужирный

| Стр. 63: [24] Отформатировано | Anna | 20.01.2017 2:57:00 |
| --- | --- | --- |

Шрифт: не полужирный

| Стр. 63: [24] Отформатировано | Anna | 20.01.2017 2:57:00 |
| --- | --- | --- |

Шрифт: не полужирный

| Стр. 63: [24] Отформатировано | Anna | 20.01.2017 2:57:00 |
| --- | --- | --- |

Шрифт: не полужирный

| Стр. 63: [25] Отформатировано | Anna | 20.01.2017 2:57:00 |
| --- | --- | --- |

Шрифт: не полужирный

| Стр. 63: [25] Отформатировано | Anna | 20.01.2017 2:57:00 |
| --- | --- | --- |

Шрифт: не полужирный

| Стр. 63: [25] Отформатировано | Anna | 20.01.2017 2:57:00 |
| --- | --- | --- |

Шрифт: не полужирный

| Стр. 63: [26] Отформатировано | Anna | 20.01.2017 2:57:00 |
| --- | --- | --- |

Шрифт: полужирный

| Стр. 63: [27] Отформатировано | Anna | 20.01.2017 2:57:00 |
| --- | --- | --- |

Шрифт: не полужирный

| Стр. 63: [27] Отформатировано | Anna | 20.01.2017 2:57:00 |
| --- | --- | --- |

Шрифт: не полужирный

| Стр. 63: [28] Отформатировано | Anna | 20.01.2017 2:57:00 |
| --- | --- | --- |

Шрифт: не полужирный

| Стр. 63: [29] Отформатировано | Anna | 20.01.2017 2:57:00 |
|---|---|---|

Шрифт: не полужирный

| Стр. 63: [30] Отформатировано | Anna | 20.01.2017 2:57:00 |
|---|---|---|

Шрифт: не полужирный

| Стр. 63: [31] Отформатировано | Anna | 20.01.2017 2:57:00 |
|---|---|---|

Шрифт: не полужирный

| Стр. 63: [31] Отформатировано | Anna | 20.01.2017 2:57:00 |
|---|---|---|

Шрифт: не полужирный

| Стр. 63: [32] Отформатировано | Anna | 20.01.2017 2:58:00 |
|---|---|---|

Шрифт: не полужирный

| Стр. 63: [32] Отформатировано | Anna | 20.01.2017 2:58:00 |
|---|---|---|

Шрифт: не полужирный

| Стр. 63: [33] Отформатировано | Anna | 20.01.2017 2:58:00 |
|---|---|---|

Шрифт: не полужирный

| Стр. 63: [33] Отформатировано | Anna | 20.01.2017 2:58:00 |
|---|---|---|

Шрифт: не полужирный

| Стр. 63: [34] Отформатировано | Anna | 20.01.2017 2:58:00 |
|---|---|---|

Шрифт: не полужирный

| Стр. 63: [34] Отформатировано | Anna | 20.01.2017 2:58:00 |
|---|---|---|

Шрифт: не полужирный

| Стр. 63: [35] Отформатировано | Anna | 20.01.2017 2:58:00 |
|---|---|---|

Шрифт: не полужирный

| Стр. 63: [35] Отформатировано | Anna | 20.01.2017 2:58:00 |
|---|---|---|

Шрифт: не полужирный

| Стр. 63: [36] Отформатировано | Anna | 20.01.2017 2:58:00 |
|---|---|---|

Шрифт: не полужирный

| Стр. 63: [36] Отформатировано | Anna | 20.01.2017 2:58:00 |
|---|---|---|

Шрифт: не полужирный

| Стр. 63: [37] Отформатировано | Anna | 20.01.2017 2:58:00 |
|---|---|---|

Шрифт: не полужирный

| Стр. 63: [37] Отформатировано | Anna | 20.01.2017 2:58:00 |
|---|---|---|

Шрифт: не полужирный

| Стр. 63: [37] Отформатировано | Anna | 20.01.2017 2:58:00 |
|---|---|---|

Шрифт: не полужирный

| Стр. 63: [38] Отформатировано | Anna | 20.01.2017 2:58:00 |
|---|---|---|

Шрифт: не полужирный

| Стр. 63: [38] Отформатировано | Anna | 20.01.2017 2:58:00 |
|---|---|---|

Шрифт: не полужирный

| Стр. 63: [39] Отформатировано | Anna | 20.01.2017 2:58:00 |
|---|---|---|

Шрифт: не полужирный

---

## Author Response (AR3)

We would like to thank the reviewers for the thorough reviews and detailed comments. Here we provide answers to the questions raised in the reviews.

Comments of the reviewers are in blue, our answers are in green, and corrections in the paper text are in black.

**Reviewer 1.**

This is a review of the 2$^{nd}$ revision of this study by Kozachek et al., and I was also reviewer of the original version and the 1$^{st}$ version. I appreciate the authors efforts into investigation the seasonal divisions of the ice core data and the manuscript has generally improved a lot, although I still struggle a bit to understand how exactly the seasons are defined in the new version. However, it is clear that the variability changes a lot depending on the definition of the seasons and that the new definitions have greater decadal variability. If the author address the (minor) points in the comments below I think the manuscript could be accepted for publication.

Detailed comments:

L22 "allowed" changed to "allow"

Changed

There is a distinct seasonal cycle of the isotopic composition which allow dating by annual layer counting

L23 Remove extra punctuation mark after "shallow cores"

Removed with additional data from the shallow cores.

L78-79 The potential for reconstructing the NAO using Greenland ice cores was suggested by Vinther et al. (2003) rather than proven. They show that the relation between the NAO and the main variability of the Greenland d18O from ice cores is not stable.

Correected

The strong influence of the NAO pattern on the Greenland ice cores isotopic composition has been discovered and the possibility to use the ice cores data for the past NAO changes reconstruction was suggested

L158 "18" should be superscript in d18O. Check for other instances. For example in L172, L332, L333 and L416

Corrected
L171-172 I don't understand what you mean here: We changed the borders when needed in order to
avoid ascribing minimum of d18O to the warm season and maximum to the cold season". In figure 4 of
Vinther et al. (2010) warm and cold seasons is defined similarly as in your Figure 3? Vinther et al
performed a correlation analysis to define the extent of the seasons. How did you do this? How much
accumulation was assigned to each season?
There is a slight difference between fig. 4 of Vinther et al. (2010) and our fig. 3. We basically used the
same approach as there is an obvious seasonal cycle of $\delta18O$ which is coherent with the seasonal cycle
of temperature in the region. We therefore assume that the maximum value of $\delta18O$ in the annual cycle
corresponds to July and the minimum value corresponds to January and put the boundary so that these
extreme values are in the middle of a season. However, there were several situations when this approach
could potentially lead to assign minimum values to summer and maximum to winter. In order to avoid
this problem we used the middle point between minimum and maximum as a border between seasons in
such cases.
The amount of accumulation assigned to each season varies depending on the accumulation rate in-
between minimum and maximum values of the annual cycle.
We basically used the same approach as there is an obvious seasonal cycle of $\delta18O$ which is coherent
with the seasonal cycle of temperature in the region. We therefore assume that the maximum value of
$\delta18O$ in the annual cycle corresponds to July and the minimum value corresponds to January and put
the boundary so that these extreme values are in the middle of a season. However, there were several
situations (six for the whole ice core record) when this approach could potentially lead to assign
minimum values to summer and maximum to winter. In order to avoid this problem we used the middle
point between minimum and maximum as a border between seasons in such cases.
L173 "We stacked…" you stacked the ice core data? And you assign the maximum and minimum to the
center of the warm and cold seasons, respectively? This sentence is not clear.
It was a typo. We reformulated the paragraph and this sentence was removed.
L177-178 What is the motivation for this definition of warm and cold seasons?
The motivation was based on the coherence between seasonal cycle of $\delta18O$ and air temperature. The
details are given above
L183 "We didn't…" the use of contractions is in general too informal for academic writing. Check for
other instances. For example in L372
Corrected

L228 Concider using the PC-based NAO index, although it doesn't make much of a difference during
winter.
We did this and got almost the same result. We used PC-based NAO from National Center for
Atmospheric Research Staff (Eds). Last modified 03 Mar 2017. "The Climate Data Guide: Hurrell
North    Atlantic    Oscillation    (NAO)    Index    (PC-based)."    Retrieved    from
https://climatedataguide.ucar.edu/climate-data/hurrell-north-atlantic-oscillation-nao-index-pc-based.
The graph are shown here in the fig. A1 for the warm season, and A2 for the cold season.

[Figure]

Fig. A1. Comparison of the δ18O and PC-based NAO index in the warm season

[Figure]

Fig. A2. The same as A1 but for the cold season.

L264-265 The definition of summer and winter is generally done using temperature, or one might talk of dry and wet seasons of summer and winter doesn't exist. Think of southern versus northern hemisphere. Use months to define the seasons in relation to temperature.

Agree. We use terms "warm season" and "cold season" for the ice core data. In the other cases we use months to define the seasons.

The lapse rate is lowest in December-February (2.3°C per 1000 m) and highest (5.2 °C per 1000 m) in June-August (Fig. S3).

L420 Remove punctuation mark.

Removed

**Reviewer 2.**
**General**
A 100 year record of water stable isotopes and accumulation derived from the combination of multiple alpine shallow cores and a deep ice core collected at Mt. Elbrus in the Caucasus is presented. The high annual net accumulation rate at the site allows for high temporal resolution and a seasonally resolved data set. Meteorological data, reanalysis temperatures, GNIP isotope data and isotope modeling results as well as atmospheric circulation indices are used to investigate the regional climate and to investigate the parameters recorded by the ice core. The study concludes that for the ice core site the isotopic composition in the warm season is related to local temperature for certain time periods whereas in the cold season the atmospheric circulation is the main driver of modulation. The accumulation data is used to derive a reconstructed precipitation record for the Caucasus highlands for the time period prior to reliable observations.

The successful drilling and subsequent analysis of the presented ice core is already an impressive achievement on its own. The drilling location is characterized by limited surface melt and the ice cores and analysis performed are of high quality. The presented records with clear seasonal variations are certainly useful to gain further insight into the past climate and atmospheric conditions in the studied region which lacks of high-elevation meteorological data. In the current version of the manuscript, most of the issues raised previously in the review process were addressed and implemented. In particular, the approach to split the data into seasonal values is now much more convincing. Still, some issues remain which need more careful investigation and discussion. Addressed later on in more detail, this concerns in particular a) the lack of discussion regarding the dating uncertainty and its effect on the performed correlation analysis, b) the different conclusions drawn for the relation between T and precipitation and their respective ice core proxies which might be caused simply by the different length of the available time series of meteorological data and c) the choice in this version to use the altitude adjusted T (lapse rate corrected station data) for the correlation analysis which might have resulted due to a misunderstanding of a previous review request. Further, some of the figures presented in this version contain serious mistakes which also may or may not have occurred when performing the statistical analysis. This potentially may be a very serious issue and in any case is certainly very unfortunate to happen at this stage of review. Considering the above points, the interpretation and final conclusions drawn by the authors cannot be convincing. Also, the language still needs further improvement, which however is a minor issue.

Taking into consideration all the excellent work and big efforts already undertaken to receive the presented data, it would be a pity to reject this study for publication despite the still existing flaws. I therefore suggest once again major revisions but at the same time would like to urge the authors to invest additional effort and time to carefully reconsider their analysis and interpretation, also being open for potentially different final conclusions even when requiring rewriting substantial sections of the manuscript.

**More detailed major comments:**
Line numbering refers to the current revised version (version 4 I think).
**2.1.4 Dating:**
Lines 161-162: What is the estimated dating uncertainty at the bottom of the presented record?

We discuss the dating uncertainty below

The depth given here as 126 m is confusing because in fact 1914-2013 is contained in the 15 m covered by the shallow cores plus the 126 m covered by the deep core, thus around 140 m in total.

Line 164: Also here, 1914-2013 is not contained in 126 m. Please reformulate accordingly, e.g. "…which corresponds to the total of 140 m presented in this study (the 15 m covered by the shallow cores plus the 126 m covered by the deep ice core.".

Accordingly, please reconsider formulation also elsewhere in the manuscript.

Reformulated

Hereafter, we focus our analysis on one hundred years, from 1914 till 2013, which corresponds to the total of 140 m of ice thickness studied here (the 15 m covered by the shallow cores plus the 126 m covered by the deep ice core

Line 164-165: The formulation regarding the dating uncertainty ("…relatively small...") is extremely vague. Please indicate a number for the estimated dating uncertainty.

The number has been indicated. The dating uncertainties are discussed in more details below.

This period has been chosen because of relatively small dating uncertainty (±2 years)

Line 166: Reformulate to "In the bottom part of the core the cycles in the isotopic composition are less prominent and dating becomes less reliable leading to a significant increase in uncertainty."

Reformulated

In the bottom part of the core the cycles in the isotopic composition are less prominent and dating becomes less reliable leading to a significant increase in uncertainty.

The threshold of exactly 126 m seems arbitrary. I assume the uncertainty already increased above compared to the top let's say 50 m or so. So the estimated dating uncertainty should definitely be indicated as a number somewhere (also see later comments).

We took the threshold of 100 years for the discussion. Yes, the depth is arbitrary; we could have discussed 102 or 98 years with the same uncertainty. However, a round figure of 100 years seems more beautiful for the discussion.

Line 173: I do not understand what you mean by "stacked"? It is used later on in line 328 where it refers to the overlap of the various cores. This does not seem to be the same thing since here this refers to the entire record of which most is covered by the deep core only. Please explain and clarify accordingly in the manuscript.

In line 173 it was a typo. We meant sticked. We reformulated this paragraph following the comments of the reviewer 1.

Line 182 and Figure 3: To me it is not evident at all that "two seasons (one warm and one cold) are partially missing". If so, this would be a year with an exceptional low accumulation. So this certainly is one year of dating uncertainty.

We agree this gap can cause age scale uncertainty. The missing 75 cm of the core are not the sum of two seasons. This sum is higher (it is about 1 m actually) which leads to the layer thickness of 0.5 meter per season. This value is close to the average. In the opposite situation, if we consider that the whole gap corresponds to one winter, then we get a winter with extremely high accumulation rate. However, for the estimation of the dating uncertainties we used the absolute age markers. These markers are tritium peak in 1963 and sulfate peak in 1912 (this year is not discussed in this paper still we know its depth) which corresponds to Katmai eruption. The uncertainty was calculated as the difference of age estimation using different methods at these dates. The maximum difference was 2 years.

Also Line 183: It is unclear what you mean by "we did not use these values for the correlation analysis"? If you have a gap, i.e. not data/value of course you cannot use it. Do you mean that for this missing year xy (if it really is one year considering the then very low accumulation...) you also did not include a value for the meteorological data? This seems trivial and I just hope you did not shift the age scale of the two records against each other when performing the analysis... Please re-check carefully.

Of course we didn't shift the age scales. We show on the figures the value obtained from the 2004 ice core where the sampling resolution was 50 cm and these values were not used for the correlation analysis because their low reliability.

Figure 3: Again, regarding the dating uncertainty to me it seems questionable if the minima in both d18O and Ammonium is really occurring in summer (double peak) or if this does not rather indicated another winter minima (with a rather high winter d18O). I think this is rather challenging to judge and cannot be decided without some uncertainty. The point is that this should probably be assigned with another year of dating uncertainty. Together with the gap, this would make ± 2 years of uncertainty just for this section shown in Fig. 3. So the total uncertainty for the year 1914 (including the dating of > 90 additional annual layers) is very likely much higher than ±2 years.

The comparison of different dating methods on age control points shows that the overall error of our timescale at these two depth levels does not exceed ±2 years which means that independent dating uncertainties discussed by the reviewer should compensate each other at this points

For the estimation of the dating uncertainties we used the absolute age markers. These markers are the tritium peak in 1963 and the sulfate peak in 1912 which corresponds to the Katmai eruption (Mikhalenko et al., 2015). The comparison of different dating methods on age control points shows that the overall error of our timescale at these two depth levels does not exceed ±2 years which means that independent dating uncertainties should compensate each other at this points

Line 329 and Fig. 2S: The inter-core disagreement is indeed small. However, there is at least half a year of disagreement between the very bottom of the 2013 shallow core and the 2009 deep core (around 5 m depth in Fig. S2). This indicates that even in the top seven years (2007-2013) with 2 available absolute time markers (the drilling date of the 2012 and 2009 cores) there exists uncertainty in the dating. For the 93 years before with no absolute time markers available, the dating uncertainty will certainly be quite substantial and will definitely affect the correlation analysis particularly on an annual or seasonal scale. So when discussing the correlations found in Section 3.3. this should be addressed more carefully (a first step has been made by including 3, 5 and 7 yr running averages to the analysis).

We disagree with the point that "…the 93 years before with no absolute time markers available" as there are two markers corresponding to 1912 and 1963 that are described in (Mikhalenko et al., 2015). The question of the correlation analysis is addressed below.

Line 187 / Figures 5, 6 and also 8, 9 and 10:
I do not see a gap there for the missing year (or season) you discuss for Figure 3?
Please correct.

We used data from 2004 ice core. However this value is less reliable than the values for the neighboring years because of 50-cm sampling resolution in this part of the core. We excluded it from the correlation analysis but show it on pictures for the uniformity.

3.1 Regional climate:
Lines 260-263 and line 270:
According to the comment made in the previous revision it would be helpful to show the precipitation data for all the stations discussed (lines 260-263). As written in line 270 the authors intended to follow this suggestion but it seems they unfortunately have forgotten to actually include all the data in Fig. S4. Please add.

We used stations Klukhorskiy Pereval as a representative of the Southern stations and Mineralnye Vody as a representative of the Northern stations. We include several other stations to the fig. S4. For the data from another stations in the region the reviewer is referred to Shahgedanova et al. (2014), Shahgedanova et al. (2007), Tielidze (2016) and references therein.

Line 269: It is unclear how the temperature for the drill site was calculated based on the determined lapse rate? Was the seasonal cycle in the lapse rate considered? Please clarify in the text.

Yes, the seasonal cycle of the lapse rate was taken into account. We added this point to the text.

Based on the lapse rate we calculated temperature at the drilling site taking into account its seasonal variability shown on fig. S3.

Also, the authors followed the suggestions made in the previous review regarding the loss of
information (namely the d18O/T relation) when only showing normalized T data. Unfortunately it
seems a misunderstanding occurred. The reviewer's idea was this lapse rate adjusted temperature ("drill
site T") to be used to determine the d18O/T relationship (i.e. in a way the calibration of d18O as a proxy
for temperature). Whereas the correction for the lapse rate is a necessary step to do so, it is not required
for the correlation analysis. This is where the misunderstanding happened. The authors now also used
this adjusted T data for the correlation analysis (and accordingly also in the figures 8, 9 and 10). This
was not suggested! In fact it does not make sense for the following reasons: Because the determined
lapse rate certainly comes with an uncertainty (also a change in the rate over time cannot be excluded),
an additional source of uncertainty will be introduced to the data set. This will bias the correlation
analysis.
To include such a bias is unnecessary because the d18O recorded in the ice core also reflects processes
taking place on a larger regional scale such as evaporation temperature in the moisture source region,
re-evaporation processes etc. and therefore a regional T (i.e. the station average) is likely most
representative for the potential T proxy recorded in the ice core (i.e. d18O). This is a different matter for
the precipitation data for which the closest/high altitude stations are most relevant and the authors
decision to only use those is a reasonable choice (precipitation and as well as accumulation may vary
significantly within regional scale because of orography/altitude effects etc.).
In summary, for the correlation analysis the authors should absolutely stick to the averaged T including
all stations as in the previous version (i.e. divided into N and S). It thereby does not matter if they are
normalized or simply averaged as for the correlation the results will be the same. For the figures, I
suggest to not show the normalized data.
Now we use the normalized values for the correlation again. The normalization was used in order to
avoid introducing of the errors. The stations are situated at the different altitude levels. Consequently,
despite the same tendencies in the temperature changes they are characterized with the different
absolute values of temperature. For example, if one year of observations is missing at the coldest
station, the simple average will be higher. If we use the normalized values for construction of the
regional temperature record we do not introduce these errors.
3.2 Ice core records:
Line 332-334: In the authors response you wrote: "We calculated continental gradient and lapse rate for
δ18O using the data from the GNIP stations in the region that are situated at the lower elevations and in
our opinion one should be very cautious when using this data for the high elevations ice cores study.
The lapse rate is -0.25 ‰/100 m and continental gradient is -0.85 ‰ /100 km. The mean value of δ18O
for Kazbek ice core should be 1.25‰ more positive because of elevation difference and 1.7‰ more
negative due to continentality factor."
I think the fact that these calculated effects actually match up with what is observed in the two ice cores
is a very nice and interesting result. Please include this more detailed description and results given in
the above answer to the manuscript.
Added

This is a result of a mutual compensation of δ18O increase due to lower elevation position (Kazbek
drilling site is 500 m lower) and of δ18O decrease because of continentality effect (Kazbek is 200 km
further from the sea). We calculated continental gradient and lapse rate for δ18O using the data from the
GNIP stations in the region that are situated at the lower elevations. The lapse rate is -0.25 ‰/100 m
and continental gradient is -0.85 ‰ /100 km. The mean value of δ18O for Kazbek ice core should be
1.25‰ more positive because of elevation difference and 1.7‰ more negative due to continentality
factor.
3.3 Comparison of ice core records with regional meteorological data:
Line 363-385 and Fig. 9 and 10: In those figures the meteorological temperature data is shifted on the
age scale by around 42 years! Shown is 1870-1970 instead of 1910-2013, see the combined figure
created form the manuscript figs 8 & 9 and 8 & 10 included on the next page. The same mistake may
have occurred when performing the statistical analysis (correlations). Please correct and check
carefully!
We used the correct dataset for the correlation. Unfortunately the error occurred for the graph.
Following several reviewer's comments we changed fig. 8, 9, and 10.
Line 386-390: This is not very convincing. The problem is that you draw different conclusions for T-
d18O and Precipitation-Accumulation relation which might only be caused by the difference in length
of the available meteorological time series. In other words, a reasonable correlation was also found for
warm season T with d18O for the younger part of the record. Still, the correlation is lost in the older
section. How can you exclude the exact same thing is true for the precipitation data?
Unfortunately this problem cannot be resolved with the available data. However, it is easy to imagine,
for instance, that change of the moisture source lead to change of the precipitation isotopic composition
at the same air temperature. Thus the correlation between T and d18O is unstable in time. It is much less
probable that the correlation between accumulation rate and precipitation rate varies in time.
Also, layer thickness is not equal net accumulation! If precipitation is reconstructed from ice core
derived accumulation data, one needs to account for layer thinning (Cuffey and Paterson, 2010). See for
example in Mariani et al., 2014 ("The reconstructed net accumulation can be regarded as precipitation
proxy, considering few caveats. (i) In order to account for thinning effects, such reconstructions require
an accurate description of the glacier ice flow by means of physical models.").
Therefore, please address following the literature (e.g. Schwerzmann et al., 2006; Herren et al., 2013 or
probably easiest Equations 1 both in Henderson et al, 2006 and Mariani et al., 2014 which is based on
the Nye model).
We used the Dansgaard-Johnsen model for the correction of the layer thickness or the layer thinning. It
is pointed in the text (line 190 of version 4). We also added this to the discussion of the accumulation
section.

The seasonal accumulation rate is seasonal layer thickness corrected for densification using the density
profile from Mikhalenko et al. (2015) and for the layer thinning due to glacier flow using the Nye model
(Nye, 1963; Dansgaard and Johnsen, 1969). It is linked to the precipitation rate…
Line 376-378 and lines 432-433: The conclusions and results of Mariani et al., 2014 are still not stated
correctly.
In their response to the previous review the authors stated: We agree, that in (Mariani et al., 2014) the
authors found strong link between temperature and δ18O on seasonal cycle scale. While on annual scale
the signal is biased by other factors. Though they report correlation between δ18O and precipitation
weighted temperature, this result is not useful for palaeoclimatology. Citation: "For such a glacier site, a
paleotemperature reconstruction is not feasible."
When re-reading the study in question, the authors will realize that 2 separate ice cores are discussed
therein: "We assume that at the Grenzgletscher the non-uniform snow deposition throughout the year is
more pronounced than at Fiescherhorn (see Section 3.2.1), as it is generally the case in the Southern
Alps compared to the Northern Alps (Frei and Schär, 1998; Eichler et al., 2004; Sodemann and Zubler,
2009)."
Obviously, those 2 ice cores are located in meteorologically significantly different regions. So whereas
your above statement about the annual scale and the need for precipitation weighting is true for
Grenzgletscher it is different for Fiescherhorn (no p weighting was necessary and performed for this
core).
Because for the ice core in your study where you point out the relatively equal distribution of
precipitation between the seasons, the conclusions/results you should cite are the ones related to
Fiescherhorn. Accordingly the results/conclusion from Mariani et al. which you should consider are:
- 3.1.2 Annual scale: "The annual Fiescherhorn δ18O correlates significantly with the Jungfraujoch
annual temperature (r=0.44, p<0.01, period 1961-2001). The resulting slope is (0.50±0.16)‰/°C which
is consistent with the result based on the seasonal values."
- Conclusions: "For a glacier site with homogeneously preserved accumulation throughout the year the
mean temperature signal is partly preserved on annual scale." The difference of your finding should be
stated accordingly (or as it might change considering the previous comment it might turn out to still be
the agreement). So please reformulate.
Reformulated
Our results are comparable to those obtained in the Alps by Mariani et al. (2014) for the Fiescherhorn
glacier where the authors found significant though weak correlation between temperature and δ18O.
However for the Elbrus ice core this correlation was found in the warm season only.
**Concerns regarding final results and conclusion.**
Out of curiosity after compiling the two figures shown further above, I created the two additional
Figures A and B shown below. In this case, I adjusted the scales against each other the way they should be (the aforementioned 42 year shift). Also, following the comment made earlier (see 3.1 Regional climate – Line 269), I here used the earlier version of Fig. 8 (more precisely the normalized T data for the cold and warm period respectively). Assuming reasonable uncertainty in the dating (see comments regarding the dating above) I allowed the age scale of the T data to stretch until reaching a best fit (determined visually due to lack of the actual data which is admittedly a very crude method). For better visibility the d18O records from Fig. 9 and 10, respectively were copied, the background removed and these curves were directly overlaid with the according normalized T from the earlier version Fig. 8 (either warm or cold season T data). For a shift of 8 years, which does not seem unreasonable considering the potential dating uncertainty (i.e. an offset in dating by 8 years out of 100), a strikingly good agreement between T and d18O can visually be seen, particularly for the cold season for which some very characteristic features in the T record can also be found in the d18O record, e.g. between around 1908 and 1935 or 1990 to 2013 with ages referring to the T age scale (upper scale) (Figure A). Also the overall trend is in close agreement except maybe between a short period around 1965 -1970 (Figure A). For the warm season also some characteristic features in both T and d18O exist for the period around 1908-around 1930 and from around 2000-2013 (Figure B), although not as closely related as for the cold season. Also the trend for the warm season does not seem to agree as well as for the cold season. For the location and setting of the ice core site, a reasonable explanation why d18O might be more closely related to T during the cold season than during the warm season could be that in the cold season re-evaporation processes are reduced and transport from the source region is more direct (see manuscript supplement Fig. S1).

In any case, these findings completely disagree with the results and conclusions of the reviewed manuscript although it is based on the exact same data figures:

See for example line 28-30: "In the warm season (May - October) the isotopic composition depends on the local temperature, but the correlation is not persistent in time, while in cold season (November – April), the atmospheric circulation is the predominant driver of the ice core isotopic composition. "

Also line 367-368: "A significant correlation (r = 0.44, p<0.05) emerges for warm season data, when calculated for the period since 1984.", "line 372-373: We didn't find any statistically significant correlations when compared 3-, 5-, 7-years running means of these parameters."

or line 432: "We found no persistent link between ice cores δ18O and temperature on interannual scale…".

Even for the visually good agreement between T and d18O (see Figure A top panel), I would not expect a very high correlation because as stated somewhere earlier, even a 1 year offset can potentially destroy any correlation. However, on a multi-annual scale (3, 5 or 7 year running means) I would expect the correlation to be high. I thus strongly suggest to carefully revisiting your dating and subsequent data analysis, evaluation and interpretation of your results. Previous publication of the dating can certainly not be a justification for not reconsidering. The potential finding that d18O does indeed reflect T and thus could be used as a T proxy would make this ice core archive certainly much more valuable.

We thank the reviewer for the huge effort put into looking on the age scale of our record. However we stay strong by our dating. There are several reasons for this position. We use the chronology elaborated for this ice core and described in (Mikhalenko et al., 2015). This chronology based on the count of annual cycles in isotopic composition and ammonium concentration. Also there are two absolute age markers: tritium peak in 1963 and sulfate peak in 1912 (this year is not discussed in this paper still we know its depth) which corresponds to Katmai eruption. This time scale is also confirmed by the ice flow modeling. We do not have any reason for the change of the dating.

However, following the advice of the reviewer we have made changes in the text that allow more flexibility in interpreting the obtained data, as the reviewer requests.

Obviously, the above inferences strongly depend on the uncertainties of the timescale used. If one concedes that the error of the timescale could be significantly greater than ±2 year, quite different conclusions may be reached by adjusting the scale of the $\delta^{18}O$ and T records against each other. For instance, by contracting the $\delta^{18}O$ record by 8 years with respect to the initial timescale in Figs 9 and 10, one would find much better correlation between $\delta^{18}O$ and temperature, thus reaching the conclusion that the local temperature is the main driver of the $\delta^{18}O$ variability. However, based on various experimental evidences, as discussed in the dating section, we argue that the timescale developed for the Elbrus ice core is accurate within ±2 years.

Therefore, the most realistic conclusion of those that can be drawn from the data obtained is that the temperature is weakly correlated with the $\delta^{18}O$, and that this correlation is unstable in time.

As mentioned in the paper there is no correlation between running means of d18O and temperature. We include the figures showing the running means of δ18O and temperature here (fig. A3 and A4). There are some common features in the recent period in the warm season that is discussed in the paper. For the other periods no correlation found. As was pointed by the reviewer in the previous review, the correlation based on running means is insignificant because of lower number of degrees of freedom, we do not include these figures to the paper.

[Figure]

Fig. A3. The running means (3-, 5-, 7-years) of isotopic composition and regional temperature in the
warm season. Thin line represents the raw data. The thickest and darkest lines represent 7-years running
means

[Figure]

A4. The same as A3 but for the cold season.

**Minor comments:**
**Table 3 and 4:**
Some significant correlations are not in bold.

Corrected

**Language (due to lack of time just one example for one of the newly written sections):**

Line 170: …we used *a* slightly…

Line 172: …ascribing *minima* in….and *maxima*…

Line 174: ……using *the* criteria…described *by*…

Line 177: …using the *seasonal signal in the isotopic composition*…

Line 177-178: *For the* meteorological data we *selected the period* from November to April for the cold season and *the* period…

Line 179: There *are* some gaps…

The language was checked by the native speaker

[revised manuscript text omitted]

---

## Author Response (AR4)

We thank the reviewer for the huge effort dedicated to the improvement of our paper. We address the
detailed comments below. The comments of the reviewer are in blue, the answers are in green and the
relevant changes in the text are in black.
This is the 4th review of the present study by Kozachek et al.: Large-scale drivers of Caucasus climate
variability in meteorological records and Mt Elbrus ice cores. I also reviewed the 2nd and 3rd version.
The manuscript has constantly improved with each version. I appreciate the efforts by the authors to
address the issues raised in the previous reviews. By now, almost all of these points are convincingly
considered. Still, the splitting of the data into seasons seems not quite settled yet, a concern also raised
by another reviewer. Further, there are a few explanations provided in the authors' response which did
not find their way into the manuscript text although they seem important for clarity. Unfortunately,
again the wrong dataset is plotted in one of the figure panels. Although such repeated, basic oversights
do not help, I take the authors by their word, trusting that no comparable mistakes happened when
evaluating the datasets. The language has improved significantly between the previous and this version,
but nevertheless I encourage the authors to take advantage of the language editing offered by the
journal. In conclusion, if the (generally) minor points in the comments below can be addressed, I think
the manuscript could be accepted for publication.
**Detailed comments:**
**L178-181** This is still unclear. If the maximum values are always assigned to July and the minimum
values are always assigned to January, how can in some occasions minimum values suddenly become
assigned to summer and maximum values to winter? This makes no sense. The approach described
above is similar to the one in Vinther et al. (2010) and the underlying assumption for such an approach
(max=summer, min=winter) is 50% winter and summer accumulation. The boundary between summer
and winter is then defined by the middle between these two extreme (depth scale in m w.e.). This
approach to split the record into seasonal data (cold, warm) allows comparison with the meteorological
data separated into the seasons in the way described in the manuscript. However, the way I understand
the approach described in the present version it seems the minima and maxima are always assigned to
the middle of the respective season (cold and warm, respectively). But still I cannot imagine why
minima/maxima suddenly should become assigned to winter/summer. In any case the described
approach (or my interpretation of its description) contains two additional assumptions: (1) the month of
lowest (highest) T is always in the mid-season and (2) this has not changed over the investigated period.
Both points and their potential consequences for the analysis due to the fact that both introduce
additional uncertainty should in this case be discussed. With the station T data at hand it is easily
possible to investigate these two assumptions. Therefore, e.g. plot the months with lowest/highest T
against time. The results should show the months of most extreme temperatures to (1) lie in the middle
of the respective season and (2) the months when these minima/maxima were observed did not change
over time. Actually (1) has been shown to be valid by the data presented in manuscript Fig. 4. Please
explain your approach accordingly or adjust your methodology following the description in Vinther et al. (2010). The subsequent evaluation of the data (correlation analysis etc.) and its interpretation should
then also be revised.
Yes, these two assumptions were used and they are confirmed by the weather observations. The middle
of the warm season is the end of July – beginning of August. During the whole period of observation
the maximum temperature was observed outside this period in 1969 only, when the maximum
temperature was in June. In the cold season the middle of the season is the end of January – beginning
of February. The minimum values of temperature were observed outside this period in 1971, 1985,
1995, and 1997. We therefore consider this assumption being valid for the whole period of time
discussed in the paper.
In some occasions minimum values could become assigned to summer and maximum values to winter.
That happened in the cases when minima and maxima were close to each other but the previous extreme
was far away. Thus, it was impossible to keep the methodology as in these cases the extreme value was
obviously not at the middle of the season. It worth being noticed that these years are not the years listed
in the previous paragraph. As these cases were observed just six times over the whole period
investigated, we removed this additional explanation from the text to avoid confusion.
We assume that the maximum value of $\delta18O$ in the annual cycle corresponds to July and the minimum
value corresponds to January and put the border so that these extreme values are in the middle of a
season. This method is based on two assumptions. Firstly, the months of the most extreme temperature
lie in the middle of the corresponding season. Secondly, the validity of the first assumption does not
change over time. Both assumptions are confirmed with the weather observations in the region. The
middle of the warm season is the end of July-beginning of August. During the whole period of
observation the maximum temperature was observed outside this period in 1969 only, when the
maximum temperature was in June. In the cold season the middle of the season is the end of January–
beginning of February. The minimum values of temperature were observed outside this period in 1971,
1985, 1995, and 1997. We therefore consider the first assumption being valid for the whole period of
time discussed in the paper. We also used ammonium concentration as an independent marker, using
criteria described on (Mikhalenko et al., 2015).
**L168-170** With the additional information about the two absolute time markers (1963 and 1912) now
provided in L163-164, a lot about the dating was clarified including the reasoning behind the selection
of the 100 year period. Clearly, this period was not selected because of the beauty of the number 100 as
suggested in the authors' response, but rather because at that depth the age scale is well defined by the
1912 time horizon. I thus suggest, adding this relevant information to the manuscript along the lines:
"This period has been chosen because at this depth, the age scale is well defined by the time horizon
found slightly below (Katmai 1912) resulting in a relatively small dating uncertainty of ±2 years, and
because of the availability of other records such as local meteorological observations."
Added

This period has been chosen because at this depth, the age scale is well defined by the time horizon
found slightly below (Katmai 1912) resulting in a relatively small dating uncertainty of ±2 years, and
because of the availability of other records such as local meteorological observations

**L279-280 and Fig. 8** Is the uncertainty of the defined lapse rate (not indicated in Fig. S3) propagated?
In any case, the upper panel (annual means) looks very much the same as in the previous version 4.
Please check if the correct dataset is plotted and include the lines indicating the standard deviation
across the individual records.

We added the uncertainty of the lapse rate to the Fig. S3, and the standard deviation to the Fig.8. The
uncertainty of the lapse rate has a seasonal cycle. It is higher in DJF (±0.2 °C/km) and lower in JJA
(±0.1 °C/km). The dataset in Fig.8 has been checked, it is correct.

[Figure]

Fig. S3: Calculated monthly mean lapse rate, based on available regional meteorological data for the
1966-1990 period. Thin lines show the uncertainty of the lapse rate estimation.

[Figure]

Fig. 8. Normalized regional temperature record based on meteorological data, with respect to the
reference period 1966-1990, expressed as annual anomalies (°C). The thin lines illustrate the standard
deviation across the individual records after accounting for the lapse rate from Fig. S3, the blue line
shows 10 year running mean, and the horizontal purple line demonstrates the decadal mean value. The
upper panel shows the annual means, the middle panel shows the warm season, and the lower panel
shows the cold season
**L280-284 (and Fig. S4)** As suggested, Fig. S4 has now been updated so that it is now consistent with
the manuscript text. When now also adding the information about the station altitude to the figure
legend, it becomes visually obvious that precipitation variability and particularly for the cold season
precipitation amount has a strong altitude effect. With this, the choice to only use the two high altitude
stations for precipitation data is clarified. The according text should be added to the manuscript.
Therefore, please add altitudes to the legends in Fig. S4 and adjust the text along the lines: "All the
precipitation data available for this region since 1966 is shown in fig. S4. Because of the obvious
altitude dependence of both precipitation variability and precipitation amount (particularly for the cold
season) only the data from the two high altitude stations Klukhorskiy Pereval (???? m asl.) and
Mineralnye Vody (???? m asl.) were used for the calculations here. The two stations are further
representative for stations with and without a prominent seasonal cycle (Mineralnye Vody and
Klukhorskiy Pereval, respectively)."
We added the altitude to the Fig. S4. We disagree with the assumption that these stations were chosen
because of their high-altitude position. The Mineralnye Vody station is situated at the altitude of 315 m
a.s.l. The reason for the choice of Mineralnye Vody station is the uninterrupted record for the whole
period of observation. Because of relatively sparse weather observations, it is difficult to estimate the altitude effect on the precipitation rate. For example, at Sulak, the highest station in the region, the precipitation rate is not the highest because of the influence of continentality. It is also an orographic effect that influence the precipitation rate. Stations situated to the South from the Caucasus receive several times more precipitation than those on the Northern slope. The precipitation rate at any of the stations is the combination of altitude, continental and orographic effects. It is difficult to calculate the influence of each of these factors.

[Figure]

[Figure]

Fig. S4: Precipitation rate in warm season (upper panel) and in cold season (lower panel). Numbers in brackets indicate the altitude of the station above the sea level.

**L437-438** It is unclear for what season NAO is correlated with regional temperature. I suggest changing to: "For the cold season, the ice core d18O record shows…"

Changed

For the cold season, the ice core d18O record shows a positive correlation with the NAO index (r = 0.41),…

**Fig. 7** There are only 11 increments for the 12 months which is confusing. To be consistent within the manuscript and with the commonly used way for display, please adjust the x-axis scale similar to manuscript Fig. 4.

Changed

[Figure]

**Fig. 9** Even though the temperature record in this figure (and in Fig. 10) is now at least plotted on the
correct age scale, it is still not correct. In the lowermost panel, not d18O is plotted but normalized
temperature (most obvious by the y-axis scale). Please correct.
Fig. 9 has been corrected

[Figure]

**Table 4 (and according sections in the text)** Although in this version, the normalized temperatures are
again used for the correlation analysis as it was suggested, none of the correlation coefficients changed.
I do not expect extreme changes but at least some considering the fact that in the previous version only
the station data from Klukhorskiy Pereval and Mineralnye Vody was used. Further, for the field in the
upper left corner, annual means - T vs d18O – the value of 0.16 (n=100) is not significant and should
not be bold.

Checked and corrected. The same correlation coefficients were obtained due to very close agreement of
temperature changes at all the stations and low uncertainty of the lapse rate calculation.
**Language:**
When accepted for publication, please take advantage of the language editing service offered by the
journal. One easy fix: in many cases $\delta$18O should be replaced with $\delta^{18}O$.
The language has been checked by the native speaker once again. The spelling of $\delta^{18}O$ has been
corrected.

[revised manuscript text omitted]